# A fast ceramic mixed OH⁻/H⁺ ionic conductor for low temperature fuel cells

Peimiao Zou [1], Dinu Iuga [2], Sanliang Ling [3], Alex J. Brown [1], Shigang Chen[1], Mengfei Zhang [1], Yisong Han [2], A. Dominic Fortes [4], Christopher M. Howard [4] & Shanwen Tao [1,5] ✉

Low temperature ionic conducting materials such as OH⁻ and H⁺ ionic conductors are important electrolytes for electrochemical devices. Here we show the discovery of mixed OH⁻/H⁺ conduction in ceramic materials. $SrZr_{0.8}Y_{0.2}O_{3-\delta}$ exhibits a high ionic conductivity of approximately 0.01 S cm⁻¹ at 90 °C in both water and wet air, which has been demonstrated by direct ammonia fuel cells. Neutron diffraction confirms the presence of OD bonds in the lattice of deuterated $SrZr_{0.8}Y_{0.2}O_{3-\delta}$. The OH⁻ ionic conduction of $CaZr_{0.8}Y_{0.2}O_{3-\delta}$ in water was demonstrated by electrolysis of both $H_2^{18}O$ and $D_2O$. The ionic conductivity of $CaZr_{0.8}Y_{0.2}O_{3-\delta}$ in 6 M KOH solution is around 0.1 S cm⁻¹ at 90 °C, 100 times higher than that in pure water, indicating increased OH⁻ ionic conductivity with a higher concentration of feed OH⁻ ions. Density functional theory calculations suggest the diffusion of OH⁻ ions relies on oxygen vacancies and temporarily formed hydrogen bonds. This opens a window to discovering new ceramic ionic conducting materials for near ambient temperature fuel cells, electrolysers and other electrochemical devices.

Fuel cells are important electrochemical devices to convert chemical energy in fuels into electricity at high efficiency. There are different types of fuel cells based on various ionic conducting electrolytes, with a wide operating temperature from near ambient temperature (≤100 °C) (NAT) e.g., around 80 °C for proton exchange membrane fuel cells (PEMFCs), to high temperature such as 500–900 °C for solid oxide fuel cells (SOFCs)[1,2]. SOFCs have been developed for stationary power generation while poor durability, due to materials sublimation and cross-diffusion at high operating temperatures, is the major obstacle for the large-scale applications[3]. The typical electrolyte materials for SOFCs are doped $ZrO_2$ or $CeO_2$ with fluorite structures or doped $BaCeO_3$ (BCO)/$BaZrO_3$ (BZO)/$LaGaO_3$ (LGO) with perovskite structures[1,4–12]. The representative high temperature proton conductors based on doped perovskite oxides BCO/BZO were discovered in 1980 and 1990s respectively[4,8], while the high temperature O²⁻ ion conductor based on doped LGO was discovered in 1994[5,13]. However,

the ionic conductivity of these oxides is not high enough for use in applications when the operating temperature is below 500 °C[1]. The minimum required ionic conductivity is approximately 0.01 S cm⁻¹ for use as electrolytes for fuel cells or electrolysers when thin film technology is applied[1]. To solve these problems, it is highly desirable to reduce the operating temperature of SOFCs, ideally to NAT. The first task to achieve NAT operation of SOFCs is to identify good H⁺ / O²⁻ / OH⁻ ionic conducting materials to be used as the electrolyte.

It has been reported that doped zirconates such as $BaZr_{0.8}Y_{0.2}O_{3-\delta}$ (BZYO20) are robust proton conductors with ionic conductivity above 10⁻⁴ S cm⁻¹ at a temperature above 600 °C[8,14,15]. In high temperature proton conductors, BZYO20 exhibits excellent chemical stability in the presence of steam and $CO_2$[9,16]. However, the proton conductivity of perovskite zirconates are normally measured in wet air or wet $H_2$ by passing the gases through room temperature water with moisture contents of 3 mol%[14]. The steam partial pressure may not be high enough to

[1]School of Engineering, University of Warwick, Coventry CV4 7AL, UK. [2]Department of Physics, University of Warwick, Coventry CV4 7AL, UK. [3]Advanced Materials Research Group, Faculty of Engineering, University of Nottingham, Nottingham NG7 2RD, UK. [4]ISIS Neutron and Muon Spallation Source, Rutherford Appleton Laboratory, Harwell Science and Innovation Campus, Chilton, Oxfordshire OX11 0QX, UK. [5]Department of Chemical Engineering, Monash University, Clayton, Victoria 3800, Australia. ✉e-mail: S.Tao.1@warwick.ac.uk

achieve the highest possible ionic conductivity. On the other hand, the $OH^-$ ionic conductivity of polymeric alkaline membranes is measured in water[17]. For oxide ionic conductors, a high open circuit voltage (OCV) has been reported for electrochemical cells using either 8 mol% $Y_2O_3$ stabilised $ZrO_2$ or $Ce_{0.8}Sm_{0.2}O_{2-\delta}$ as the electrolyte while one side is submerged in water, although the real mechanism for this high OCV is still unclear[18]. To the best of our knowledge, reports measuring the ionic conductivity of oxides in liquid water are scarce. In this work, we find that for Y-doped $AZrO_3$ (A = Ca, Sr, Ba), $SrZr_{0.8}Y_{0.2}O_{3-\delta}$ (SZYO20) exhibits the highest ionic conductivity of approximately 0.01 S cm$^{-1}$ at 90 °C when exposed in water or humidified air, which opens the window to develop low temperature ceramic $OH^-$ ionic conductors and makes it possible to develop solid oxide fuel cells with an ambient operating temperature, i.e., NAT-SOFCs. Some of the main problems of conventional SOFCs with operating temperatures above 500 °C, such as materials sublimation and cross-diffusion, should be alleviated due to the significantly reduced temperatures of NAT operation.

## Results

### Creation of pathways for ions in perovskite oxide $SrZr_{1-x}Y_xO_{3-\delta}$

For a good ionic conductor, it is necessary to have possible pathways for the diffusion of ions. With the existence of pathways, the concentration of charge carriers must be sufficient in order to achieve high ionic conductivity. For doped zirconates, it is widely accepted that the oxygen vacancies and the associated proton defects during the reaction between oxygen vacancies and steam (or water) form the pathways for the proton conduction[19,20]. A schematic diagram showing the formation of pathways in a perovskite oxide is shown in Fig. 1. Perovskite oxide $SrZrO_3$ (SZO) was chosen as the parent phase. When part of Zr at the B-site in SZO is replaced by Y, for charge compensation, positively charged oxygen vacancies ($V_O^{\bullet\bullet}$) are formed. For charge balance, the negatively charged $Y'_{Zr}$ defects are formed simultaneously. It can be described by Kröger-Vink notations.

$$2Zr_{Zr}^{\times} + O_O^{\times} + Y_2O_3 \rightarrow 2ZrO_2 + 2Y'_{Zr} + V_O^{\bullet\bullet} \quad (1)$$

When this Y doped SZO, with a general formula of $SrZr_{1-x}Y_xO_{3-\delta}$, meets water or steam, proton defects $OH_O^{\bullet}$ will be formed.

$$V_O^{\bullet\bullet} + O_O^{\times} + H_2O \rightarrow 2OH_O^{\bullet} \quad (2)$$

At high temperatures, typically above 600 °C, the mobility of formed proton defects increases leading to high protonic conductivity in doped BCO/BZO[4]. In a previous study, density functional theory (DFT) calculations indicate that proton migration in the perovskite $BaZr_{0.1}Ce_{0.7}Y_{0.1}Yb_{0.1}O_{3-\delta}$ is in the form of $OH^-$ in the presence of oxygen vacancies[21]. This means, under certain conditions, $OH^-$ ions can also migrate in perovskite oxides.

The ionic conductivity of Y-doped SZO, $SrZr_{0.95}Y_{0.05}O_{3-\delta}$ (SZYO05) is approximately $7.0 \times 10^{-4}$ S cm$^{-1}$ at 600 °C, which is not high enough for use as electrolyte for NAT-SOFCs[22]. In order to improve the ionic conductivity at low temperatures, it is necessary to increase the concentration of charge carriers, i.e., the proton defects. Therefore, we synthesized $SrZr_{1-x}Y_xO_{3-\delta}$ (x = 0, 0.1, 0.2) samples (Fig. 2a and Methods) and put the samples in liquid water to maximise the interaction between water and oxygen vacancies to maximise the concentration of proton defects. When the concentration of these proton defects is high enough, the neighbouring proton defects may couple or exchange with each other, in the form of $H^+$ or $OH^-$ ions, forming a continuous pathway, resulting in high $OH^-/H^+$ ionic conductivity. It is presumed the formed proton defects may exchange charges with the neighbouring negatively charged ions such as $O^{2-}$ ions or defects such as $Y'_{Zr}$ to form $OH^-$ ions. Under certain conditions, these $OH^-$ ions may move along the pathways then the ceramic material will be an $OH^-$ ionic conductor. The possibility of $OH^-$ ion formation in perovskite oxides has been reported by theoretical calculations in previous studies[23–27] although high $OH^-$ ionic conducting ceramic materials have not been discovered yet. In this study, the high mixed $OH^-/H^+$ ionic conductivities in hydrated ceramic oxides SZYO20 and $CaZr_{0.8}Y_{0.2}O_{3-\delta}$ (CZYO20) are reported.

### Materials characterizations of SZYO20

Based on the analysis above, it is anticipated that, doped zirconates may exhibit high ionic conductivity in water or, in the presence of high partial pressure of steam, $pH_2O$. At 100 °C, the reported bulk proton conductivity of BZYO20 is in the order of $10^{-5}$ S cm$^{-1}$ in air, too low to be used as an electrolyte for NAT-SOFCs[15]. In reported papers, at the same temperature, generally doped BZO exhibits higher ionic conductivity than doped SZO because doped BZO has a larger lattice volume thus more 'free volume' for diffusion/jumping of $O^{2-}$ ions in the lattice[15,22]. The diffusion of point defect protons does not require high 'free volume' but may need the right binding energy. Similar to $O^{2-}$ ions, the diffusion of large $OH^-$ ions also needs relatively large 'free volume'. To identify the effect of A-site ions on the ionic conductivity of Y-doped zirconates, three compounds with composition $AZr_{0.8}Y_{0.2}O_{3-\delta}$, where A = Ca, Sr, Ba were synthesized. X-ray diffraction (XRD) experiments indicate sample $CaZr_{0.8}Y_{0.2}O_{3-\delta}$ (CZYO20) is a single phase, BZYO20 has a negligible amount of reported second phase $Y_2O_3$[28,29], while there is a small amount of secondary phase $SrY_2O_4$ in SZYO20 (Supplementary Fig. 1 and Supplementary Table 1). Sample $SrZr_{0.9}Y_{0.1}O_{3-\delta}$ (SZYO10) also contains a small amount of secondary phase $SrY_2O_4$ indicating reducing the Y-doping level cannot eliminate it. The $SrY_2O_4$ secondary phase was also observed in SZYO20 when synthesised by pulsed layer deposition method[30] and commonly exists in Y doped perovskite oxides such as $Sr(Ce_{0.6}Zr_{0.4})_{0.8}Y_{0.2}O_3$[31] and $Sr(Ti_{0.9}Y_{0.1})O_3$[32]. A possible reason is, $SrY_2O_4$ is a stable phase, which can be easily formed at a relatively low temperature before the formation of perovskite phase. Once it is formed, it is difficult to get rid of. The primary particle size of SZYO20 was about 200–250 nm and an agglomeration of some particles was observed by both scanning electron microscopy (SEM) and annular dark field – scanning transmission electron

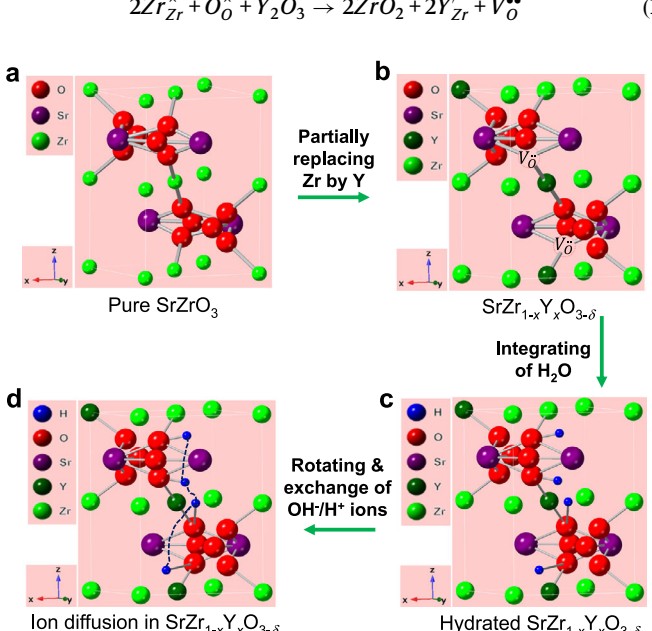

**Fig. 1 | The proposed schematic diagram for the formation of pathways for $OH^-$/$H^+$ ions in $SrZr_{1-x}Y_xO_{3-\delta}$ enabled by water/steam. a** The structure of $SrZrO_3$ parent phase. **b** The schematic structure of Y doped $SrZrO_3$. Part of Zr at the B-site in a is replaced by Y then oxygen vacancies ($V_O^{\bullet\bullet}$) are formed. **c** The schematic structure of hydrated $SrZr_{1-x}Y_xO_{3-\delta}$. Proton defects $OH_O^{\bullet}$ are formed in $SrZr_{1-x}Y_xO_{3-\delta}$ after integrating of water or steam. **d** The schematic diagram for $OH^-$/$H^+$ ions diffusion in $SrZr_{1-x}Y_xO_{3-\delta}$.

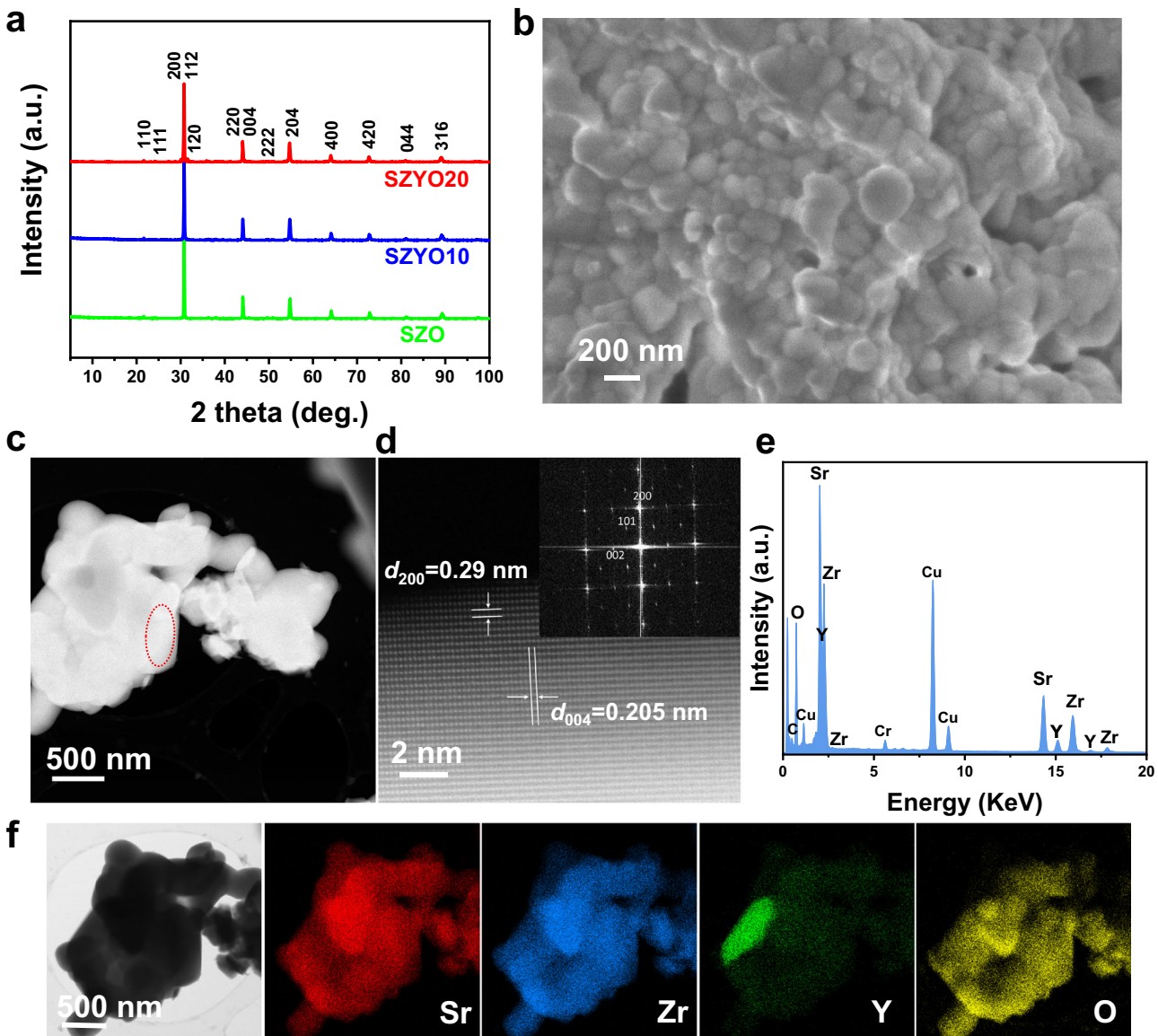

**Fig. 2 | The characterizations of doped zirconates. a** XRD patterns of SrZr$_{1-x}$Y$_x$O$_{3-\delta}$ ($x$ = 0, 0.1, 0.2, denoted as SZO, SZYO10, SZYO20). **b–f** SEM and TEM analysis of SZYO20 powders. **b** SEM image, **c** ADF-STEM image, **d** High-resolution ADF-STEM image taken from marked area in (**c**) (inset: corresponding FFT of the image), **e** Integrated EDX spectra from the STEM-EDX analysis in (**f**), f BF-STEM image and the corresponding EDX maps of Sr, Zr, Y, O.

microscopy (ADF-STEM) (Fig. 2b, c). In Fig. 2d, the two labelled spacings were associated with the (200) and (004) lattice planes of the orthorhombic phase of SrZrO$_3$ (ICDD:04-014-8276), respectively. The Energy dispersive X-ray spectroscopy (EDX) elemental maps clearly show that the amount of secondary phase is small and Sr, Zr, Y and O elements are distributed evenly in the grains (Fig. 2e, f). The Y-rich area of the SZYO20 sample was confirmed to be SrY$_2$O$_4$ phase (Supplementary Fig. 2). The spacing labelled in Supplementary Fig. 2c was associated with the (200) lattice plane of the orthorhombic phase of SrY$_2$O$_4$ (ICDD:01-074-0264).

### Removal of secondary phase SrY$_2$O$_4$ from the samples

It has been reported that the secondary phase of the sample, SrY$_2$O$_4$, may react with water at high temperatures to form SrCO$_3$, Sr(OH)$_2$·8H$_2$O and gel phase Y(OH)$_3$[33]. Therefore, before the conductivity measurements, SrZr$_{1-x}$Y$_x$O$_{3-\delta}$ samples were heated in hot water at 90 °C for 20 h and washed three times in order to remove as much secondary phase SrY$_2$O$_4$ and the soluble hydrated product Sr(OH)$_2$·8H$_2$O from the sample as possible. From STEM and EDX

analysis of pre-washed SZYO20 samples (Fig. 3a–c and Supplementary Fig. 3), the measured spacing in Fig. 3c was associated with the (002) lattice plane of the orthorhombic phase of SrZrO$_3$ (ICDD:04-014-8276). This indicates the perovskite phase is stable after hot water treatment at 90 °C while the secondary phase SrY$_2$O$_4$ has been almost removed. The washed sample contains ~2% SrCO$_3$ which is the residual product of the reaction between SrY$_2$O$_4$ and hot water[33]. Sr(OH)$_2$·8H$_2$O was not observed either in XRD (Supplementary Table 1) or STEM/EDX observation because it is highly soluble in water. The quantity of residual Sr(OH)$_2$·8H$_2$O should be very small if there is any. In addition, STEM and EDX results indicate the presence of an amorphous material in the Y-rich area B (Supplementary Fig. 3b, c), which is likely to be Y(OH)$_3$. This amorphous material stays on the grain boundary or presents as isolated particles among perovskite particles (marked area in Fig. 3a). The elemental analysis of SZYO20 samples before and after being washed in hot water indicated that the Sr-content slightly decreased for the washed sample, confirming the soluble hydrolysis products Sr(OH)$_2$·8H$_2$O had been successfully removed (Supplementary Table 2). The hydrolysis of SrY$_2$O$_4$ at 90 °C was also confirmed by XRD

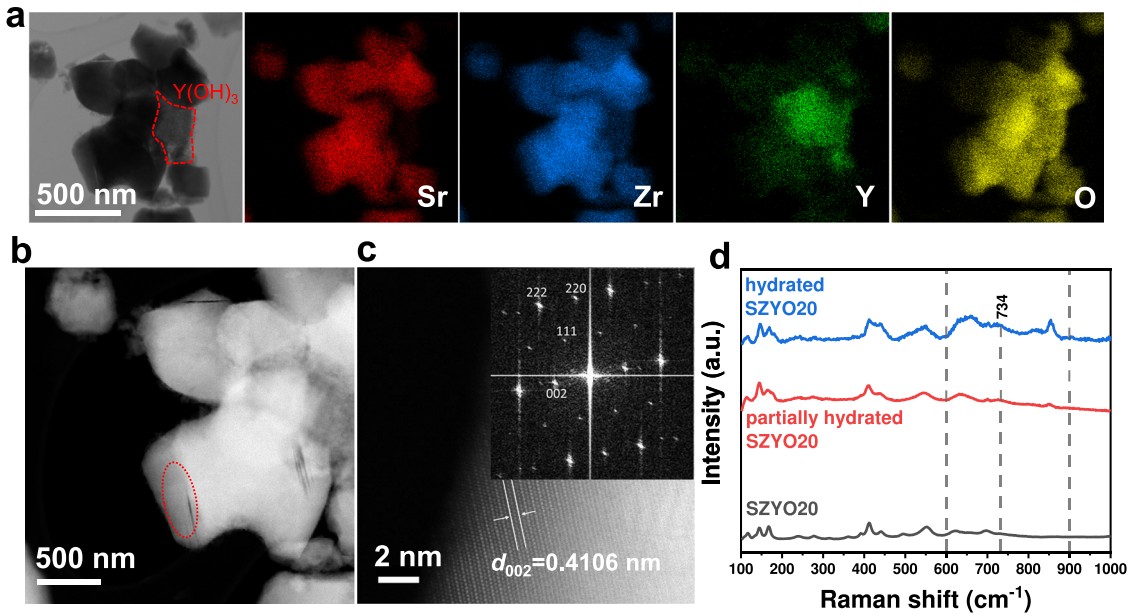

**Fig. 3 | TEM and Raman characterizations of washed SZYO20. a–c** TEM analysis of washed SZYO20 powders. **a** BF-STEM image and the corresponding EDX maps of Sr, Zr, Y, O, **b** ADF-STEM image, **c** High resolution ADF-STEM image taken from marked area in (**b**) (inset: corresponding FFT of the image). **d** Raman spectra of SZYO20 samples with different hydration levels.

analyses (Supplementary Fig. 4). All samples for conductivity measurements were treated in water at 90 °C for 20 h (repeated three times) to remove as much $SrY_2O_4$ as possible to minimise the possible contribution of $SrY_2O_4$ and its hydrolysed product $Sr(OH)_2 \cdot 8H_2O$ on the measured conductivity. However, we do not rule out that the trace residual $Sr(OH)_2$ from hydrolysis of $SrY_2O_4$ may provide feeding $OH^-$ ions to SZYO20, which contributes to the conductivity to some extent.

## Importance of hydration and conductivity measurement

The measurement of traditional protonic conducting perovskite oxides such as doped zirconates or cerates are normally measured in wet air or wet $H_2$, passing the gas through room temperature water with the steam partial pressure of 0.03 bar, i.e., 3 mol% steam in the humidified air or hydrogen. The measured conductivity at NAT is very low[15]. According to our analysis above (Fig. 1 and Eq. 1 & 2), the water or steam concentration mentioned in reported papers may not be enough to form sufficient proton defects in order to form continuous pathways for $OH^-/H^+$ ions. Therefore, we decided to directly measure the conductivity and ion transfer number of the prepared and pre-washed $AZr_{0.8}Y_{0.2}O_{3-\delta}$ (A = Ca, Sr, Ba) pellets in water (Supplementary Fig. 5), which is the method commonly used for conductivity measurement in polymeric alkaline membranes[17]. In our study, at room temperature, the conductivity of un-washed SZYO20 pellet in water kept increasing against time (Supplementary Fig. 6). It increased from $2.96 \times 10^{-4} S\,cm^{-1}$ to $2.28 \times 10^{-3} S\,cm^{-1}$ in 10 h indicating hydration of the oxide to form proton defects is very important in order to achieve high ionic conductivity (Eq. 2). After aging in water at room temperature, the ionic conductivity of SZYO20 in water, is already higher than the protonic conductivity of SZYO05, $7.0 \times 10^{-4} S\,cm^{-1}$, at 600 °C when measured in $H_2$[22]. The conductivity of pure $SrY_2O_4$ in water at room temperature remains around $1.7 \times 10^{-4} S\,cm^{-1}$ against time, only ~7.5% of that for fully hydrated SZYO20 (Supplementary Fig. 6c), indicating the increased conductivity of the un-washed SZYO20 is mainly due to the hydration of SZYO20, or in other words, the interaction between SZYO20 and liquid water. This experiment indicates it is difficult to discover the real high ionic conductivity of hydrated SZYO20 if the duration of submerging the pellet in water is not long enough.

To rule out the possible contribution of residual $Y_2O_3$ or $Y(OH)_3$ in the pre-washed SZYO20 sample, conductivities of pure $Y_2O_3$ and $Y(OH)_3$ in water were measured indicating they have very low ionic conductivity, in the range of $10^{-5} \sim 10^{-4} S\,cm^{-1}$ at 90 °C in water (Supplementary Fig. 7). This means the observed Y-rich phase in STEM (Fig. 3a) will have little contribution to the observed overall high ionic conductivity of SZYO20 at elevated temperatures.

The interaction between oxygen vacancies and water was also confirmed by Raman spectra (Fig. 3d). The peaks in the 600–900 $cm^{-1}$ region are slightly wider when the SZYO20 sample is wetter. When the hydration level of SZYO20 sample is higher, the Raman peak shifts towards higher values (734 $cm^{-1}$), which is consistent with the Raman features of proton insertion in oxygen vacancies in Yb-doped SZO and BZO[34].

The jumping or diffusion of ions in a lattice, is not only related to the high concentration of proton defects and the formed $OH^-$ species when it is heavily hydrated, but also to the mobility of ions in a given lattice, and in some cases, this is correlated with the 'free volume' (for large $OH^-$ ions) and jumping distance between neighbouring available sites (Fig. 1). In general, larger lattice parameters or bond lengths results in higher 'free volume', which may help to reduce the transition barrier of ionic diffusion, therefore favouring the mobility of large $OH^-$ ions, while longer jumping distance for ions will reduce the mobility. These two effects are opposite on the ionic conductivity. Therefore, there must be an optimal lattice size, which compromises between 'free volume' and jumping distance, exhibiting the highest ionic conductivity. It was found that sample SZYO20 exhibits the highest ionic conductivity in the $AZr_{0.8}Y_{0.2}O_{3-\delta}$ (A = Ca, Sr, Ba) series (Fig. 4a and Supplementary Fig. 8), which is consistent with the observed proton conductivity of $AZr_{0.95}In_{0.05}O_{3-\delta}$ (A = Ca, Sr, Ba) in $H_2$ at a temperature of 600 – 1000 °C in which $SrZr_{0.95}In_{0.05}O_{3-\delta}$ exhibits the highest conductivity[8]. There is also a possibility that mobile ions like $OH^-$ diffuse through the grain boundary to promote ionic conductivity. To understand whether the ion transfer in both bulk and grain boundaries contributes to the conductivity, electrochemical impedance spectrum (EIS) was acquired up to a high frequency of 10 MHz, and equivalent electrochemical circuits were fitted and simulated to extract the bulk

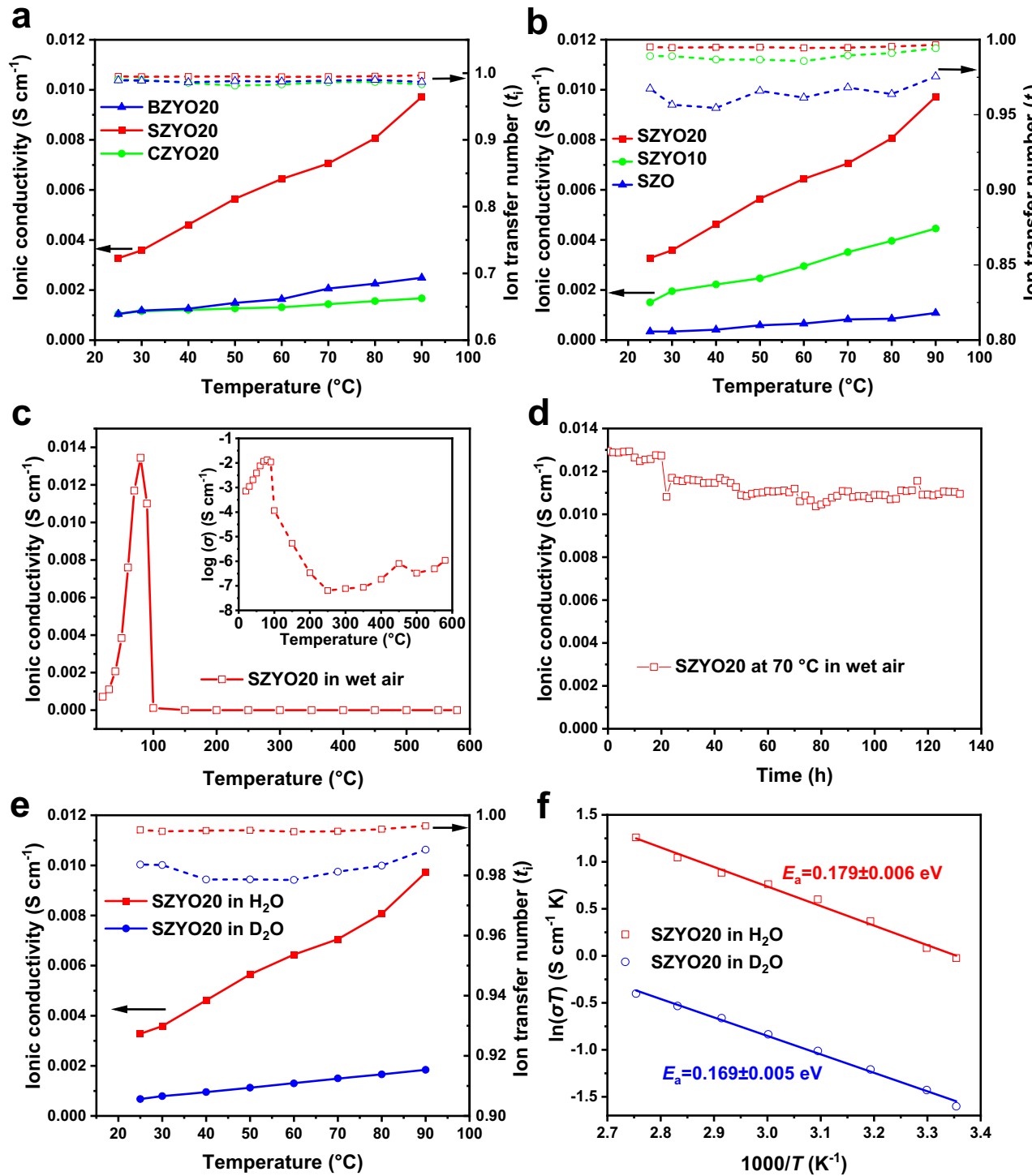

**Fig. 4 | The ionic conductivity of doped zirconates under different conditions.** **a** $AZr_{0.8}Y_{0.2}O_{3-\delta}$ (A = Ca, Sr, Ba, denoted as CZYO20, SZYO20, BZYO20), measured in water. **b** $SrZr_{1-x}Y_xO_{3-\delta}$ (x = 0, 0.1, 0.2, denoted as SZO, SZYO10, SZYO20), measured in water. **c** SZYO20 in wet air at different temperatures. **d** Stability of the conductivity of SZYO20 in wet air at 70 °C. **e**, **f** Conductivity (**e**) and activation energy (**f**) of SZYO20 in $H_2O$ and $D_2O$.

and grain boundary resistances. In order to observe a complete grain boundary response, the impedance spectrum was also recorded at −25 °C (Supplementary Fig. 9). All the impedance dataset was validated by performing Kramers-Kronig analyses, i.e., the real and imaginary residuals are lower than 1%[35,36]. It is interesting that at −25 °C the bulk conductivity of BZYO20 is the highest as expected due to its larger lattice volume, but when the conductivity test temperature reached room temperature and above (i.e., liquid water environment), not only

the grain boundary conductivity but also the bulk conductivity of SZYO20 increases rapidly, both of which contribute to the highest total conductivity among the three $AZr_{0.8}Y_{0.2}O_{3-\delta}$ (A = Ca, Sr, Ba) oxides (Supplementary Fig. 8c, d). This shows to a certain extent that the conductive ions not only diffuse in the perovskite lattice but may also transport through the grain boundaries. After hydration, the presence of hydrogen and additional oxygen atoms from hydroxide filling vacancies in the bulk of SZYO20 has been identified in later neutron

diffraction discussions, which verified the bulk diffusion occurring in the perovskite lattice. On the other hand, although we believe the very trace amounts of $Sr(OH)_2$ and $Y(OH)_3$ residues in the washed sample are not enough to directly provide sufficient conductivity, we do not rule out that the free unbonded proton and $OH^-$ ions in the environment use their O-H channels to transport across the grain boundaries and therefore contribute to a high conductivity of SZYO20. At high temperatures, the reported proton conductivity in doped BZO is generally higher than that of doped SZO. However, when the conductivity of these oxides was measured in water, in the $AZr_{0.8}Y_{0.2}O_{3-\delta}$ (A = Ca, Sr, Ba) series, it is SZYO20 that exhibits the highest ionic conductivity. This implies the charge carriers may not be protons or may not be dominated by protons. When the charge carriers are different, the conduction mechanism could be different thus may not necessarily follow the mechanism for proton diffusion/jumping at high temperature. Actually, in our later concentration cell measurements, it has been demonstrated that heavily hydrated SZYO20 is a mixed $OH^-/H^+$ conductor while $OH^-$ ions are the dominant charge carrier (Supplementary Table 3). This is very different from the previously reports that, at high temperature, doped BZO and SZO are proton conductors. The same materials, when exposed to different environments, can have distinct charge carriers which contribute to the conductivity.

A low temperature concentration cell measurement was applied to determine which charge carriers are the conduction ions (Supplementary Fig. 10 and Supplementary Note 1)[37]. According to the measured ion transport number of SZYO20, at room temperature, about 70% of the measured ionic conductivity is due to the transfer of $OH^-$ ions (Supplementary Table 3). The proton transference number was determined to be -0.27 at room temperature (Supplementary Fig. 11), which is consistent with the cation transport number measured by the concentration cell. This indicates heavily hydrated SZYO20 is a mixed $OH^-/H^+$ ion conductor and predominantly presents $OH^-$ conduction, which is very different from the previous knowledge of conventional doped zirconates, which are considered as mixed $H^+/O^{2-}$ ionic conductors above 500 °C depending on the temperature range[38]. In this study, it has been observed that, not only protons but $OH^-$ ions may also jump or diffuse via the oxygen vacancies or proton defects after hydration with liquid water. Hydroxide ionic conduction was also previously proposed in similar perovskite oxides $SrCe_{0.95}Yb_{0.05}O_{3-\delta}$ (SCYb) and $BaCe_{1-x}Gd_xO_{3-\delta}$ (BCG) in the presence of water vapour at high temperature (> 500 °C) while the observed ionic conduction was very low because $p$H$_2$O is not high enough[23,39]. The reason for this difference is, the conductivity is measured under different conditions. The ionic conductivity of SZYO20 in this study was measured in water or heavily humidified gas (passing air through boiling water at 100 °C) while in previous studies, the conductivity was typically measured in an atmosphere passing room temperature water, as stated above.

In most oxides that are high temperature mixed $O^{2-}/H^+$ ionic conductors, such as the doped BZO/BCO perovskites, oxygen vacancies play a critical role for high ionic conductivity[1]. To determine the role of oxygen vacancies on the resulting high ionic conductivity of SZYO20, SZO and SZYO10 were also synthesised. XRD results indicate that SZO is a single phase, while SZYO10 contains a small amount of secondary phase $SrY_2O_4$ (Supplementary Figs. 1c, 12). Both SZO and SZYO10 were pre-washed at 90 °C in water for three times to remove any soluble impurities and minor $SrY_2O_4$ secondary phase. As shown in Fig. 4b, among the three oxides, the order of conductivity and ionic transfer number is SZYO20 > SZYO10 > SZO. This is because SZYO20 has the highest doping level thus the highest concentration of oxygen vacancies (Eq. 1), thus more proton defects (Eq. 2), which means there will be more $OH^-$ species formed through charge exchange with neighbouring negatively charged $O^{2-}$ ions or $Y'_{Zr}$ defects, leading to the highest ionic (both $OH^-$ and $H^+$ ions) conductivity. The same phenomenon was observed in Y doped BZO and $CaZrO_3$ (CZO). For oxides with negligible or no secondary phases, such as BZO, BZYO20, CZO, and

CZYO20, the conductivity of the Y-doped oxides in water is much higher (approximately 6 times) than that of un-doped ones (Supplementary Fig. 13a–d). This indicates the importance of extrinsic oxygen vacancies in order to achieve high ionic conductivity in water. It can be reasonably deduced that the high ionic conductivity is mainly due to the deliberately introduced extrinsic oxygen vacancies through Y-doping, rather than the residual $Sr(OH)_2\cdot8H_2O$. In addition, it is worth noting that, without $SrY_2O_4$ secondary phase, the conductivity of pure SZO is also high, exhibiting the highest conductivity among $AZrO_3$ (A = Ca, Sr, Ba) as shown in Supplementary Fig. 13e, which means SZO is the lattice with optimised 'free volume' and jumping distance for $OH^-$ ions. It is possible there could be a small amount of intrinsic oxygen vacancies in the fired pure oxides such as CZO/SZO/BZO, which allows the formation of proton defects according to Eq. 2 and the diffusion of generated $OH^-$ ions. However, we cannot completely rule out the possibility that the alkaline elements Ca, Sr or Ba in the perovskite might slowly dissolve in water to form metal hydroxide contributing to the measured conductivity of the oxides. Therefore, we measured the pH value and conductivity of the water used in the conductivity measurement, immediately after the completion of ionic conductivity measurement by a.c. impedance. The conductivity of the used water is always much lower than the measured conductivity of pellets except for BZO which has the lowest conductivity in water (Supplementary Table 4). For SZYO20, at room temperature, the total conductivity of used water was 407.1 µS cm$^{-1}$, which is only 19% of the total conductivity of SZYO20 pellet, 2174.98 µS cm$^{-1}$. The contribution of any species from slow dissolution of the perovskite in water to the measured total conductivity of SZYO20 is negligible. Therefore, the majority of the measured conductivity of the oxide pellets is from the hydrated oxide itself while any other species from the dissolved perovskite oxide in water may have a small contribution to the measured total conductivity, presuming water with the dissolved species enters the pores of the oxide pellets.

In water, the ionic conductivity of sample SZYO20 is $3.28 \times 10^{-3}$ S cm$^{-1}$ at 25 °C, and $9.71 \times 10^{-3}$ S cm$^{-1}$ at 90 °C respectively with ion transfer numbers higher than 0.995 (Fig. 4a). This indicates it is a predominantly ionic conductor when exposed in water. This is consistent with the knowledge of solid state ionic conductors as there are no multi-valent elements in these oxides. The ionic conductivity and transfer number of SZYO20 is sufficient to be used as electrolytes for fuel cells and electrolysers at 90 °C when thin-film technology used for conventional SOFCs is applied[1]. When only considering its ability to transport hydroxide ions, the $OH^-$ conductivity of SZYO20 is close to that of some polymer-based alkaline membranes as shown in Supplementary Table 5.

For fuel cell applications, the cathode side may be exposed to humidified $O_2$ or humidified air such as conventional PEMFCs based on acidic Nafion membrane electrolyte. Therefore, it is very important to measure the ionic conductivity in humidified air. In conventional proton conductivity measurements, the samples are passing through room temperature water with ~3% water vapour. In our study, air was passing through 100 °C boiling water in order to fully humidify the oxides. The real temperature of the oxide pellet was recorded by a thermocouple next to the sample. At 90 °C, the conductivity of SZYO20 in as-humidified air is $1.1 \times 10^{-2}$ S cm$^{-1}$ while it suddenly drops at above 100 °C (Fig. 4c and Supplementary Fig. 14a–c). This means the high ionic conductivity of SZYO20 is related to the presence of liquid water either in pure water or a mixture of liquid water and stream as the case of humidified air by passing air through 100 °C boiling water. At 70 °C, the conductivity of SZYO20 in wet air is stable at 0.01 S cm$^{-1}$ for the measured 130 h (Fig. 4d), which is reflected in the a.c. impedance spectra (Supplementary Fig. 14d, e). This indicates the ceramic ionic conductor SZYO20 has excellent stability in humidified air. After conductivity measurements, the chemical composition was still perovskite oxide, confirmed by both XRD and element mapping

(Supplementary Fig. 15). Pre-washed SZYO20 sample contains 2% residual $SrCO_3$ due to hydrolysis of second phase $SrY_2O_4$ and the $SrCO_3$ remains 2% in SZYO20 sample after conductivity measurement in humidified air at 70 °C for 130 h indicating excellent chemical compatibility with $CO_2$ in air. If there is any residual $SrY_2O_4$, it may hydrolyse to form $Sr(OH)_2 \cdot 8H_2O$ during the conductivity stability measurement then the yielded $Sr(OH)_2 \cdot 8H_2O$ may further react with $CO_2$ in humidified air to form $SrCO_3$, leading to increased $SrCO_3$. The consistency of $SrCO_3$ content before and after the conductivity measurement indicates there is negligible residual $SY_2O_4$ or $Sr(OH)_2 \cdot 8H_2O$ in the pre-washed sample.

The excellent chemical stability of SZYO20 in humidified air is anticipated. It is well-known that BZYO20 is a stable proton conductor, which has been widely investigated as electrolyte for SOFCs, due to its excellent chemical stability in steam and $CO_2$[16,40]. Theoretically, in $AZrO_3$ where A = Ca, Sr, Ba, the order of chemical stability is BZO < SZO < CZO because the large $Ba^{2+}$ ions are the most polarised while the small $Ca^{2+}$ ions are the least polarised. From this point of view, SZYO20 should exhibit higher chemical stability than BZYO20. This is consistent with the observed excellent chemical stability of SZYO20 in humidified air, which means 'free' air can be directly used as the oxidant in fuel cells when humidified SZYO20 is used as the electrolyte. In conventional alkaline fuel cells based on KOH solution electrolyte or, polymeric alkaline membrane fuel cells based on quaternary ammonium groups, air cannot be directly used as the oxidant at the cathode of alkaline fuel cells (AFCs) or alkaline membrane fuel cells (AMFCs) because $CO_2$ may react with KOH to form $K_2CO_3$ which may block the holes in electrode in AFCs or, react with polymeric alkaline membrane with significantly reduced $OH^-$ ionic conductivity in AMFCs. The discovery of high $OH^-$ ionic conductivity in ceramic oxide SZYO20 will provide a low temperature $OH^-$ ionic conductor to be used as electrolyte for low temperature fuel cells such as NAT-SOFCs or electrolysers for splitting of water for hydrogen production. This eliminates the problem of poor $CO_2$ compatibility of the electrolyte in AFCs and AMFCs.

### Kinetic isotope effect

For proton conducting materials, the conductivity in $D_2O$ will be reduced due to the decreased mobility of $D^+$ ions[8,41]. This is also called the kinetic isotope effect (KIE). The conductivity of SZYO20 in pure $D_2O$ was also measured to investigate its effect on proton conduction (Fig. 4e). At 90 °C, the conductivity in $H_2O$ is 5.28 time of that in $D_2O$, which is reflected in the a.c. impedance spectra in Supplementary Fig. 16. For pure proton conductors, the KIE is usually no less than 1.4 for Grotthuss mechanism, while it is close to 1.2 for vehicle mechanism[41]. As the KIE is much larger than 1.4, it is presumed that SZYO20 is not a pure protonic conductor. This is consistent with the ion transfer number measured through concentration cell, i.e., 70% of the charge carriers are $OH^-$ ions. Similarly, in a previous report, the observed $OH^-$ ionic conduction in the perovskite oxides SCYb and BCG in the presence of water vapour does not follow the kinetic isotope effect either[23,39].

The large difference between the conductivity in $H_2O$ and $D_2O$, implies the large $OH^-$ ions are the major charge carriers, while jumping or diffusion of large $OD^-$ ions could be more difficult than that for $OH^-$ ions. The activation energy of SZYO20 in $H_2O$ and $D_2O$ was 0.179 ± 0.006 eV and 0.169 ± 0.005 eV respectively (Fig. 4f). It is close to the 0.17 eV for Nafion membrane in $H_2O$[41] and within the range of activation energy for $OH^-$ conducting polymers (0.12–0.26 eV)[42]. The activation energy of doped SZO at low temperature (<100 °C) in this study is very different to that reported for doped SZO at high temperature with the range of 0.4–0.6 eV at 700–1000 °C when the dominant charge carriers are $H^+$ and $O^{2-}$ ions[22]. This indicates the charge carriers in liquid water may be different, predominantly $OH^-$ ions. This explains why the hydrated SZYO20 is a mixed $OH^-$/$H^+$

conductor, while the transfer number for $OH^-$ ions is much higher than that for cations, $H^+$ ions (Supplementary Table 3).

The particle size remained unchanged after sampleSZYO20 had been measured in water, $D_2O$ or wet air (Supplementary Fig. 17), further confirming the excellent chemical stability of SZYO20.

### Solid state NMR measurements

In order to further investigate the conduction mechanism of SZYO20 in water, solid-state nuclear magnetic resonance (NMR) has been employed to study the dry and partially hydrated SZYO20. Figure 5a shows the solid state $^1H$ NMR spectra of dry (red) and partially hydrated (blue) SZYO20. The $^1H$ spectrum of dry SZYO20 shows three resolved signals at 4.4, 3.5 and 0.7 ppm that can be assigned to $H_2O$, $OH^-$ groups bound on defects or surface and, $H^+$ bound to oxygen of the Sr-O-Y or Sr-O-Zr environments, respectively. Akin assignments of proton signals on similar samples have been reported[43–45]. As the sample gets partially hydrated the $^1H$ signal increases dramatically and shows two broad peaks at 4.6 and 2.3 ppm which can be assigned to water at 4.6 ppm and to signals from $OH^-$ and $H^+$ moieties and their exchange at 2.3 ppm. The deconvolution of the partially hydrated sample is shown on Supplementary Fig. 18. The fast MAS experiments were performed with a 1.3 mm probe that has limited variable temperature capabilities. Figure 5b shows the solid state $^1H$ NMR (MAS 60 kHz) spectra of partially hydrated (top) and dry (bottom) SZYO20 measured with the sample at −5 °C (blue) and +30 °C (red). On both samples, the water peak shifts slightly with the temperature (approximately 0.1 ppm / 10 °C)[46]. At both temperatures, the dry and the wet samples, the signals assigned to $OH^-$ and to $H^+$ are getting broader as the temperature increases which suggests the presence of exchange between the two moieties. $^1H$ nuclear overhauser effect spectroscopy (NOESY) spectra of the dry and hydrated samples are shown in Fig. 5c, d and prove proximity between different $^1H$ moieties.

To further investigate the temporary formation of hydrogen bond and its disappearance above 100 °C, the $^1H$ MAS spectra at 25, 95 and 125 °C on partially hydrated SZYO20 were recorded with a 1.6 mm HXY Phoenix NMR probe spinning at 30 kHz (Supplementary Fig. 19). The sample used for high temperature measurements was slightly less hydrated than the one used to record the $^1H$ spectra shown in Fig. 5a such that the peak at 0.7 ppm is visible on the partially hydrated sample. At higher temperatures the intensity of this peak, assigned to $H^+$, increases and clearly becomes narrower at above 100 °C. At 125 °C the position of the peak is slightly shifted to 0.8 ppm. This shift can be clearly identified on the spectra measured at 95 °C when probably two forms of $H^+$, one that is bound (0.7 ppm) and one that is more mobile (0.8 ppm), coexist. This means that there will be more mobile $H^+$ ions in SZYO20 at 125 °C compared to that at 95 °C. The more the free unbounded $H^+$ ions, the fewer hydrogen bonds. The hydrogen bonds mean both the hydrogen bonds to the neighbouring oxygen in a water molecular and a lattice oxygen from the perovskite oxide. On the other hand, the peak at 2.3 ppm assigned to signal from $OH^-$ and $H^+$ exchange partially splits to peaks at 3.8 ppm and 0.8 ppm at 125 °C as indicated in Supplementary Fig. 19. Considering the peak shift at higher temperatures and error bars for measurements (± 0.1 or so ppm), this is consistent with the signals at 3.5 ppm and 0.7 ppm for dry samples and signal at 2.3 ppm for partially hydrated sample measured at room temperature (Fig. 5a), indicating the conversion of SZYO20 sample from hydrated to dry level when temperature increases to over 100 °C, representing the conversion of $OH^-$ and $H^+$ moieties in full exchange to part exchange then to no exchange. This is consistent with the low conductivity of SZYO20 when the temperature is above 100 °C at ambient pressure, which is because in the absence of liquid water, it is hard to facilitate the hydrogen bond formation to diffuse $OH^-$ ions.

We have also performed $^{89}Y$ MAS NMR experiments on the partially hydrated sample as shown on Fig. 5e. The $^{89}Y$ direct polarisation (DP) signal measured with a spin echo experiment shows two broad

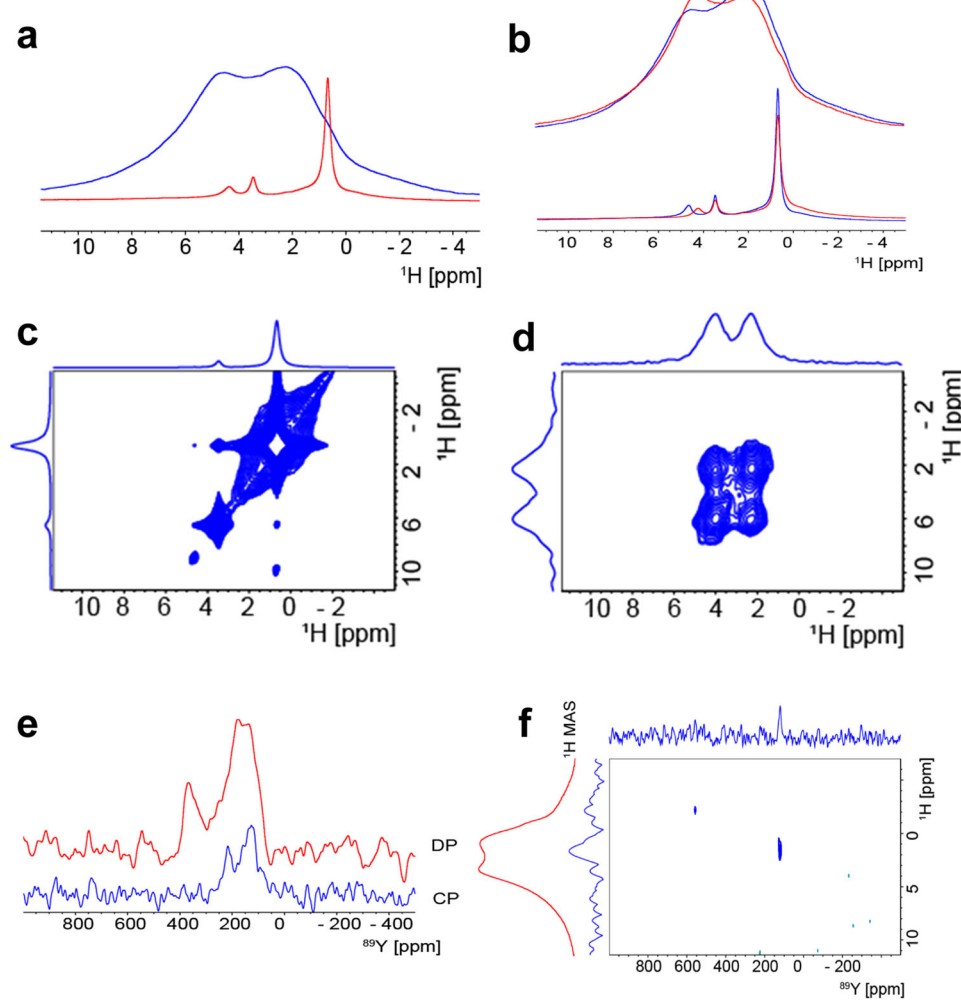

**Fig. 5 | Solid state NMR spectra of dry and partially hydrated SZYO20. a** The solid state $^1$H NMR spectra of dry (red) and partially hydrated (blue) SZYO20. **b** The solid state $^1$H NMR (MAS 60 kHz) spectra of partially hydrated (top) and dry (bottom) SZYO20 performed whit the rotors kept at -5 °C (blue) and +30 °C (red). **c**, **d** $^1$H-$^1$H NOESY correlation spectra (MAS 60 kHz) of dry (**c**) and partially hydrated

(**d**) SZYO20 mixing time (**c**) 0.1 s and (**d**) 1 s. **e** $^{89}$Y MAS at 8 kHz DP spectrum measured with a spin echo (red) and cross polarisation (blue) from $^1$H of partially hydrated SZYO20. **f** $^1$H-$^{89}$Y heteronuclear correlation experiment obtained with 6 ms cross polarisation, MAS 8 kHz.

signals at 370 and 160 ppm indicating Y sites are coordinated by 6, 7 and 8 oxygen atoms[24,47]. The $^1$H-$^{89}$Y cross polarisation (CP) spectrum shows two $^{89}$Y signals at 219 and 130 ppm (mainly 7 and 8 coordinated sites) in proximity to $^1$H nuclei. The $^1$H-$^{89}$Y heteronuclear correlation experiment, performed on the partially hydrated sample is displayed in Fig. 5f. It shows that the $^{89}$Y signals correlates with the $^1$H signal at 2.3 ppm which we assign to the OH$^-$ and H$^+$ exchange signal. This indicates the mobile OH$^-$ and H$^+$ ions are associated with Y$^{3+}$ sites since Y$^{3+}$ doping introduces oxygen vacancies in the SZO lattice, as shown in Eq. 1 and Eq.1. This is also consistent with the Raman results (Supplementary Fig. 20 and Supplementary Note 2). In water, the real composition of hydrated SrZr$_{1-x}$Y$_x$O$_{3-\delta}$ is hydrated or unhydrated oxyhydroxide, with a general formula, SrZr$_{1-x}$Y$_x$O$_{3-\delta-y}$(OH)$_{2y}$·$z$H$_2$O (= SrZr$_{1-x}$Y$_x$O$_{3-\delta}$·$n$H$_2$O, $n = y + z$), where the upper limit for $y$ is the oxygen deficiency $\delta$. With $y = \delta$ reflecting a hydrogen content consistent with complete filling of the oxygen vacancies. For $z = 0$, it is unhydrated oxyhydroxide while for $z > 0$, it is hydrated oxyhydroxide. The OH$^-$ ions are presented in the form of protonic defects, $OH_O^\bullet$, which are associated with oxygen vacancies. These proton defects may further form OH$^-$ species through charge exchange with neighbouring negatively charged species. The exchange and coupling of OH$^-$ and

H$^+$ ions in hydrated SZYO20 nearby Y$^{3+}$ ions has been confirmed by solid-state NMR observation which helps us for a better understanding of the conduction mechanism of SZYO20 in water.

To further confirm the existence of hydrogen and hydrogen-related species such as OH$^-$ ions in the perovskite oxide, the same solid-state $^1$H and $^{89}$Y NMR experiments were also carried in CZYO20 powders. As shown in Supplementary Fig. 21a, the DP $^{89}$Y MAS spectrum of water-hydrated CZYO20 is comparable with the $^{89}$Y spectra of partially hydrated SZYO20, but the signal at 200 ppm is narrower and does not extend over the 160 ppm region where the $^{89}$Y signal of the $^1$H -$^{89}$Y CP spectrum of SZYO20 shows higher coordination $^{89}$Y in proximity with $^1$H. For CZYO20, the $^1$H fast MAS spectra shown in Supplementary Fig. 21b and the $^1$H NOESY correlations shown in Supplementary Fig. 21c, d are comparable with the $^1$H spectra of SZYO20 with the similar composition shown in Fig. 5a−d, but the signal we assigned to OH$^-$ is much broader and distributed over a larger frequency range. Based on the chemical composition of hydrated oxides, we believe both H$^+$ and OH$^-$ ions co-exist in hydrated perovskite oxides. Under certain conditions, theoretically both ions can diffuse in the oxide lattice, forming a mixed OH$^-$/H$^+$ ionic conductor.

## TG and raman spectra analysis

Simultaneous Thermal Analysis (STA) was carried out on partially hydrated SZYO20. According to the Differential Scanning Calorimetry (DSC) and Thermogravimetry (TG) curves in Supplementary Fig. 22, the weight loss gradually increases with temperature. The weight loss detected below 100 °C is from physically absorbed water and gases, which is not used for the hydration analysis. Chemically bound water associated with the carbonate traces present on the sample surface normally contributes to the loss observed at about 300 °C[34], which is also the temperature at which OH⁻ in perovskite-like hydroxides usually departs[48,49]. The weight loss step observed between 800 and 850 °C in Supplementary Fig. 22a is assigned to the departure of protonic species associated with conductive properties which has been reported for other similar oxides[34,48]. According to the oxygen occupancy parameters obtained by the XRD Rietveld refinement for SZYO20 samples (Supplementary Table 1), the oxygen vacancies are mainly at the O2 sites and the total oxygen concentration in the perovskite *Pbnm* phase is 2.928 (~ 2.93), thus the investigated perovskite phase can be written as $SrZr_{0.86}Y_{0.14}O_{2.93}$. This is also consistent with the oxygen occupancy obtained in neutron diffraction analysis, implying ~2.4% oxygen vacancies ($\delta$ ~ 0.07) in the unhydrated SZYO20 (Supplementary Table 6). The Zr/Y atomic ratio of washed SZYO20 obtained by STEM-EDX (Supplementary Table 2) is close to and slightly lower than the Zr/Y atomic ratio of precise $SrZr_{0.86}Y_{0.14}O_{2.93}$ phase. Therefore, it is reasonable to speculate that except for a tiny amount of Y-rich residues, the perovskite phase $SrZr_{0.86}Y_{0.14}O_{2.93}$ is still stably retained in the washed samples, possibly forming the hydrated oxyhydroxide $SrZr_{0.86}Y_{0.14}O_{2.86}(OH)_{0.14} \cdot zH_2O$. Theoretically, the OH⁻ species in this oxyhydroxide would completely decompose before 850 °C as the weight of the sample became stable above this temperature. Therefore, the water content $n$ in hydrated oxyhydroxide $SrZr_{0.86}Y_{0.14}O_{2.93} \cdot nH_2O$ is estimated to be $n = 0.552$. Similarly, for water-hydrated CZYO20, the TG curve plateaus above 695 °C (Supplementary Fig. 22b) thus the water content $n$ in $CaZr_{0.8}Y_{0.2}O_{2.9} \cdot nH_2O$ is estimated to be $n = 0.065$. This is consistent with what was observed in the conductivity measurement (Fig. 4a), i.e., the ionic conductivity of CZYO20 is much lower than SZYO20 in water because the water content in hydrated CZYO20 is much smaller and is fully dehydrated at a lower temperature compared to hydrated SZYO20. This further demonstrates the importance of hydrated water in order to achieve high ionic conductivity.

The loss of hydrogen bonds and OH⁻ species in hydrated oxides at high temperatures is further confirmed by the in situ high temperature Raman spectra shown in Supplementary Figs. 23, 24, which were measured on heating from 22 °C to 1000 °C. Supplementary Fig. 23a shows the spectra associated with the vibrational dynamics of the SZYO20 oxide lattice. It has been reported that intercalation of water fills vacancies in the lattice, thus resulting in less peak splitting in the spectrum[34,50]. A slight broadening of bands is observed from 150 °C to 400 °C. After further thermal dehydration, the Raman features in 400–900 cm⁻¹ region slightly shift to a lower wavenumber, and a new peak is formed at 820 cm⁻¹ at temperatures above 800 °C, suggesting the removal of protons from oxygen vacancies[34] and a phase transition away from the hydrated structure[48]. The peaks above 3600 cm⁻¹ in Supplementary Figs. 23b, 24b are assigned to the hydroxyl (OH⁻) groups formed in the perovskite oxide upon hydration[50-52]. These peaks almost disappear at above 300 °C indicating the departure of OH⁻. The extra peak at 469 cm⁻¹ in Supplementary Fig. 24a for water-hydrated CZYO20 can also be assigned to a vibration related to oxygen vacancies filled with hydroxyl groups[52], which disappear at above 600 °C, consistent with the TG analysis.

## Neutron diffraction analysis

Due to a lack of sensitivity of X-ray scattering to protons and oxygen atoms in metal oxides, the proton positions and oxygen occupancy of the water-hydrated sample could not be precisely refined by XRD. In order to identify the position of OH⁻ and/or hydrogen in the structure and more reliably determine oxygen positions, neutron diffraction data was collected on deuterated samples. The washed SZYO20 powder was further hydrothermally deuterated at 225 °C for 12 h prior to neutron diffraction measurement (Supplementary Note 3). The deuterated sample was not found to contain secondary $SrY_2O_4$ phase or residual $Sr(OH)_2 \cdot 8H_2O$, while some $Y(OH)_3$ (6.9% phase fraction) was found to be present, which corroborates with the Y-rich area observed in the STEM analysis of the washed sample (Fig. 3a). The mixed Zr/Y site occupancies were not able to be reliably refined as they have very similar neutron scattering lengths (Zr 7.16 fm, Y 7.75 fm)[53]. $Y(OH)_3$ has very low solubility in water, therefore an overall Zr:Y ratio of 4:1 for the total deuterated sample was used to calculate the stoichiometry of the perovskite phase $SrZr_{1-x}Y_xO_{3-\delta}$. A value of $x = 0.14$ for the yttrium concentration was calculated which was used for the subsequent refinements. Refinement of the strontium positions for deuterated SZYO20 were not found to significantly differ from the anhydrous structure (Supplementary Fig. 25 and Supplementary Tables 6, 7). The strontium site occupancy was refined to a value very close to one and was therefore fixed to a value of one. Refinement of the oxygen occupancies were found to converge towards unity, consistent with uptake of OD into vacancies. Deuterium was found to stably refine to a position of (0.663(6), 0.158(6), 0.042(4)), with a corresponding decrease in the goodness of fit and the standard uncertainties of the refined parameters after the addition of D in the structure. The deuterium occupancy was refined to a value of 0.065(5), within error of the 0.07 occupancy expected for this site due to the complete filling of the oxygen vacancies with OD groups. Hence for the final refinement, the D occupancy was fixed to 0.07. The final refined structure has an approximate composition of $SrZr_{0.86}Y_{0.14}O_{2.86}(OD)_{0.14}$, which is consistent with the approximate amount of oxygen vacancies expected from X-ray and neutron diffraction of the dried sample. An illustration of the refined $SrZr_{0.86}Y_{0.14}O_{2.86}(OD)_{0.14}$ structure is shown in Supplementary Fig. 25b, c, and the structural parameters in Supplementary Table 7. The O-D lengths refined to a value of 1.03(4) Å, consistent with expected O-H(D) lengths and with other reported O-H(D) containing perovskites[54-56]. The presence of additional OH/OD bonds in the bulk perovskite confirmed by neutron diffraction explains that the considerable conductivity of the hydrated SZYO20 sample is indeed partly derived from the migration of protons and OH⁻ in the lattice.

## Ab initio molecular dynamics simulations

To get a better understanding on the atomistic mechanism of proton and OH⁻ diffusion in SZYO20, we performed DFT based ab initio molecular dynamics (AIMD) simulations at 400 K. We started from a 2 × 2 × 3 supercell of $SrZrO_3$, which has a total of 72 × Sr atoms, 72 × Zr atoms and 216 × O atoms. We then created 8 oxygen vacancies, which are charge compensated by substitution of 16 × $Zr^{4+}$ cations for 16 × $Y^{3+}$ cations. This gives an effective chemical formula of $SrZr_{0.78}Y_{0.22}O_{2.89}$. Finally, we introduce 4 × $H_2O$ molecules, which are split into 4 × OH⁻ and 4 × H⁺, with the 4 × OH⁻ each occupying an oxygen vacancy (after which only 4 oxygen vacancies remain in the cell), and the 4 × H⁺ each attaching to an $O^{2-}$ anion. This leaves a total of 8 × OH⁻ (or 8 × H⁺) in the cell, and the AIMD simulations were performed for this cell. Due to the significant computational cost of running AIMD simulations for such a large cell, we were only able to run the AIMD simulations for a duration of 20 ps (a total of 40,000 MD time steps). Nevertheless, we were able to observe several interesting events, which include one local OH⁻ diffusion, several OH⁻ rotations, and multiple proton hopping. To illustrate the proton hopping during our AIMD simulations, in Fig. 6a, we show the distance between a selected proton and its neighbouring O atom (O6, as in the first frame of the production MD run) as a function of MD simulation time. For comparison, we show the distances between all the 8 protons and their neighbouring O atoms in the

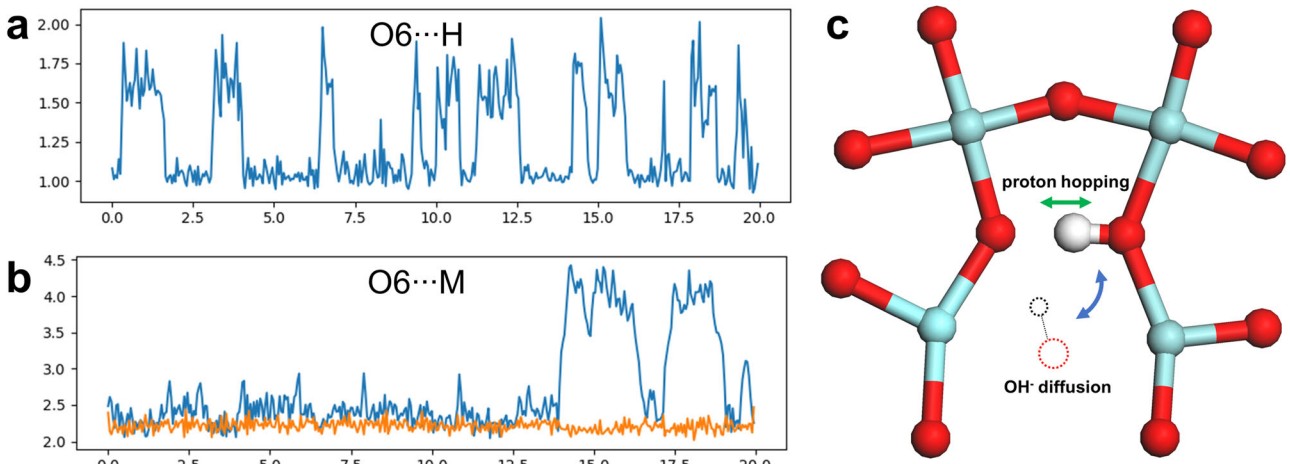

**Fig. 6 | Molecular dynamics simulation of SZYO20. a** Distance (in Å) between a selected proton and its neighbouring O atom as a function of MD simulation time (in ps). **b** Distance (in Å) between a selected oxygen atom (which was bonded to a proton initially) and its neighbouring B-site cations as a function of MD simulation time (in ps). The blue and orange lines represent the distances between an $O^{2-}$ anion and its two neighbouring B-site cations, respectively. **c** Schematic representation of possible atomistic mechanisms of proton hopping and $OH^-$ diffusion. The dotted circles indicate new positions of $OH^-$ after diffusion, with the previous site now becoming an oxygen vacancy. All other atoms are omitted for clarity. Colour code: red – oxygen, white – hydrogen, light green – zirconium or yttrium.

Supplementary Fig. 26. We find that apart from two protons (bonded to O1 and O7), most of the other protons hopped between neighbouring oxygen atoms, reflected by the changing $O \cdots H$ distances relative to the starting point of the MD simulation, and in a few cases, protons can hop back and forth, see e.g., the proton bonded to O6 (Fig. 6a). For the proton bonded to O4 initially, interestingly, it quickly hopped onto another nearby $O^{2-}$ anion, which was followed by two separate $OH^-$ rotations, and this proton did not come back to the original site within the 20 ps of AIMD simulation (reflected by the large $O \cdots H$ distance which is over 4 Å, see Supplementary Fig. 26). Similar $OH^-$ rotations were also observed at the site of $O2 \cdots H$.

To obtain further insight, we also looked into the distances between the 8 oxygen atoms that were bonded to protons initially, and their neighbouring B-site cations, which can be either $Zr^{4+}$ or $Y^{3+}$, and we show our results in Fig. 6b and Supplementary Fig. 27. As expected, majority of the O-metal distances show moderate oscillation behavior due to atomic vibrations at finite temperature (characterised by small and reversible changes in O-metal distances). However, the distance between O6 and one of the neighbouring B-site cations increased to ~4 Å at -15 ps and -18 ps (Fig. 6b), respectively, and we found this was related to $OH^-$ diffusion. More specifically, the $OH^-$ diffusion involves a $OH^-$ ion exchanging position with a neighbouring oxygen vacancy as shown in the schematic in Fig. 6c. As can be seen, this process was stabilised by strong hydrogen bonding with another $O^{2-}$ anion nearby, i.e. the strong hydrogen bonding is likely to reduce the transition barrier of this process in comparison with the diffusion of an individual $O^{2-}$ anion. In combination with the $OH^-$ rotations as observed in our AIMD simulations, this may help to explain the favourable $OH^-$ conductivity observed in our experiment at a relatively low temperature of 90 °C.

With respect to the comparatively low proton conductivity at this temperature, we believe this is largely due to the orthorhombic crystal structure of SZYO20 at 90 °C, which leads to close oxygen–oxygen separations between the vertices of adjacent octahedra (Fig. 6c and Supplementary Table 8), and therefore protons are trapped between $O \cdots O$ (forming a very strong hydrogen bond)[57]. At higher temperatures, SrZrO₃ goes through the *Pbnm* to *I4/mcm* phase transition, where the octahedral tilting becomes smaller, and the $O \cdots O$ distance between adjacent octahedra becomes larger (Supplementary Table 8). This means the kinetic barrier of the rate-limiting step for proton diffusion, i.e., proton hopping from one side of the octahedron to

another (involving $OH^-$ rotation), becomes smaller, and therefore the possibility that the proton gets trapped between $O \cdots O$ is much smaller, which may explain why proton conduction dominates at higher temperature in our experiment.

The $OH^-$ ion diffusion involves the migration of $OH^-$ ions to the neighbouring oxygen vacancies, the rotation of the $OH^-$ ions in the vacancies, and the formation of strong hydrogen bonding with another neighbouring oxygen (Fig. 6c). Therefore, oxides with large cell volume, such as doped BZO, will facilitate the rotating of $OH^-$ ions while those with small cell volume, such as doped CZO, will facility the diffusion of $OH^-$ ions to the neighbouring oxygen vacancies and the formation of strong hydrogen bonding due to the short distance. Between BZO and CZO, an intermediate lattice, such as SZO, facilitates the rotation of $OH^-$ ions, with a short distance for the diffusion of $OH^-$ ions to the neighbouring oxygen vacancies and the formation of strong hydrogen bonding. This could be the reason why both pure and doped SZO exhibit the highest ionic conductivity in both AZrO₃ and AZr₀.₈Y₀.₂O₃₋δ (A = Ca, Sr, Ba) series (Fig. 4a, Supplementary Fig. 13e).

### Fuel cell feasibility study

After the discovery of a new ionic conductor, it is important to demonstrate its feasibility or potential in real applications. To further confirm the ionic conduction of SZYO20, a primary H₂/air fuel cell and an NH₃/air fuel cell using thick SZYO20 pellet as the electrolyte were fabricated (Supplementary Fig. 28). Figure 7a shows the OCV of the H₂/air fuel cell at a temperature of 20 °C. The OCV gradually increased against time in the first three hours due to hydration then reached a stable value of 1.07 V. This is consistent with the conductivity measurements of unwashed SZYO20 in water at room temperature – although the initial conductivity is not high, over time the conductivity continues to increase until fully hydrated (Supplementary Fig. 6c). Therefore, pre-humidification is necessary in order to achieve high ionic conductivity and thus high OCV of a fuel cell. The observed OCV is fairly close to the theoretical value of a H₂/O₂ fuel cell (1.23 V at 25 °C) considering humidified air instead of pure O₂ was used at the cathode. From the observed OCV, it can be deduced that SZYO20 is a nearly pure ionic conductor at 20 °C in humidified atmosphere, which is consistent with the measured ionic transfer number (Fig. 4a). At 20 °C, a maximum current density of 1.25 mA cm⁻² with maximum power density of 0.34 mW cm⁻² was achieved respectively (Supplementary Fig. 29). The low power density is due to poor solid electrode/

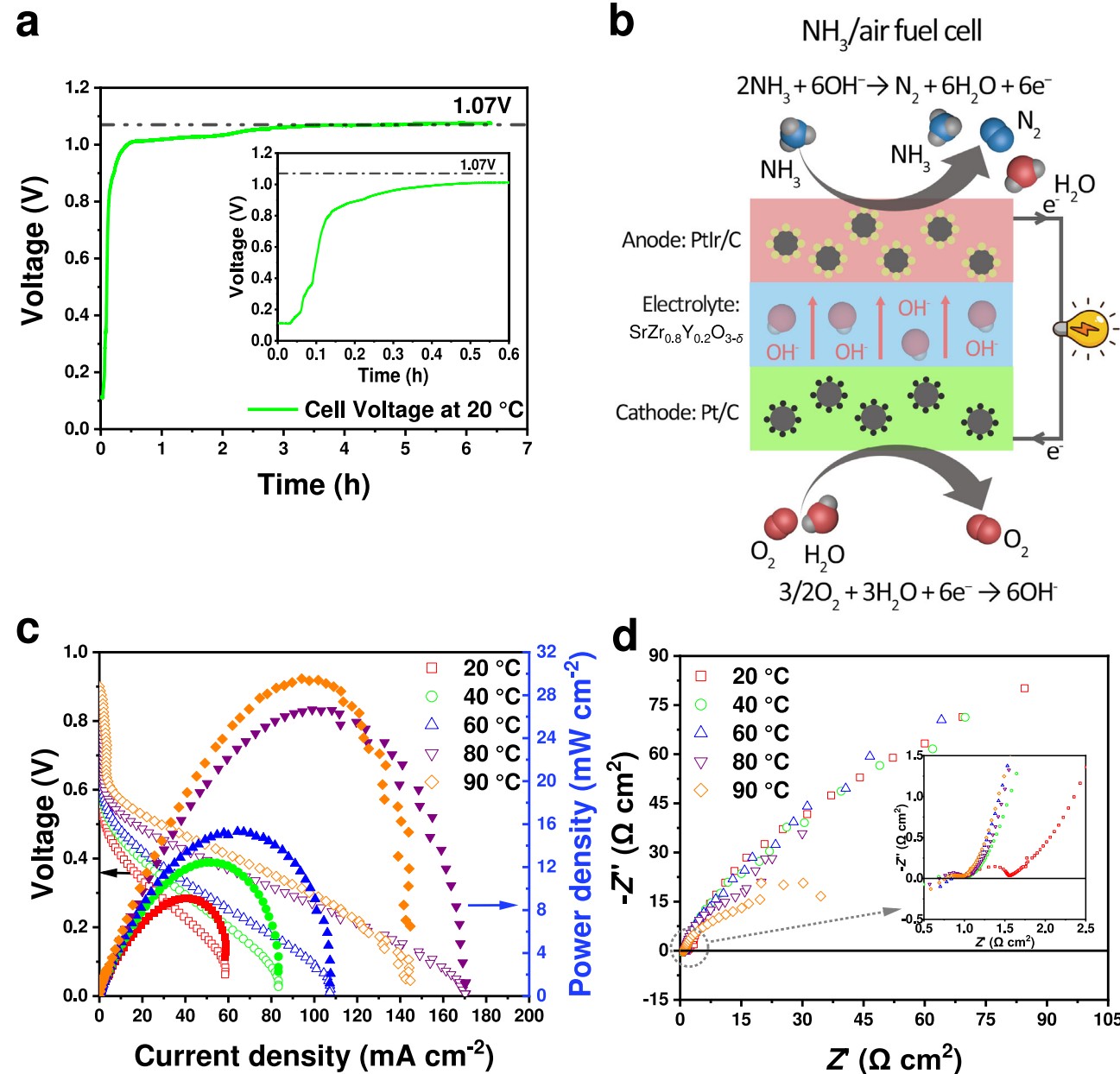

**Fig. 7 | Fuel cell measurements when SZYO20 was used as electrolyte. a** The OCV change against the time of a H₂/air fuel cell at 20 °C. **b** Working principle of low temperature SOFCs for an NH₃/air fuel cell. **c**, **d** The performance (**c**) and impedance (**d**) of an NH₃/air fuel cell (35 wt% NH₃H₂O solution + 3 M KOH as the fuel) at different temperatures, enlarged impedance spectra are displayed in the insert.

electrolyte interfaces and unoptimized electrode microstructure. Although the power density is not attractive, compared to the state-of-the-art hydrogen fuel cells based on an acidic Nafion membrane, this key experiment indicates it is possible to develop NAT-SOFCs for different applications. The main purpose of this experiment was to confirm the OH⁻/H⁺ ionic conduction of SZYO20 electrolyte through OCV and fuel cell measurement thus the performance was not optimised. However, ceramic oxide mixed OH⁻/H⁺ conductor SZYO20 can work in neutral (water) or alkaline conditions, therefore potentially low-cost non-precious metal catalysts, such as perovskite oxides used in high temperature SOFCs, can be used as efficient electrocatalysts in both anode and cathode, which can dramatically reduce the overall cost of the fuel cell based on ceramic oxide electrolytes. In our recent study, it has been demonstrated that perovskite oxide $LaCr_{0.25}Fe_{0.25}Co_{0.5}O_{3-\delta}$ exhibits comparable activity for oxygen reduction reaction (ORR) to Pt/C cathode, demonstrated by comparable performance in direct

ammonia fuel cells (DAFCs) with a polymeric alkaline membrane electrolyte[58]. Replacing the expensive acidic Nafion membrane in conventional PEMFCs with a mixed ceramic OH⁻/H⁺ conductor such as SZYO20 may significantly reduce the cost for both the electrolyte membrane and the electrode materials, and thus the cost of the whole fuel cell stack[10,59].

Ammonia fuel cells are important electrochemical devices to convert the carbon-free fuel ammonia to electricity at high efficiency[60–63]. Potentially ammonia fuel cells can be used to power electric vehicles, bypassing the problems of storage and transport of hydrogen[60,64]. The ceramic oxide ionic conducting electrolyte can be fully humidified if a liquid fuel such as ammonia solution is used at the anode. Therefore, the ceramic oxide SZYO20 is particularly suitable to be used as electrolyte for fuel cells using liquid fuels when the anode side is exposed to water, or electrolysers for splitting liquid water into hydrogen when both anode and cathode sides are exposed to water.

Here, ammonia was also tested as the fuel for this type of NAT-SOFC based on a thick SZYO20 electrolyte (Fig. 7b). When 35 wt% ammonia solution was used as the fuel, the current density of the fuel cell electrolyte was comparable to the fuel cell using commercial anion exchange membrane (AEM) (Supplementary Figs. 30, 31). The DAFC retained a relatively good durability for over 20 h when using pure ammonia solution as a fuel (Supplementary Fig. 30c) indicating good chemical compatibility of SZYO20 with $CO_2$ in air at the cathode. The oscillation of the measured voltage is due to the unstable cell temperature from the home-made testing jig using heating cartridges to provide the required temperature.

Considering the thickness difference between SZYO20 pellet (1300 μm) and AEM (50 μm), the conductivity of SZYO20 pellet, is estimated from the series resistance of the a.c. impedance spectra of the fuel cells, are higher than the commercial AEM (Supplementary Fig. 32). The observed low ionic conductivity of commercial AEM is due to its poor chemical compatibility with $CO_2$ in air, which was used as the oxidant at the cathode. It has been reported that adding KOH in ammonia solution can significantly improve the fuel cell performance when an alkaline membrane was used as the electrolyte[62,63]. Further tests were performed with a 35 wt% ammonia solution, with added 3 M KOH as the fuel for a DAFC based on SZYO20 electrolyte. The relevant I-V curves and a.c. impedance spectra of this DAFC are shown in Fig. 7c, d. At 90 °C, a maximum power density of 30 mW cm$^{-2}$ was achieved, which is comparable to the power density of an $NH_3$/air fuel cell at 80 °C, based on commercial polymeric alkaline membrane electrolyte and $CO_2$-free air[58]. It should be noted that the ammonia fuel cell in this study was measured under ambient pressure with compressed air with $CO_2$ while the ammonia fuel cells in the previous study were carried out under an applied back-pressure[58]. The mechanical strength of the SZYO20 pellet is not high, thus the ammonia fuel cell measurement was not carried out under pressure. The mechanical strength of oxide materials can be tailored/improved by doping strategy. Considering the thickness difference of the electrolyte, 1300 μm of the oxide electrolyte SZYO20 vs. 50 μm of the commercial polymeric AEM electrolyte, there is a huge potential to further improve the power density of the DAFCs when thin-film technology used in conventional SOFCs or solid-state lithium-ion batteries, such as tape-casting, laser deposition, are used in the preparation of the NAT-SOFCs[40]. In conclusion, good ionic conduction of SZYO20 at NAT has been demonstrated in both hydrogen and ammonia fuel cells. The aim of the fuel cell study is to demonstrate the feasibility in real applications while the power density can be greatly improved through the reduction of the thickness of the electrolyte, improvement in the solid electrolyte/electrode interfaces, optimisation of the microstructure of the electrode to achieve a level for real applications. The high ionic conductivity of the ceramic oxide electrolyte at a temperature below 100 °C and the excellent chemical stability and compatibility with $CO_2$ in air, have overcome the key problems associated to conventional AFCs and AMFCs. Similar to conventional PEMFCs based on acidic Nafion membrane, humidifying the fuels and air is necessary in order to maintain high ionic conductivity of the electrolyte unless liquid fuels such as ammonia/methanol/ethanol solution is used at the anode or/and liquid oxidant such as $H_2O_2$ solution is used at the cathode[65].

## Conductivity measurements using fuel cell testing jig

As shown in Supplementary Table 4, some of the perovskite may partially dissolve in water over time, leading to increased liquid water conductivity. The liquid conductivity due to additional dissolved species could result in a slightly higher measured conductivity compared to the real conductivity when using the testing jig shown in Supplementary Fig. 5. To avoid the influence of the ionic conductivity of surrounding water or solution, the fuel cell testing jig shown in Supplementary Fig. 33 was used to measure the conductivity of SZYO20 and CZYO20 pellets. When the fuel cell testing jig was used, it

is inevitable that some ions will transfer outside the silver electrodes (1 cm$^2$), contributing to the measured conductivity, thus it is also difficult to obtain a completely accurate conductivity value from our fuel cell setup. Therefore, we calculate the conductivity in two areas, the area of the whole pellet (dot lines in Supplementary Figs. 34, 35) and the area covered by silver electrode. The real conductivity should be between these two values. For SZYO20, the conductivity measured by the conventional conductivity testing jig (Supplementary Fig. 5) is between the two values calculated using the different pellet areas (Supplementary Fig. 34a, b), which supports the conductivity as measured by the original jig as reliable. When comparing the conductivity of SZYO20 sintered at 1300 °C and 1500 °C respectively, in water, the conductivity of the 1500 °C sintered pellet is slightly lower than that sintered at 1300 °C (Supplementary Fig. 34c), which implies that the conductivity is not only related to the bulk, but also to the surface of the pellet. The relative density of SZYO20 is 94% when sintered at 1500 °C, which is higher than 80% when sintered at 1300 °C (Supplementary Table 9 and Supplementary Note 4). Ions particularly protons may diffuse along the grain boundary together with water molecules via a vehicle mechanism, similar to that in Nafion membrane, causing high H$^+$ ionic conduction. Additionally, surface protonic conductivity in wet atmospheres has been reported on porous oxides including monoclinic $ZrO_2$[66] and porous nanoscopic $CeO_2$[67], which may partially explain the ionic conductivity of some relatively low relative-density materials investigated in our work. A similar phenomenon was also observed for 1300 °C and 1500 °C sintered CZYO20 pellets while the sample sintered at higher temperatures with higher density exhibited relatively lower conductivity (Supplementary Fig. 35a). For dense CZYO20 pellet sintered at 1500 °C, the measured conductivity in water using the fuel cell testing jig calculated from the electrode coverage area (1 cm$^2$) is comparable to that measured in the conventional conductivity measurement jig (Supplementary Fig. 35b).

## Conductivity of dense CZYO20 pellets in KOH solutions

The key advantage of the conductivity measurement using the fuel cell type testing jig is, it can remove the effects of ionic conductivity of surrounding solution such as water or any other solutions. In our study of DAFCs, it was found that the normalised series resistance of the fuel cell is only 1.1 Ω cm$^2$ when 35 wt% $NH_3H_2O$ solution + 3 M KOH was used as the fuel (Fig. 7d) when the operating temperature is above 40 °C. This corresponds to an ionic conductivity of 0.11 S cm$^{-1}$ considering a thick pellet electrolyte was used. It is likely that the added KOH will increase the ionic conductivity of the SZYO20 pellet. Considering the relative density of the 1300 °C SZYO20 pellet used in the DAFC measurements was only 80%, KOH may also diffuse into the holes of the pellet, leading to increased OH$^-$ ionic conductivity, similar to the case of alkaline membrane fuel cells based on polymeric alkaline membranes. To eliminate the effects of holes, we tested the ionic conductivity of dense CZYO20 pellets sintered at 1500 °C with a relative density of 99% (Supplementary Table 9) in water, 1 M, 3 M and 6 M KOH solutions, using the fuel cell testing jig. As shown in Supplementary Fig. 35c, the ionic conductivity continued to increase with increased KOH concentration. At 90 °C, when the dense CZYO20 is exposed in 6 M KOH solution, its conductivity is approximately 0.1 S cm$^{-1}$, comparable to that of Nafion membrane. This is also 100 times higher than the ionic conductivity of CZYO20 in pure water at 90 °C (Supplementary Fig. 35b). At 25 °C, the conductivity is still as high as $4.8 \times 10^{-2}$ S cm$^{-1}$, which is high enough to be used as an electrolyte for fuel cells and electrolysers. We also tested the ionic conductivity of dense CZO pellets sintered at 1500 °C (dash lines in Supplementary Fig. 35d), which was one-fifth of that of dense CZYO20 in 1 M KOH and half of that in 6 M KOH. This further confirms that oxygen vacancies in the lattice produced from doping still play a critical role in high ionic conductivity of dense pellets even when the concentration of charge carriers, OH$^-$, is quite high in the surrounding

environment. These dense oxide materials can be used as the electrolyte in alkaline or alkaline membrane electrolysers while both sides of the separating membrane are soaked in concentrated aqueous alkaline solution. This kind of electrolyser based on solid oxide electrolyte such as CZYO20 can be called near ambient solid oxide electrolytic cells (NAT-SOECs).

### Electrolysis of $H_2^{18}O$ and $D_2O$ using dense CZYO20 as the electrolyte

In order to have a better understanding of the key role of $OH^-$ ionic conduction of the perovskite oxides, we carried out electrolysis experiments in 20% $H_2^{18}O$, pure $D_2O$, and 1 M KOH dissolved in $D_2O$ when dense CZYO20 was used as the electrolyte. Following the electrolysis experiments, Raman spectroscopy was used to detect the $^{18}O$ labelled water, while Fourier-transform infrared spectroscopy (FTIR) was used to detect the $H_2O$, HDO and $D_2O$ species in solutions (Supplementary Figs. 36–38 and Supplementary Note 5).

One side of the electrolytic cell, for hydrogen evolution reaction (HER), was circulated with 20% $H_2^{18}O$ in $H_2^{16}O$, the other side, for oxygen evolution reaction (OER), was circulated with pure $H_2^{16}O$. It has been reported that an extra band at 684 $cm^{-1}$ was observed for $^{18}O$-labelled NaOCl in Raman spectroscopy, along with the O-Cl stretching band of NaOCl observed at 711 $cm^{-1}$ [68]. Therefore, the NaOCl solution was used as an additive to detect the presence of $H_2^{18}O$ and $^{18}OH^-$ species during electrolysis. The transfer of oxygen ($^{18}O$) from the HER side to the OER side after applying 2 V constant voltage through the electrolytic cell for a certain time was confirmed by the observed 683 $cm^{-1}$ Raman spectra in Supplementary Fig 36c–e, while no $^{18}O$ labelled species were detected on the OER side when there was no potential applied (green line in Supplementary Fig. 36e). This indicates the $H_2^{18}O$ did not spontaneously diffuse through the dense CZYO20 pellet without any potential applied, and the conduction of $^{18}O$ labelling species (e.g., $^{18}OH^-$) through the pellet has been verified.

Similarly, pure $D_2O$ was circulated in the HER side of the electrolyser for $D_2$ evolution reaction. When there is no potential applied for 43 h, no $D_2O$ was detected on the other side, indicating that $D_2O$ itself cannot automatically diffuse through the dense CZYO20 pellet, as expected (blue line in Supplementary Fig. 37b). However, after applying 2 V constant voltage through the cell for 100 h, the HDO peaks at absorbance bands of 1450 $cm^{-1}$ and 2504 $cm^{-1}$ [70] in FTIR spectra were observed in pure $H_2O$ on other side, indicating the material conducting $OH^-$ or $OD^-$ ions. Along with the confirmation of oxygen exchange in the electrolysis of $H_2^{18}O$, the $OH^-$ ionic conduction of dense CZYO20 is further demonstrated by the $D_2O$ electrolysis experiments. It was found the HDO signal became stronger with extended duration of electrolysis (Supplementary Fig. 37c). Weak $D_2O$ signal, the absorbance band at 1200 $cm^{-1}$ [69], was found when the duration of applied 2 V was over 35 h, indicating $D_2O$ can diffuse across the dense pellet under certain applied DC voltage for a long time. The diffusion of smaller $H_2O$ is expected to be easier thus dense CZYO20 can be potentially used for water separation when a DC voltage (or current) is applied through the cell. This has potential applications in seawater desalination or wastewater treatment.

In the experiments of electrolysis of both $H_2^{18}O$ and $D_2O$ respectively, under applied DC voltage, both $^{18}O$ and D can transport through the dense CZYO20 pellets in water, which indicates most likely the labelled $^{18}O$ and D isotopes are transferred through the $OH^-$ ions.

To further investigate the $OH^-$ ionic conduction, 1 M KOH dissolved in $D_2O$ was used in the HER catalytic side of the electrolytic cell. In order to investigate the crossover of KOH through the dense electrolyte pellet, $CO_2$ gas was purged into the solution of OER side after electrolysis. If there are KOH species in the solution, the 1370 $cm^{-1}$ peak assigned to $K_2CO_3$ should be observed in FTIR spectra when $CO_2$ is present[70]. From the blue line in Supplementary Fig. 38c, no HDO/$D_2O$ peaks were observed when one side of the cell was exposed to 1 M

$KOH \cdot D_2O$ solution indicating no $D_2O$ can diffuse through the dense CZYO20 pellet, the same as the case for pure $D_2O$. When 2 V DC voltage was applied through the cell for 16 h, the signal of HDO was observed in pure water on the other side. However, no $K_2CO_3$ and thus no KOH was observed, indicating $OD^-$ ions diffused from the $KOH \cdot D_2O$ side to the $H_2O$ side while KOH molecules did not (Supplementary Fig. 38d). This experiment indicates the $OD^-$ ions were transferred through the oxide materials. When 1 M KOH was used, the $OH^-$ concentration was much higher than that in pure $H_2O$ which relies on the self-dissociation of $H_2O$,

$$2H_2O = H_3O^+ + OH^- \tag{3}$$

The concentration of $OH^-$ ions from self-dissociated $H_2O$ (Eq. 3) is very low, about $10^{-7}$ M at 25 °C. When the dense CZYO20 was exposed to 1 M $KOH \cdot D_2O$ solution, the concentration of $OH^-$ ions from KOH was about 1 M, about 7 orders of magnitude higher than that of pure $H_2O$. The $OH^-$ ions from KOH therefore act as 'feeding ions', i.e., when in contact with the nearest oxygen vacancies, these ions can diffuse through the lattice, leading to very high $OH^-$ ionic conduction. This has the effect of suddenly increasing the concentration of mobile ions. This also explains why the conductivity of CZYO20 in 6 M KOH is 100 times higher than that in pure $H_2O$. As the pellet is very dense, it is reasonable to speculate that the high $OH^-$ ionic conductivity is also related to the bulk material, i.e., $OH^-$ ions are also transferred through the oxide lattice.

The SEM images of the cross-section of the dense CZYO20 pellet before and after the conductivity in water and KOH solution are shown in Supplementary Figs. 39, 40 respectively. Element mapping indicates there is a weak K signal along the cross-section after the conductivity measurement. This suggests that $K^+$ ion can also diffuse into the grain boundary of the CZYO20 pellet. No $K_2CO_3$ was observed on the pure $H_2O$ side in the electrolysis of 1 M $KOH \cdot D_2O$ solution, because the direction of applied voltage only allows negatively charged $OH^-$ or $OD^-$ ions to diffuse to the waterside while the positively charged $K^+$ ions should diffuse to the opposite direction, the $D_2O$ side. The presence of KOH in the cross-section of dense CZYO20 pellets after the conductivity measurements in KOH, indicates KOH can also diffuse into the grain boundary of the oxide, which may form another pathway for $OH^-$ ions, leading to the high ionic conductivity.

From the analyses above, $OH^-$ ions may transfer through both the oxide lattice and the grain boundary where KOH has been diffused. Both may make contributions to the high $OH^-$ ionic conductivity of CZYO20 in concentrated KOH solution. However, the ionic conductivity of doped CZYO20 with extrinsic oxygen vacancies is much higher than that of pure CZO with limited intrinsic oxygen vacancies in both 1 M and 6 M KOH aqueous indicating the oxygen vacancies in the oxide lattice play a crucial role for the higher ionic conductivity in perovskite oxides (Supplementary Fig. 35d).

From solid-state NMR measurements, the hydrogen or hydrogen-related species such as $OH^-$ ions in CZYO20 is also associated with $Y^{3+}$ ions, where the oxygen vacancies are in the vicinity of $Y^{3+}$ ions, thus it is believed the $H^+$ and/or $OH^-$ ions are associated to oxygen vacancies (Supplementary Fig. 21). This is the same as that for SZYO20. It is therefore reasonably deduced that both SZYO20 and CZYO20 share the same ionic conduction mechanism.

## Discussion

In summary, the mixed $OH^-/H^+$ conduction of perovskite oxides such as SZYO20 and CZYO20 with high ionic conductivity in water and humidified air was discovered. Concentration cell measurements indicate about 70% of the charge carriers in fully-hydrated SZYO20 at room temperature are $OH^-$ ions which means it is mainly an $OH^-$ ionic conductor. This is very different from the known proton conduction in doped zirconates at high temperatures, typically above 500 °C. XRD

and ADF-STEM have confirmed the formation and excellent stability of the perovskite phase SZYO20. Neutron diffraction and XRD refinement show that there are oxygen vacancies in unhydrated SZYO20 (with $\delta$ ~ 0.07). This ceramic oxide OH$^-$/H$^+$ ionic conductor exhibits excellent chemical stability in water/steam and CO$_2$ in air, overcoming key problems of the electrolytes in conventional AFCs and AMFCs. Equivalent electrochemical circuit simulation on EIS has extracted the bulk and grain boundary conductivity of samples, which suggests, that for SZYO20, the conductive ions not only diffuse in the perovskite lattice but may also transport through the grain boundaries. After being fully hydrated, the presence of hydrogen and additional oxygen atoms in the SZYO20 bulk has been verified by neutron diffraction analysis. A combination of intra-proton transfer and OH$^-$ migration within the lattice could be taking place resulting in the observed high conductivity.

Solid state NMR study reveals that the exchange between OH$^-$ and H$^+$ ions in partially hydrated SZYO20 and, transfer of OH$^-$ and H$^+$ ions is coupled with dopant Y$^{3+}$ ions at the B-sites indicating oxygen vacancies play an important role for the ionic conduction. This is consistent with the DFT calculations, which showed that the migration of OH$^-$ ions in SZYO20 relies on the diffusion of OH$^-$ ions to the neighboring oxygen vacancies, which then rotate at a right angle to form a strong hydrogen bond with the closest oxygen. The high ionic conductivity of these perovskite oxides requires the presence of liquid water to facilitate hydrogen bonding and the corresponding OH$^-$ ion migration. In situ high-temperature NMR and Raman spectra have further confirmed the loss of hydrogen bonds and mobile OH$^-$ in hydrated oxide at high temperatures, consistent with the measured conductivity data against temperature. Thick SZYO20 pellets (1.8 mm and 1.3 mm for H$_2$/air and NH$_3$/air fuel cells respectively) were used to make fuel cells and the feasibility of the ceramic mixed OH$^-$/H$^+$ ionic conductor in fuel cell application has been demonstrated.

It was found denser SZYO20 and CZYO20 pellets exhibit relatively lower ionic conductivity in pure water. This indicates the ionic conduction is not only the bulk but could also be the surface conduction through the grain boundary. Protons might also diffuse through the grain boundary via a vehicle mechanism, similar to H$_3$O$^+$ ions in polymeric Nafion membrane.

Slowly dissolution of perovskite oxides in water has been observed through conductivity and pH measurements, while the contribution of the dissolved species from the perovskite to the measured high ionic conductivity of the oxides is negligible. To eliminate the effects of dissolved species from the perovskite in water on the measured conductivity, a fuel cell testing jig was used in the conductivity measurement of both SZYO20 and CZYO20. The measured conductivity from a conventional conductivity measuring jig and the fuel cell testing jig is comparable. The conductivity of dense CZYO20 and CZO pellets in concentrated KOH aqueous solution was also measured by the fuel cell testing jig. It was found that the ionic conductivity of dense CZYO20 in 6 M KOH is approximately 0.1 S cm$^{-1}$ at 90 °C, which is 100 times that in pure water. This shows that when a ceramic material is submerged in a concentrated KOH solution, the ionic conductivity can be significantly increased. Compared to CZO, much higher ionic conductivity was observed in doped CZYO20 indicating oxygen vacancies in the oxide lattice greatly facilitate the transfer of ions, such as OH$^-$ ions. Through the experiments of electrolysis of H$_2^{18}$O, D$_2$O and 1 M KOH·D$_2$O, as well as the EDX element analysis of dense CZYO20 after conductivity measurement in KOH, the possible pathways of OH$^-$ ions could be both the oxide lattice and KOH diffused into the grain boundary of oxide pellets. The phenomenon of ionic conductivity of CZYO20 significantly enhanced in KOH solution suggests the possibility that trace amounts of residual Sr(OH)$_2$ and/or Y(OH)$_3$ from hydrolysis of SrY$_2$O$_4$ in a sample with nominal composition of SZYO20 may similarly provide some OH$^-$ feed ions or O-H

transport channels, facilitating the OH$^-$ ion transfer resulting in high ionic conductivity.

Perovskite oxides have been reported as excellent electrochemical catalysts for both high-temperature solid oxide fuel cells and low-temperature electrochemical devices such as fuel cells and electrolysers[2,58,59,71,72]. If some multi-valence transition elements are introduced into the B-sites of SZYO20, mixed ionic-electronic conductors may be formed which could be potential electrocatalysts for electrochemical devices.

The discovery of low-temperature mixed OH$^-$/H$^+$ ionic conduction in oxide materials such as SZYO20 opens a window to discovering new low-temperature OH$^-$/H$^+$ ionic conducting materials in oxides or other ceramic materials. Adding KOH to the fuel of DAFCs significantly increased the OH$^-$ ionic conductivity, which has been further demonstrated with conductivity measurements in concentrated KOH and electrolysis of H$_2^{18}$O and D$_2$O using a dense CZYO20 pellet as the electrolyte. The significantly increased ionic conductivity of both CZYO20 and CZO in KOH solutions will provide a new route to develop new ionic conducting materials for electrochemical devices, such as electrolysers, fuel cells and batteries. A similar phenomenon has also been observed in other oxides such as doped cerates in our lab, while the investigation remains ongoing. For practical application of these oxide ionic conducting materials in electrochemical devices, the matched electrode materials and the electrode/electrolyte interface, etc. must show good performance and be compatible which requires further investigation.

## Methods

### Synthesis of AZr$_{1-x}$Y$_x$O$_{3-\delta}$ (A = Ca, Sr, Ba, $x$ = 0, 0.1, 0.2)

The perovskite oxides SrZr$_{0.8}$Y$_{0.2}$O$_{3-\delta}$ (SZYO20) were synthesized by a combustion method. 10.80 g of Sr(NO$_3$)$_2$ (98%, Alfa Aesar), 3.83 g of Y(NO$_3$)$_3$ · 6H$_2$O (99.9%, Alfa Aesar), 13.15 g of ZrOCl$_2$ · 8H$_2$O (98%, Alfa Aesar) and 5 mL of nitric acid (70%, Sigma Aldrich) were directly dissolved in deionized water to prepare a mixed solution. Then 38.81 g of citric acid (99 + %, Alfa Aesar) was added into the solution and magnetically stirred at 90 °C for 12 h on a hot plate to form a gel. Then the gel was dried at a constant temperature of 400 °C for 1 h to be ignited for combustion. After the organic components in the mixture burned off, the powder was ground in an agate mortar and calcined in air at 400 °C for 3 h, then 1000 °C for 2 h. After this, the as-prepared powder was reground and pressed into pellets with a diameter of 13 mm and 20 mm respectively under a pressure of 6 tons, and then sintered in air at 1300 °C for 24 h with a heating/cooling rate of 5 °C min$^{-1}$ to form SZYO20 phase[73]. Pellets were also sintered at 1500 °C to get higher relative density.

The perovskite oxides SrZrO$_3$ (SZO) and SrZr$_{0.9}$Y$_{0.1}$O$_{3-\delta}$ (SZYO10) were synthesized with the same combustion method. Sr(NO$_3$)$_2$, Y(NO$_3$)$_3$ · 6H$_2$O and ZrOCl$_2$ · 8H$_2$O according to the corresponding stoichiometric molar ratio with a small amount of nitric acid were used in precursor solution. The molar ratio of citric acid to total metal ions was 2:1[74]. The target perovskite phase was obtained after pelletized and fired at 1300 °C, the same as for preparation of SZYO20.

The perovskite oxides CaZrO$_3$ (CZO), CaZr$_{0.8}$Y$_{0.2}$O$_{3-\delta}$ (CZYO20), BaZrO$_3$ (BZO) and BaZr$_{0.8}$Y$_{0.2}$O$_{3-\delta}$ (BZYO20) were synthesized by the same combustion process. 16.70 g of Ca(NO$_3$)$_2$ · 4H$_2$O (99%, Sigma Aldrich) or 13.20 g of Ba(NO$_3$)$_2$ (99%, Sigma Aldrich) was used as precursors respectively to synthesize AZrO$_3$ and AZr$_{0.8}$Y$_{0.2}$O$_{3-\delta}$ (A= Ca, Ba). The usage of other precursors and igniting conditions were the same as for SZO and SZYO20. The CZO and CZYO20 pellets were sintered at 1300 °C or 1500 °C while the BZO and BZYO20 pellets were sintered at 1500 °C.

The relative densities of the pellets are detailed in Supplementary Note 4 and Supplementary Table 9.

The as-prepared pellets with a diameter of about 13 mm were used for conductivity measurements while those with a diameter of about

20 mm were used for concentration cell and fuel cell jig measurements.

## Synthesis of $SrY_2O_4$ and $Y(OH)_3$

In order to identify the effect of the secondary phase $SrY_2O_4$ in SZYO20 on the ionic conductivity, the single phase $SrY_2O_4$ was synthesised by the same combustion method. 6.48 g of $Sr(NO_3)_2$, 23.00 g of $Y(NO_3)_3 \cdot 6H_2O$ and 34.93 g of citric acid were directly dissolved in deionized water and the mixed solution was magnetically stirred at 80 °C for 12 h on a hot plate to form a gel. The igniting conditions were the same as for SZYO20. The target $SrY_2O_4$ sample was obtained after pelletized and fired in air at 1300 °C for 24 h.

In order to rule out the contribution of the hydrolysis products of $SrY_2O_4$ on the conductivity measurement, $Y(OH)_3$ sample was chemically deposited through the reaction between 0.5 mol $L^{-1}$ $Y(NO_3)_3 \cdot 6H_2O$ and 1.5 mol $L^{-1}$ of NaOH (98%, Alfa Aesar). The synthesized powder was thoroughly washed and dried at 60 °C for 8 h, and then pelletized for conductivity measurement. The commercial $Y_2O_3$ (99.9%, Alfa Aesar) powder was also pelletized and fired at 1300 °C for 4 h to measure its conductivity in water. (Supplementary Fig. 7)

## Structural characterizations

The X-ray diffraction (XRD) was carried out on a third-generation Malvern Panalytical Empyrean equipped with multicore (iCore/dCore) optics and a Pixel3D detector operating in 1D scanning mode with a Cu $K\alpha$ radiation (1.5419 Å) to identify the crystalline phases present in the samples. The diffraction scans were collected over a $2\theta$ range from 5° to 100° at a step size of 0.013° with a counting time of 110 s per step and were analyzed using the Malvern Panalytical Highscore Plus 4.9 software and the latest ICDD PDF-4+ database. The $2\theta$ scan range was widened to 5–140° along the diffraction scanning time extended to 4 or 15 h to obtain high-quality XRD data for refinement. Rietveld refinement of the representative perovskite oxides was carried out by GSAS-II[75]. The crystallographic refinement parameters are listed in Supplementary Table 1.

Scanning electron microscopy (SEM) observation of the microstructure was carried out on a Zeiss SUPRA 55-VP scanning microscope. Energy dispersive X-ray spectroscopy (EDX) was used to analyze the cross-section of pellets and determine the element composition of the samples through elemental mapping analyses.

Annular dark field (ADF) and bright field (BF) scanning transmission electron microscopy (STEM) imaging and EDX elemental mapping were carried out on a double aberration-corrected JEOL ARM200F TEM, operated at 200 kV, equipped with a 100 mm² Oxford Instruments windowless EDX detector. The SZYO20 powders for TEM measurement were ground from sintered pellets. Some pellets were washed in water at 90 °C three times to get rid of the second phase $SrY_2O_4$ and hydrated products and then ground into powders, labelled as washed SZYO20 samples. It is noted that Cu, Cr and C signals are artefacts generated during the STEM-EDX acquisition.

Raman spectra at room temperature were recorded on a Renishaw inVia Reflex Raman Microscope equipped with DPSS laser at 532 nm (10% power nominally 2 mW) and Renishaw CCD detector. Objective of X50 LWD and an acquisition time of 10 s was used during testing. For these measurements, the pellets were cleaved, and the fracture surface was analysed. The sample washed in water at 90 °C and dried in an oven for overnight was labeled as partially hydrated sample, while the sample dried in air for a moment before Raman measurement was labelled as hydrated sample. High-temperature Raman spectra of hydrated sample powders were recorded from room temperature to 1000 °C. Measurements were performed during the heating process, using the same instrumentation and conditions. The samples were mounted in a temperature-controlled heating stage, which was placed under the objective lens of the microscope for the high-temperature measurements. It adopted an average heating rate

of 20 °C min⁻¹. Raman spectra of $^{16}O$ and $^{18}O$ labelled liquid samples were obtained at 514.5 nm laser (100% power nominally 26 mW) with an 1800 l/mm grating and an acquisition time of 20 s × 10 accumulation. Liquid samples were put in a Helma Quartz Cuvette and mounted in a Cuvette holder accessory for measurement.

Simultaneous Thermal Analysis (STA) was carried out by a TA Instruments SDT 650 operating under an air atmosphere at 50 ml min⁻¹. The sample was loaded into a pre-tared 90 μl alumina pan. The analysis was performed from 25 to 1000 °C at a constant heating rate of 10 °C min⁻¹.

Fourier-transform infrared spectra (FTIR) of $D_2O$ and $H_2O$ liquid droplets were acquired using a Jasco FTIR-4200 (Type A) spectrometer equipped with a MIRacle Single Reflection ATR accessory with Diamond/ZnSe crystal (Pike Technologies). Measurements were collected between 4000 and 400 cm⁻¹ at a resolution of 2 cm⁻¹ with 100 scans per sample. Standard High-intensity ceramic source and a triglycine sulphate (TGS) detector were used. In order to remove atmospheric $CO_2$ and $H_2O$ peaks from sample measurements, background spectra and sample spectra were collected while continuous purging the instrument with $N_2$.

## Solid-State NMR spectra measurements

All SZYO20 powder samples for solid state nuclear magnetic resonance (NMR) measurements were heated in hot water at 90 °C for 20 h and washed to get rid of second phase $SrY_2O_4$ and the hydrated products. This process was repeated three times for each sample. The as-treated powder was dried in a fume cupboard at room temperature overnight, labelled as partially hydrated sample. Some of the as-treated powder was dried in a vacuum oven at 120 °C overnight to get rid of the hydrated water. This sample was labelled as dry SZYO20 sample.

CZYO20 powder samples for solid state NMR measurements were pre-treated through the same process as SZYO20 sample, labelled as water-hydrated CZYO20 and dry CZYO20 respectively. The sample after conductivity measurement in KOH solution was labelled as KOH-hydrated CZYO20.

NMR were performed on a Bruker Avance Neo spectrometer with a Larmor frequency of 850.2 MHz and 41.6 MHz for $^1H$ and $^{89}Y$, respectively using a 1.3 mm Bruker triple resonance HXY probe spinning at 60 kHz and a 4 mm Bruker double resonance HX low gamma probe spinning at 8 kHz. The $^{89}Y$ spectra were referenced to solid $Y(NO_3)_3 \cdot 6H_2O$ with the $^{89}Y$ peak set to -53.2 ppm. The $^1H$ NMR spectra were referenced to $^1H$ peak of solid adamantane set to 1.8 ppm. $^{89}Y$ MAS spectra were acquired with a spin echo pulse sequence with a 90 and 180 pulses set to 6 μs and 12 μs, respectively and with cross polarisation experiment with 6 ms contact time. $^1H$ MAS (60 kHz) NMR spectra were measured using a background suppression pulse sequence consisting of a 180 pulse followed by two 90 pulses. $^1H$ MAS spectra at 25, 95 and 125 °C on partially hydrated SZYO20 were recorded with a 1.6 mm HXY Phoenix NMR probe spinning at 30 kHz.

## Neutron diffraction measurements

SZYO20 powders dried in a vacuum oven at 180 °C for 40 h were labelled as vacuum-dried SZYO20. Some of the powders were pre-washed in hot water at 90 °C for 20 h for three times to get rid of the second phase $SrY_2O_4$ and the hydrated products. Then the washed powders were heated with an excess amount of $D_2O$ liquid in a hydrothermal bomb at 225 °C for 12 h, labelled as deuterated SZYO20.

Neutron powder diffraction (NPD) patterns were collected at room temperature on the time-of-flight high-resolution powder diffractometer (HRPD) at ISIS[76]. Data were collected for vacuum-dried and deuterated samples of SZYO20. The samples were loaded into thin-walled (150 μm) cylindrical vanadium foil sample holders with 7.94 mm diameter. The sample holders were mounted on the rotary sample changer and maintained under the sample vacuum tank ( ~ 0.15 mbar)

for the duration of the measurements. Full intensity profiles were recorded with all three detector banks: bank 1 the highest resolution backscattering bank, bank 2 the 90-degree detectors and bank 3 the low-resolution forward scattering array. Samples were each measured for several hours. Data were normalised to the incident spectrum and corrected for instrument efficiency using a V:Nb null scattering standard, and then corrected for sample attenuation using the software Mantid[77]. The attenuation was calculated from the total scattering cross section, the absorption cross-section and the number density of each sample based on its chemical stoichiometry, volume and mass. As the exact amount of $D_2O$ in the deuterated samples was not known at the time, their scattering and absorption cross-sections were calculated on the basis of the anhydrous chemical composition. Since the concentration of $D_2O$ in the structure proved to be very small, its contribution to the attenuation will be negligible. The collected NPD patterns were refined using the Rietveld method with the GSAS-II software[75]. Only data from bank 1 was used for refinement. Details of the refinement, including the process by which the deuterium atoms were located, are given in the Supplementary Note 3.

### Measurements of ion conductivity and ion transfer number

To measure the conductivity of sintered oxide pellets with a diameter around 13 mm, two side surfaces of a pellet (~2 mm in thickness) are coated by Silver Conductive Ink (Alfa Aesar) to form Ag electrodes. The Ag painting layers were dried in an oven at 130 °C for 150 min. A sandwich-structure cell with a layer of pellet between two layers of silver mesh was then immobilized in a home-made jig. Since the conduction of hydroxide ions requires water or steam as a medium, the ionic conductivity was measured either in a beaker fulfilled with deionized water (Supplementary Fig. 5) or in a sealed quartz tube in which the humidified compressed air was passing through. For the conductivity measurements in water, the temperature of water was detected and controlled by a hot plate connected with thermocouple. For the conductivity measurements in wet air, the compressed air was humidified by passing through boiling water at a flow rate of 100 mL min$^{-1}$ and then flowed into the jig sealed in a vertical furnace to be heated from 20 to 600 °C. The temperature of wet air around the pellet was read by a thermocouple linked to Solartron 1470E CellTest System.

To rule out the effect of water or solutions surrounding pellets on the ionic conductivity, the conductivity of pellets was also measured using a home-made fuel cell jig shown in Supplementary Fig. 33. The electrolyte pellet was coated with 1 cm$^{-1}$ of silver paste on two sides, matching the active area of the fuel cell testing jig. Water or KOH (85%, Thermo Scientific) solutions were pumped passing through the two chambers of jig. The electrolyte pellet was sealed by silicone gaskets to avoid being covered and crossed by solutions.

Before the conductivity measurements, all the pellets were heated in hot water at 90 °C for 20 h and washed to get rid of second phase $SrY_2O_4$ and the hydrated products. This process was repeated three times for each sample.

Electronic conductivity of the pellet was measured by a pseudo four-terminal DC method on a Solartron 1470E CellTest System. To work out the resistance caused by electronic conduction, 1 V constant DC voltage was applied on the pellet[41]. After the current had been saturated, the direct current electrical resistance ($R_{DC}$) was calculated from applied voltage ($V$) and saturated current ($I_{sat}$)[37,78]. Then the electrical conductivity ($\sigma_e$) was calculated on the basis of thickness, $\lambda$, and effective cross-sectional area of the electrolyte pellet, $A$, as follows[37,41]:

$$R_{DC} = V/I_{sat,} \qquad (4)$$

$$\sigma_e = \lambda/(R_{DC} \times A). \qquad (5)$$

Electrochemical impedance spectrum (EIS) was acquired using a Solartron 1455 frequency response analyser (FRA) with 10 mV amplitude and frequency range of 1 MHz to 0.01 Hz to measure total conductivities ($\sigma_t$). Based on the relation between electronic conductivity and total conductivity, the ionic conductivity $\sigma_i$ is calculated as:

$$\sigma_i = \sigma_t - \sigma_e \qquad (6)$$

The electronic conduction transfer number ($t_e$) is $\sigma_e/\sigma_t \times 100\%$. Then the corresponding ion transfer number ($t_i$) can be measured as follows:

$$t_i = (1 - \sigma_e/\sigma_t) \times 100\% \qquad (7)$$

The activation energy ($E_a$) for ion transport was calculated by the following equation:

$$\ln(\sigma T) = \ln \sigma_0 - \frac{E_a}{RT} \qquad (8)$$

where $\sigma_0$ is a preexponential factor, $R$ is the gas constant, and $T$ is temperature.

EIS tested in a range from the high frequency of 10 MHz were performed using VIONIC potentiostat from Metrohm Autolab, controlled by Intello 1.4.0 software. The equivalent circuit fit, simulations, and Kramers–Kronig test were performed by using Nova 2.1.6 software.

The conductivity and pH of the water used to soak the pellets and measurement jig during the test have been recorded immediately after the completion of the conductivity measurement for pellets in water. Conductivity and temperature of water were measured with an Orion Star™ A212 Conductivity Benchtop Metre and an Orion 013005MD 4-Cell Conductivity Probe (Thermo Scientific). The pH value of water was measured with an Orion Star™ A214 PH/ISE Benchtop Meter (Thermo Scientific) and a glass bulb pH electrode (ELIT PH-2011). The results are summarized in Supplementary Table 4.

### Identification of the charge carrier in ionic conductors

In order to determine which ion was conducted by the solid ionic conductors, a concentration cell with $Ag/Ag_2O$ electrodes was made according to the method described in a previous report[37]. A perovskite oxide pellet was clamped between two chambers of the H-cell filled with NaOH solution (Supplementary Fig. 10). A Nafion™ 212 membrane (FuelCellStore) and an anion exchange membrane (AEM) (Fumapem FAA, FuelCellStore) purchased commercially were measured for comparison (details in Supplementary Note 1).

In order to further verify that the cation transport number measured in the concentration cell represents proton conduction, the proton transference number ($t_+$) of SZYO20 pellet at room temperature was determined by chronoamperometry in conjunction with EIS measurements[41,79]. The corresponding measurements were similar to the conductivity measurements of electrolyte pellet in water using the jig shown in Supplementary Fig. 33. Nafion membranes were used as blocking electrodes to fabricate Nafion|SZYO20|Nafion assembly (Supplementary Fig. 11a) to block all the ions expect protons at the electrolyte/electrode interface[78,80,81]. The initial current ($I_0$) and final steady-state current ($I_s$) were recorded when a DC polarisation voltage ($V$) of 1 V was applied across the cell. A.c. impedance measurements were carried out to determine the electrolyte resistances before ($R_0$) and after ($R_s$) the polarisation process. $t_+$ was calculated as follows:

$$t_+ = \frac{I_s(V - I_0 R_0)}{I_0(V - I_s R_s)} \qquad (9)$$

## KIE measurements

The SZYO20 electrolyte pellet was firstly treated in deionized water at 90 °C for 20 h (repeated three times), and then dried in a vacuum oven at 120 °C overnight to remove the adsorbed water as much as possible. The conductivity was then measured after the pre-treated pellet was stored in deuterium water ($D_2O$) overnight. The kinetic isotope effect (KIE) is the ratio of ion conductivity of pellets in $H_2O$ to that in $D_2O$[41].

## Density functional theory simulations

All density functional theory (DFT) simulations were performed with the CP2K code, which uses a mixed Gaussian/plane-wave basis set[82]. Double polarisation quality Gaussian basis sets[83] and plane-wave cutoff of 400 Ry for the auxiliary grid were used in conjunction with the Goedecker-Teter-Hutter pseudopotentials[84]. The DFT calculations including geometry relaxation, single point energies, and ab initio molecular dynamics (AIMD) simulations were conducted using the Perdew-Burke-Ernzerhof (PBE) exchange-correlation functional[85], with Grimme's D3 van der Waals correction (PBE + D3)[86]. All DFT calculations were performed in the Γ-point approximation with a sufficiently large supercell. AIMD simulations within the Born-Oppenheimer approximation were performed in the canonical (NVT) ensemble. A timestep of 0.5 fs was used for the integration of the equation of motion, and the simulations were run for 20 ps (40,000 AIMD steps following 2,000 steps of equilibration run that has a strong thermostat coupling). The temperature of the AIMD simulation was 400 K, which was controlled by the canonical sampling through velocity rescaling thermostat[87] using a time constant of 50 fs.

## Electrode preparation for fuel cell measurement

Plain carbon fibre cloth (0.35 mm thickness, E-TEK) was used as the substrate for the catalysts. The carbon cloth electrode ($1 \times 1$ cm²) was sonicated in diluted hydrochloric acid, deionized water, and iso-propanol for 1 min respectively. PtIr/C catalysts (20 wt% of metals loading) were prepared by the borohydride reduction process[88] using $K_2PtCl_6$ (Pt 39.6%, Thermo Scientific™), $IrCl_3 \cdot 3H_2O$ (53–56% Ir, Thermo Scientific™) and Vulcan XC72 carbon black (Cabot) as precursors. The atomic ratio of Pt:Ir is 50:50. The catalyst ink was made up of 80 mg PtIr/C powder, 500 μL isopropanol and 145 μL 5 wt% Nafion solution (Sigma Aldrich). The ink slurry was ultrasonicated for 1 h and then brushed onto the pre-treated carbon cloth. The electrode was dried in an oven at 80 °C. The loading of PtIr was about 1.2 mg cm⁻². The Pt/C electrode (20 wt% Pt on carbon black, Alfa Aesar) was prepared in the same way and the loading of Pt was about 1.3 mg cm⁻².

## Hydrogen fuel cell fabrication and measurements

A SZYO20 pellet (1.8 mm of thickness, 19.2 mm of diameter) was employed as the electrolyte. The symmetric hydrogen fuel cell was assembled using Pt/C electrode as both the anode and cathode. The loading of Pt is about 1.3 mg cm⁻². The effective area of the fuel cell was $1 \times 1$ cm². Hydrogen gas was passing through a humidifier to flow into the anode at 5 mL min⁻¹, whilst 20 mL min⁻¹ humidified compressed air was flowing into the cathode field of fuel cell system. The pressure of both hydrogen and compressed air is at ambient pressure. The polarisation curves and power density curves were measured through a Solartron 1287 A electrochemical interface controlled by electrochemical software Corr-Ware/CorrView. The EIS data of the fuel cell was collected by the Solartron 1260 A Electrochemical Station at a frequency range of 1 MHz to 0.01 Hz and fixed potential of 10 mV bias.

## Direct ammonia fuel cell fabrication and measurements

A SZYO20 pellet (1.3 mm of thickness, 19.2 mm of diameter) was used as the electrolyte. PtIr/C and Pt/C electrode was used as ammonia oxidation reaction (AOR) anode and oxygen reduction reaction (ORR) cathode respectively in direct ammonia fuel cell (DAFC) measurements. The loading of PtIr was about 1.2 mg cm⁻² and the loading of Pt was about 1.3 mg cm⁻². The effective area of the fuel cell was $1 \times 1$ cm². An ammonia solution, i.e., 35 wt% $NH_3H_2O$ (35%, Fisher Chemical™) or 35 wt% $NH_3H_2O$ + 3 M KOH, was pumped at a flow rate of 1 mL min⁻¹ into anode channels. Compressed air was passing through 100 °C humidifier at 20 mL min⁻¹ then into the cathodic chamber. The durability test of the DAFC was carried out at a fixed current density of 4 mA cm⁻² at an operating temperature of 60 °C a few days after the polarisation curves test. The cell temperature was controlled by heating cartridges sensed by thermocouples. The pressure of both the ammonia solution at the anode and compressed air at the cathode is at ambient pressure. The fuel cell performance was measured by a Solartron 1287 A Electrochemical Interface coupled with a Solartron 1260 controlled by electrochemical software CorrWare/CorrView and Z-Plot/Z-view. The a.c. impedance was measured in the frequency range between 1 MHz and 0.01 Hz at the amplitude of the a.c. signal 10 mV. The polarisation curves were obtained at different temperatures of the fuel cell and a scan rate of 5 mV s⁻¹ was used in the measurements. The commercial AEM (Fumapem FAA-3–50, FuelCellStore) with 50 μm thickness was also employed in fuel cell measurements for comparison.

## Electrolysis of $H_2^{18}O$ and $D_2O$ using dense CZYO20 as the electrolyte

$H_2^{18}O$ and $D_2O$ were bought from CK Isotopes Limited and Sigma-Aldrich respectively. A CZYO20 pellet sintered at 1500 °C (0.8 mm of thickness, 16.7 mm of diameter) was used as the electrolyte. $RuO_2$ (Sigma Aldrich) and Pt/C was used as oxygen evolution reaction (OER) anode and hydrogen evolution reaction (HER) cathode catalysts respectively in the electrolytic cell, and the electrode preparation method is the same as that of fuel cells. The effective area of the cell was $1 \times 1$ cm². A total of 20% $H_2^{18}O$, pure $D_2O$ or the solution of 1 M KOH dissolved in $D_2O$ was pumped at a flow rate of 1 mL min⁻¹ into cathode channels while pure $H_2^{16}O$ water was pumped into the anodic chamber. The electrolysis performance was measured by the same set-up as DAFC. The durability test of the electrolysis cell was carried out at a fixed cell voltage of 2 V. The polarisation curves were obtained at a scan rate of 5 mV s⁻¹.

## Data availability

The data that support the findings of this study are available from the corresponding author upon reasonable request. Source data are provided with this paper. Figshare https://doi.org/10.6084/m9.figshare.24718497 Source data are provided with this paper.

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

## Acknowledgements

The authors thank EPSRC (Grant No. EP/W035529/1) and Innovate UK (Grant Nos. 104010 and 133714) for funding. One of the authors (P.Z.) thanks the China Scholarship Council for PhD studentship (No. 201908430195). The authors thank the X-ray Diffraction Research Technology Platform for access to the XRD measurement equipment and the UK High-field solid-state NMR facility for access to the solid state NMR at the University of Warwick. The authors thank the use of the Sulis supercomputer through the HPC Midlands+ Consortium, and the

ARCHER2 supercomputer through membership of the U.K.'s HPC Materials Chemistry Consortium, which are funded by EPSRC Grant Nos. EP/T022108/1 and EP/R029431/1, respectively. The authors thank Dr. James Town for performing STA measurements. The authors thank Dr. Ben Breeze for performing Raman measurements on $H_2^{18}O$ samples and setting up high high-temperature Raman measurement stage.

## Author contributions

S.T. conceived and supervised the project. S.T., P.Z. designed the experiments. P.Z. performed the experiments except for solid-state NMR, neutron diffraction, STEM measurement, and DFT simulation, D.I. performed the experiments and data analysis of solid-state NMR. S.L. performed DFT simulations and related data analysis. AJ.B. analysed the neutron diffraction data. S.C. helped with SEM/EDX experiments, M.Z. helped with drawing and plotting. Y.H. helped with TEM measurements and analysis. AD.F. and CM.H. collected the HRPD neutron diffraction data. S.T. and AJ.B. carried out Rietveld refinement. S.T. and P.Z. analysed the data and wrote the manuscript with input from D.I. for the solid-state NMR part, input from S.L. for DFT part, and input from AJ.B. for the neutron diffraction part. All authors commented on the manuscript.

## Competing interests

S.T. and P.Z. are listed as co-inventors on a pending UK & National patent application related to this work filed by University of Warwick. All other authors declare no competing interests.
