## [Peer Review File · Nature Communications]

A fast ceramic mixed OH⁻/H⁺ ionic conductor for low temperature fuel cellsReviewer #1 (Remarks to the Author):

Generally, the search for novel proton-conducting inorganic materials is highly relevant for energy conversion/storage applications. This paper suggests SrZr_{0.8}Y_{0.2}O_{3-d} as proton conductor for low-T applications, and claims an extremely high conductivity of 0.01 S/cm at 90°C when the material is in contact with liquid water (which is quite surprising given the known high-T conductivities of related perovskite proton conductors). However, the observation of secondary phases in this sample makes it clear that the reported conductivity does not come from the SrZr_{0.8}Y_{0.2}O_{3-d} perovskite.

Overall this paper has too many and too severe issues, such that I cannot recommend it for publication (this holds for any journal! these issues need to be resolved before submitting to any other journal). The paper contains also incorrect statements/interpretations that can be misleading for a broader audience (examples below). Most probably the "impurity phases" or even capillary condensation of water in tiny pores lead to the high conductivity, not the actual SrZr_{0.8}Y_{0.2}O_{3-d}.

(1) a fundamental point: the water activity of liquid water and gaseous water (pH₂O) at same T are identical because they are in equilibrium. So - unless the exposure to liquid water leads to structural reorganization as for Nafion - there is no reason why exposure to liquid water should drastically increase the proton uptake and conductivity

(2) it remains elusive whether the samples are predominant proton or hydroxide ion conductors (cf. also comment on OH⁻ transference number below). The distinction is important because in a perovskite structure the motion of hydroxide ions needs oxygen vacancies (the structure is too closely packed to allow for interstitial hydroxide ions), thus at low T and high water activity the hydroxide conductivity in the perovskites is expected very low. In contrast, protons can move in absence of V_o.

(3) the SrZr_{0.8}Y_{0.2}O_{3-d} sample is not single phase, as stated on pg. 5 line 85. For such strong claims as made here, one has to insist on well-defined and single-phase materials, in particular when only the non-singlephase material shown the special behavior. The SrY₂O₄ phase (amount of 8% is really high!) is at least partly water-soluble, which might explain why the sample behaved different in contact with liquid than gaseous water. On pg. 8 authors discuss amorphous Y(OH)₃ along grain boundaries as conduction path. It is expected that such an "accidentally formed" conduction path creates batch-to-batch reproducibility problems.

Anyway, the claim in title and abstract that "SrZr_{0.8}Y_{0.2}O_{3-d}" is the proton conductor is undermined by the SrY₂O₄/amorphous Y(OH)₃ observation. Properly prepared single-phase SrZr_{0.8}Y_{0.2}O_{3-d} perovskite should show a behavior between BaZrO₃ and CaZrO₃

(4) what is the reason why only SrZr_{0.8}Y_{0.2}O_{3-d} forms an Y-rich secondary phase? the perovskite phase is a very stable structure, so observation of a secondary phase suggests the overall cation stoichiometry deviated from the intended values. This must be checked by chemical analysis (ICP-OES or similar), in particular since for several of the used nitrates the actual water content may deviate from the nominal values, even between different purchased batches (was this checked?). Was the preparation of SrZr_{0.8}Y_{0.2}O_{3-d} repeated independently, and does it yield the same phase composition?

other points

- pg 2 middle: for LSGM, not only Ishihara but also Goodenough should be cited
- pg 2 bottom: for BZY conductivity, not only 7,8 but also 14 should be cited here
- fig 1 would be easier to "read" if the usual orientation of perovskite structures was used
- pg 4 line 67 "it is necessary to increase the carrier concentration": BZY with 20% Y doping is fully hydrated at T below 300°C, i.e. it has already 0.2 protons per formula unit. There is not too much space for further increase of this carrier concentration before mutual proton-proton repulsion would decrease the mobility
- pg 5 line 81: in contrast to oxide ion migration which moves and extended ion through the lattice, proton migration (a proton is a point charge!) rather needs suitable binding partners for the transfer than "free volume". Arguing in terms of free volume is misleading
- pg 9 line 136 "steam concentration may not be enough to form continuous pathway for OH-/H+". This is a misconception. BZY perovskites are fully hydrated in 3% steam below 300°C, and for proton transport they do not need a continuous "proton chain" but rather empty proton acceptor sites
- fig 4c: the sharp water loss at 100°C suggests that most water is not chemically bound, but rather contained in pores e.g. by capillary condensation
- pg 22: density of the sintered pellets must be given!
- the fact that separately prepared Y₂O₃ and Y(OH)₃ (with different morphology than amorphous Y(OH)₃ along the grain boundaries) has a lower conductivity Suppl. Fig. 7 does not suffice to prove the high conductivity is in the SrZr_{0.8}Y_{0.2}O_{3-d}
- suppl table 3: "t_{Na}" sodium transference number in a dense ceramic proton conductor?? (anyway mobility of small proton and large Na⁺ cannot be compared) high t_{OH⁻} of 0.76 or undoped SrZrO₃ which has no Vo..? hydroxide transference number of 0.13 for Nafion? overall these values make the transference number results questionable
- language proofreading is needed, e.g. "the phase was commonly existed..." and other examples

Reviewer #2 (Remarks to the Author):

In this manuscript " A fast ceramic mixed OH-/H⁺ ionic conduct for low temperature fuel cells", the authors claimed that the conventional well-known protonic conducting perovskite oxides such as SrZr_{0.8}Y_{0.2}O₃ are actually mixed OH⁻ and H⁺ conductors even at room temperature when the samples are sufficiently hydrated, and the conductivity reached as high as 0.01 S cm⁻¹ at 90°C, which is comparable to the well known conductivity of protonic conducting perovskites at 600 °C. They used

different characterization techniques such as solid-state NMR spectra, Raman spectra, and KIT measurement to confirm the presence of OH⁻ conduction pathway in the system. They further fabricated near ambient temperature solid oxide fuel cells and successfully demonstrated the single cell performance use hydrogen and ammonia as fuels, although the power density is not attractive, but as a proof-of-concept, is still interesting.

As a whole, the results in this manuscript is interesting, based on our past experience in this field and their experiments, the concept is reliable. The main innovative ideal of this research is that the typical proton-conducting perovskite oxides like SrZr_{0.8}Y_{0.2}O₃ are actually also OH⁻ conductor at low temperature with surprisedly high conductivity (reaching the level of 10⁻² S cm⁻¹). Actually, protonic conducting perovskites can conduct OH⁻ has been reported, such as the authors mentioned ref. 22, 23, and also by DFT calculations (J.Phys. Chem. C, 2020, 114, 8024-8033, <http://dx.doi.org/10.1002/aenm.202200392>)(although in these two DFT calculations papers, they mistook OH⁻ conduction as proton conduction). The work could be published in Nature Communications, however, significant improvements or additional works are still needed.

- 1) The authors believed that the improved conductivity in their study compared to the others in literature is due to the higher humidity was used in their experiment, leading to higher hydration degree of their material, to support this conclusion, they need to do additional experiment, such as TGA, to confirm that the hydration degree is indeed greatly improved with the increase of humidity of the surrounding atmosphere.
- 2) The explanation of the higher conductivity of SrZr_{0.8}Y_{0.2}O₃ than BzZr_{0.8}Y_{0.2}O₃ and CaZr_{0.8}Y_{0.2}O₃ is not very convincing and supportive, additional experiments or DFT calculations are needed.
- 3) For the conductivity measurement, the authors sintered the pellets at 1300°C, typically the protonic conducting electrolyte are very difficult to be sintered to dense pellets, and a sintering temperature of higher than 1600°C is usually needed, the authors need to provide the relative density of their sintered pellets.
- 4) Again, for the conductivity measurement, the authors emerged the whole pellets in water, it is well known that the alkaline element in the alkaline contained perovskites like SrZr_{0.8}Y_{0.2}O₃ could be slowly dissolved in water with the formation of Sr(OH)₂, while Sr(OH)₂ is an electrolyte, the authors should give strong evidence that the measured conductivity is not from such contribution, at least not the main contribution. So, after the conductivity measurement, the ion concentrations in the water should be measured by ICP.
- 5) As the authors claimed that the low conductivity of previous works related to protonic conducting perovskites are due to the low humidified gas was used. However, by using high humidified gas in current study, the high concentration of water vapor will significantly dilute the air for the cathode and fuel gas for the anode, which will greatly reduce the overall cell energy efficiency. The authors should discuss about the negative points of their concept.
- 6) In their test, still precious metal Electrocatalysis are used. IN principle, by use OH⁻ conducting electrolyte, non precious metals can be used as the electrocatalysts, like in normal alkaline fuel cells, why the authors did not try other cost-effective catalysts?

7) The power output is not attractive in current study. As claimed in their discussion, the low power output is related to the thick electrolyte, but I am afraid due to the high water vapor concentration, the electrode reaction kinetics will be greatly retarded, I suggest the authors to conduct three different electrolyte thickness to find out the impact on the power output.

8) Up to hundred hours of fuel cell stability is also necessary to confirm the feasibility of the new concept. We should be careful that the conversion of Sr to form Sr(OH)₂ under high humidity could also bring OH⁻ conductivity in the material, however, this OH⁻ could be easily lost, such long-term stability test is a necessarily.

Reviewer #3 (Remarks to the Author):

Authors of this manuscript claimed the perovskite oxide SrZr_{0.8}Y_{0.2}O_{3-δ} as a high ionic conductor (0.01 S/cm at 90°C in water and wet air) for near ambient temperature solid oxide fuel cells (NAT-SOFCs). However, this conductivity is 20 times lower than that (0.2 S/cm) Nafion film. Furthermore, this measured ion conductivity is also questionable. When SrZr_{0.8}Y_{0.2}O_{3-δ} was in water, some ions can be dissolved in water and thus the dissolved ions (instead of H⁺/OH⁻ ions) would be main contributors for the measured ionic conductivity. This was consistent with its negligible cell performance (0.34 mW/cm²), which is 2000 times lower than that (>1000 mW/cm²) of proton exchange membrane fuel cells (PEMFCs). Therefore, the manuscript is not only lack of novelty, but also has serious principle issues. It should be rejected.

Manuscript ID: NCOMMS-22-36610-T

Title: A fast ceramic mixed OH⁻/H⁺ ionic conductor for low temperature fuel cells

Thank the editor and reviewers for your time on reading this manuscript. Thank the reviewers for the constructive comments. We have collected additional experimental data and carried out a DFT calculation, which are added to the revised version. The quality of this paper has been significantly improved. The responses to the reviewers' comments are highlighted in blue.

Reviewers' comments:

Reviewer #1 (Remarks to the Author):

Generally, the search for novel proton-conducting inorganic materials is highly relevant for energy conversion/storage applications. This paper suggests SrZr_{0.8}Y_{0.2}O_{3-d} as proton conductor for low-T applications, and claims an extremely high conductivity of 0.01 S/cm at 90°C when the material is in contact with liquid water (which is quite surprising given the known high-T conductivities of related perovskite proton conductors). However, the observation of secondary phases in this sample makes it clear that the reported conductivity does not come from the SrZr_{0.8}Y_{0.2}O_{3-d} perovskite. Overall this paper has too many and too severe issues, such that I cannot recommend it for publication (this holds for any journal! these issues need to be resolved before submitting to any other journal). The paper contains also incorrect statements/interpretations that can be misleading for a broader audience (examples below). Most probably the "impurity phases" or even capillary condensation of water in tiny pores lead to the high conductivity, not the actual SrZr_{0.8}Y_{0.2}O_{3-d}.

Thank this reviewer for highlighting the importance of searching for novel proton-conducting inorganic materials. More precisely, it should be wider than proton-conducting materials. Any better fast H⁺/OH⁻/O²⁻ ionic conducting materials, particularly

working at low or near ambient temperatures, are important, for use as electrolytes for fuel cells and electrolyzers etc. It is not limited to proton conductors.

In our research, we found that perovskite oxides $AZr_{0.8}Y_{0.2}O_{3-5}$, with $A = Ca, Sr, Ba$, are all mixed OH^-/H^+ conductors at a temperature below $100\text{ }^\circ\text{C}$, in which the OH^- ionic transfer number is higher than H^+ ionic transfer number, which means they are not a conventional proton conductors, more are hydroxide ionic conductors. This is different from the common knowledge in these oxide materials, they are proton conductors at high temperature ($\geq 500\text{ }^\circ\text{C}$).

We understand this reviewer's concern on 'impurity phase' caused high ionic conductivity. In our study, we also notice the second phase SrY_2O_4 and its hydrolysis product, $SrCO_3$, $Sr(OH)_2$ and $Y(OH)_3$ may lead to the high ionic conductivity. Therefore we highlight the importance of 'removal of the second phase' in the text. The as-prepared $SrZr_{0.8}Y_{0.2}O_{3-5}$ sample was washed in hot water ($90\text{ }^\circ\text{C}$) for three times to remove the SrY_2O_4 because SrY_2O_4 hydrolyses at $90\text{ }^\circ\text{C}$ although it does not hydrolyse at room temperature. The yield $Sr(OH)_2$ is water soluble thus dissolved in water and removed. After the sample has been pre-washed, from HRTEM/EDX, we cannot see the residual SrY_2O_4 and residual $Sr(OH)_2$. However, we did find some Y-rich area, which is likely to be $Y(OH)_3$ or Y_2O_3 . We have measured the conductivity of both $Y(OH)_3$ and Y_2O_3 , both exhibits low conductivity ($10^{-5} - 10^{-4}\text{ S cm}^{-1}$ at $90\text{ }^\circ\text{C}$) thus the small Y-rich phase will have little contribution to the measured high ionic conductivity of $SrZr_{0.8}Y_{0.2}O_{3-5}$.

To rule out the possibility of second phase SrY_2O_4 , we also synthesised similar oxides $CaZrO_3$, $CaZr_{0.8}Y_{0.2}O_{3-5}$, $BaZrO_3$, $BaZr_{0.8}Y_{0.2}O_{3-5}$, in which there is no SrY_2O_4 at all. In these oxides, the Y-doped $CaZrO_3/BaZrO_3$ also exhibits much higher ionic conductivity than pure $CaZrO_3/BaZrO_3$ indicating oxygen vacancies due to doping plays the critical role for the high ionic conductivity (Supplementary Fig. 10). The ionic conductivity reached 10^{-3} S cm^{-1} at $90\text{ }^\circ\text{C}$ in water for both $CaZr_{0.8}Y_{0.2}O_{3-5}$ and $BaZr_{0.8}Y_{0.2}O_{3-5}$, which is also very high, considering the temperature. High ionic conductivity in oxides such as $CaZr_{0.8}Y_{0.2}O_{3-5}$, $BaZr_{0.8}Y_{0.2}O_{3-5}$ was also observed although there is no SrY_2O_4 at all, indicating the high ionic conductivity in $SrZr_{0.8}Y_{0.2}O_{3-5}$ is also due to the presence of oxygen vacancies, the same as $CaZr_{0.8}Y_{0.2}O_{3-5}$ and $BaZr_{0.8}Y_{0.2}O_{3-5}$. However, we cannot completely rule out there could be very little residual SrY_2O_4 or $Sr(OH)_2$, which might have little contribution to the measured total conductivity in $SrZr_{0.8}Y_{0.2}O_{3-5}$. This has been further confirmed by measuring the conductivity of water used for the conductivity measurement (Supplementary Table 4).

A very important phenomenon is, the ionic conductivity of pre-washed $SrZr_{0.8}Y_{0.2}O_{3-5}$ is stable in humidified air during the measured 130 hours (Fig. 4d). If the conductivity is due to the residual $Sr(OH)_2$, then it will inevitably react with CO_2 in air to form insulator $SrCO_3$ then the measured conductivity should keep decreasing. Accordingly, the $SrCO_3$ content in the XRD for sample before and after the conductivity stability measurement should increase as well. From XRD, the $SrCO_3$ in the sample before and after the

conductivity stability measurement is both around 2%. The good stability in conductivity in humidified air again indicates the high ionic conductivity in $\text{SrZr}_{0.8}\text{Y}_{0.2}\text{O}_{3-5}$ is likely due to the oxygen vacancies, the same as $\text{CaZr}_{0.8}\text{Y}_{0.2}\text{O}_{3-5}$ and $\text{BaZr}_{0.8}\text{Y}_{0.2}\text{O}_{3-5}$.

As for 'capillary condensation of water in tiny pores lead to the high conductivity'. We do not believe this is the case. It is well known that the conductivity of pure water is very low. Although there are absorbed water in the tiny pores in oxides, the measured ionic conductivity should be very low too. If water in tiny pores of a materials can lead to high ionic conductivity, this should have been observed before. According to this assumption, any porous material may exhibit high ionic conductivity when exposed in water. This is not true. For example, the ionic conductivities of pure CaZrO_3 , SrZrO_3 and BaZrO_3 in water are quite low. This rules out the effect of 'capillary condensation of water in tiny pores lead to the high conductivity'. From the experimental data, the conductivity of used water for conductivity measurements is much lower than the measured conductivity of the hydrated oxides themselves (Supplementary Table 4). Although water with some level of conductivity is absorbed in the pores of oxides to have some kind of 'capillary condensation', it will only have a little contribution to the overall measured conductivity.

(1) a fundamental point: the water activity of liquid water and gaseous water (pH_2O) at same T are identical because they are in equilibrium. So - unless the exposure to liquid water leads to structural reorganization as for Nafion - there is no reason why exposure to liquid water should drastically increase the proton uptake and conductivity

In our paper, liquid water does not mean pure liquid water, it could be humidified gas which contains enough liquid water such as those in Figure 4d, humidified air. If the temperature is above 100 °C, under pressurized condition, there will be liquid water too, theoretically should also work.

The expose the samples in liquid water, hydration is an essential step to convert the oxygen defects into protonic defects, in another word, to convert oxide into oxyhydroxide with the formation of OH^- species.

The DFT calculation in newly added Fig. 6 and Supplementary Figs. 17,18, indicating the migration of OH^- ions, may need oxygen vacancies and temporary formation of hydrogen bond of OH^- species with the closest neighbouring oxygen. It is relatively easy to form hydrogen bonds in the presence of liquid water while this is difficult to form hydrogen bonds in gaseous water (steam). This is another reason that, in an environment containing liquid water is important in order to achieve high ionic conductivity.

(2) it remains elusive whether the samples are predominant proton or hydroxide ion conductors (cf. also comment on OH⁻ transference number below). The distinction is important because in a perovskite structure the motion of hydroxide ions needs oxygen vacancies (the structure is too closely packed to allow for interstitial hydroxide ions), thus at low T and high water activity the hydroxide conductivity in the perovskites is expected very low. In contrast, protons can move in absence of V_o.

In our concentration cell measurements, the oxides of AZrO₃, AZr_{0.8}Y_{0.2}O₃₋₅ (A = Ca, Sr, Ba), all six samples, they are all mixed OH⁻ and H⁺ ionic conduction, not pure H⁺ nor pure OH⁻ ionic conduction (Supplementary Table 3).

DFT calculation (added Fig. 6, Supplementary Figs. 17,18) indicates that at a relatively low temperature, both H⁺ and OH⁻ ions can move. The migration of OH⁻ ions relies on oxygen vacancies and the temporary formation of hydrogen bonds. This is consistent with the low conductivity of SrZr_{0.8}Y_{0.2}O₃₋₅ when the temperature is above 100 °C at ambient pressure (inserted Figure in Fig. 4c). This is because in the absence of liquid water, it is hard to facilitate the hydrogen bond formation to diffuse OH⁻ ions.

As for whether OH⁻ ions can move in the form of interstitials, it is hard to judge at this stage. One of the key papers cited by Reviewer 2,

<http://dx.doi.org/10.1002/aenm.202200392>

Protonic Conduction in La₂NiO₄₊₅ and La_{2-x}A_xNiO₄₊₅ (A = Ca, Sr, Ba) Ruddlesden–Popper Type Oxides

OH⁻ ions can move through the interstitials in oxides with K₂NiF₄ structure, which is closely related to primitive perovskite structure, also closely packed (ref. 27).

Also pointed by reviewer 2, in a previous study, density functional theory (DFT) calculation indicates proton migration in perovskite oxide BaZr_{0.1}Ce_{0.7}Y_{0.1}Yb_{0.1}O₃₋₅ is in the form of OH in the presence of oxygen vacancies (ref. 21).

<https://doi.org/10.1021/acs.jpcc.0c01284>

This means, under certain conditions, OH⁻ ions can also migrate in perovskite oxides.

(3) the SrZr_{0.8}V_{0.2}O_{3-d} sample is not single phase, as stated on pg. 5 line 85. For such strong claims as made here, one has to insist on well-defined and single-phase materials, in particular when only the non-singlephase material shown the special behavior. The SrV₂O₄ phase (amount of 8% is really high!) is at least partly water-soluble, which might explain why the sample behaved different in contact with liquid than gaseous water. On pg. 8 authors discuss amorphous V(OH)₃ along grain boundaries as conduction path. It is expected that such an "accidentally formed" conduction path creates batch-to-batch reproducibility problems.

Anyway, the claim in title and abstract that "SrZr_{0.8}V_{0.2}O_{3-d}" is the proton conductor is undermined by the SrV₂O₄/amorphous V(OH)₃ observation. Properly prepared single-phase SrZr_{0.8}V_{0.2}O_{3-d} perovskite should show a behavior between BaZrO₃ and CaZrO₃

We understand the concern of this reviewer on the role of second phase SrV₂O₄ and its hydrolysed products Sr(OH)₂ and V(OH)₃. This is the key reason we pre-wash the sample in hot water (90 °C) for three times, managing to get rid of SrV₂O₄ and the hydrolysed products, before carrying out the conductivity measurements. HR-TEM/EDX indicates there is no residual Sr(OH)₂ or it is beyond the measurement limit because it is soluble in water. The solubility of Sr(OH)₂ in water is, 0.41 g/100 mL (0 °C) 1.77 g/100 mL (40 °C) 21.83 g/100 mL (100 °C)

https://en.wikipedia.org/wiki/Strontium_hydroxide

We did observe V-rich areas under HRTEM/EDX, we believe is likely residual V(OH)₃ from the hydrolysis of SrV₂O₄. Also we cannot rule out the possibility of V₂O₃. Thus we carried out the conductivity measurements of pure V(OH)₃ and V₂O₃ in water, the conductivity of both samples is in the range of 10⁻⁵ – 10⁻⁴ S cm⁻¹ at 90 °C (Supplementary Fig. 7b), at least two orders of magnitude lower than the observed 0.01 S cm⁻¹ for SrZr_{0.8}V_{0.2}O₃₋₅ at 90 °C in water. Therefore high ionic conductivity of SrZr_{0.8}V_{0.2}O₃₋₅ is unlikely due to the presence of V-rich secondary phase.

Actually we have synthesised single phases CaZrO₃, CaZr_{0.8}V_{0.2}O₃₋₅, BaZrO₃ and BaZr_{0.8}V_{0.2}O₃₋₅. There is no SrV₂O₄ at all because there is no elements Sr in CaZr_{0.8}V_{0.2}O₃₋₅ and BaZr_{0.8}V_{0.2}O₃₋₅, neither Sr nor V in CaZrO₃ and BaZrO₃. For consistency, all samples were pre-washed at 90 °C for three times before carrying out conductivity measurements although there is no SrV₂O₄ at all.

It was found the ionic conductivity in water for the two V-doped samples, CaZr_{0.8}V_{0.2}O₃₋₅ and BaZr_{0.8}V_{0.2}O₃₋₅ are at least 5 times higher than the un-doped samples CaZrO₃ and BaZrO₃ (Supplementary Fig. 10b,d). These experiments on single phase oxides clearly indicate the high ionic conductivity is related to oxygen vacancies. This is also consistent with the conclusion of DFT calculation, i.e., at low temperature, OH⁻ ions may move

through oxygen vacancies and temporary formed hydrogen bond. That is why we need liquid water to achieve high ionic conductivity.

From the conductivity data of single phase CaZrO_3 , $\text{CaZr}_{0.8}\text{Y}_{0.2}\text{O}_{3-6}$, BaZrO_3 and $\text{BaZr}_{0.8}\text{Y}_{0.2}\text{O}_{3-6}$, it can be deduced that the high ionic conductivity in $\text{SrZr}_{0.8}\text{Y}_{0.2}\text{O}_{3-6}$ is also related to the oxygen vacancy.

In addition, the preparation and conductivity measurement results for these perovskite oxides are repeatable. The synthesis of $\text{SrZr}_{0.8}\text{Y}_{0.2}\text{O}_{3-6}$ has been repeated independently for more than 12 times. The conduction path is not accidentally formed, it is reproducible from batch to batch.

(4) what is the reason why only $\text{SrZr}_{0.8}\text{Y}_{0.2}\text{O}_{3-d}$ forms an Y-rich secondary phase? the perovskite phase is a very stable structure, so observation of a secondary phase suggests the overall cation stoichiometry deviated from the intended values. This must be checked by chemical analysis (ICP-OES or similar), in particular since for several of the used nitrates the actual water content may deviate from the nominal values, even between different purchased batches (was this checked?). Was the preparation of $\text{SrZr}_{0.8}\text{Y}_{0.2}\text{O}_{3-d}$ repeated independently, and does it yield the same phase composition?

' $\text{SrZr}_{0.8}\text{Y}_{0.2}\text{O}_{3-6}$ forms an Y-rich secondary phase' is the nature of this material system. The same phenomenon was also observed by other research groups using very different synthesis methods (see new refs 29, 30, 31). Reducing the Y-content to only 10% at the B-site, $\text{SrZr}_{0.9}\text{Y}_{0.1}\text{O}_{3-6}$, the SrY_2O_4 phase is still there with reduced quantity. A possible reason is, SrY_2O_4 is a stable phase which can be easily formed at a relatively low temperature before the formation of perovskite phase. Once it is formed, it is difficult to get rid of.

However, single phase $\text{CaZr}_{0.8}\text{Y}_{0.2}\text{O}_{3-6}$ and $\text{BaZr}_{0.8}\text{Y}_{0.2}\text{O}_{3-6}$ has been successfully synthesised and their conductivities in water is much higher than the undoped phase CaZrO_3 and BaZrO_3 , so the chemicals we used in synthesis have no problems. The preparation of these perovskite oxides is reproducible, especially the synthesis of $\text{SrZr}_{0.8}\text{Y}_{0.2}\text{O}_{3-6}$ has been repeated independently for more than 12 times and it yield the same phase composition.

other points

- pg 2 middle: for LSGM, not only Ishihara but also Goodenough should be cited

A new reference 13,

Feng, M. & Goodenough, J. B. Improved Oxide Ion Electrolytes. MRS Online Proceedings Library (OPL) 369 (1994).

has been added in revised version.

- pg 2 bottom: for BZY conductivity, not only 7,8 but also 14 should be cited here

Old reference 14 has been cited there (new ref. 15).

- fig 1 would be easier to "read" if the usual orientation of perovskite structures was used

The orientation of the perovskite structure is shown on the x, y, z direction on the left bottom corner in Fig. 1.

We did not show the perovskite structure in the conventional polyhedral type in order to show the possible pathways of ions for H^+/OH^- ions through the oxygen vacancies.

Therefore, we still retain the same style for Fig. 1.

- pg 4 lien 67 "it is necessary to increase the carrier concentration": BZY with 20% Y doping is fully hydrated at T below 300°C, i.e. it has already 0.2 protons per formula unit. There is not too much space for further increase of this carrier concentration before mutual proton-proton repulsion would decrease the mobility

We believe this is a misunderstanding. The original text is:

'The ionic conductivity of Y-doped $SrZrO_3$, $SrZr_{0.95}Y_{0.05}O_{3-\delta}$ is approximately 7.0×10^{-4} $S\ cm^{-1}$ at 600 °C, which is not high enough for use as electrolyte for NAT-SOFCs. In order to improve the ionic conductivity at low temperatures, it is necessary to increase the concentration of charge carriers, i.e., the proton defects.'

We meant the B-site doping of only 5% in $\text{SrZr}_{0.95}\text{Y}_{0.05}\text{O}_{3-\delta}$ in reported paper is not high enough thus we increase the B-site doping level from the reported 5% to 20% in this study. This strategy has been demonstrated correct because the conductivity of pre-washed oxide with nominal composition $\text{SrZr}_{0.9}\text{Y}_{0.1}\text{O}_{3-5}$ is lower than that of $\text{SrZr}_{0.8}\text{Y}_{0.2}\text{O}_{3-5}$ (Fig. 4b).

- pg 5 line 81: in contrast to oxide ion migration which moves and extended ion through the lattice, proton migration (a proton is a point charge!) rather needs suitable binding partners for the transfer than "free volume". Arguing in terms of free volume is misleading

Thanks for the comments. The relevant text has been revised. Although the migration of protons does rely on 'free volume', the migration of large O^{2-} or OH^- ions still relies on large 'free volume'. Therefore the hypothesis still applies to the large OH^- ions. This is consistent with the observed ionic conduction of the investigated 6 oxides, which are mixed OH^-/H^+ conductors rather a pure H^+ ionic conductor.

- pg 9 line 136 "steam concentration may not be enough to form continuous pathway for OH^-/H^+ ". This is a misconception. BZY perovskites are fully hydrated in 3% steam below 300°C, and for proton transport they do not need a continuous "proton chain" but rather empty proton acceptor sites

We reach this conclusion is, the high ionic conductivity of the perovskite oxides such as $\text{SrZr}_{0.8}\text{Y}_{0.2}\text{O}_{3-5}$, $\text{BaZr}_{0.8}\text{Y}_{0.2}\text{O}_{3-5}$, $\text{CaZr}_{0.8}\text{Y}_{0.2}\text{O}_{3-5}$, all are higher than $10^{-3} \text{ S cm}^{-1}$ at 90 °C in water, was not previously observed when people measure the ionic conductivity in 3% steam containing H_2 or air. The relevant text has been revised.

Through DFT calculation (new Fig. 6, Supplementary Figs. 17,18), the OH^- migration also relies on the temporary formation of hydrogen bonds. We believe the hydrogen bond formation is relative to the partial pressure of steam in the environment. This could be another reason that the high ionic conductivity, particularly the OH^- ionic conductivity is dependent on high steam partial pressure while 3% is insufficient to lead to high ionic conductivity.

- fig 4c: the sharp water loss at 100°C suggests that most water is not chemically bound, but rather contained in pores e.g. by capillary condensation

The sharp decrease in conductivity in Fig. 4c indicates the ionic conductivity relies on the presence of liquid water. At a temperature above 100 °C and one atmosphere, water will exist in the form of gaseous state or, in another word, the hydrated $\text{SrZr}_{0.8}\text{Y}_{0.2}\text{O}_{3-6}$ start to loss water, both physically absorbed and chemically adsorbed, thus the conductivity keeps decreasing till ~ 250 °C (Fig. 4c). This also emphasise the importance of hydration and the presence of liquid water. The loss of water between 100 °C and 250 °C is not only the physically absorbed water, or you can call it 'capillary condensation', but also the chemically adsorbed water, or, in another word, the hydrated bonded water with the oxygen vacancies, according to equation (2).

Please note, although water can be physically absorbed in the pores, water will not make so much contribution on the ionic conductivity because of the low conductivity of water itself. The conductivity of used water is generally lower than that of the conductivity of oxide itself except for BaZrO_3 which exhibits very low conductivity (Supplementary Table 4).

- pg 22: density of the sintered pellets must be given!

The relative density calculated from the measured real density and the calculated theoretical density based on the refined lattice parameters are provided in revised version. In the investigated 6 oxides, the relative density is in a large range from ~ 46% for $\text{BaZr}_{0.8}\text{Y}_{0.2}\text{O}_{3-6}$ to ~ 90% for CaZrO_3 . Most of the samples have a relative density between 60-80%.

- the fact that separately prepared Y_2O_3 and $\text{Y}(\text{OH})_3$ (with different morphology than amorphous $\text{Y}(\text{OH})_3$ along the grain boundaries) has a lower conductivity Suppl. Fig. 7 does not suffice to prove the high conductivity is in the $\text{SrZr}_{0.8}\text{Y}_{0.2}\text{O}_{3-d}$

In general, materials with different morphology (round, needle, square etc.) may exhibit similar level of conductivity unless single crystals. Our understanding is, you mean the $\text{Y}(\text{OH})_3$, more precisely Y-rich phase in pre-washed $\text{SrZr}_{0.8}\text{Y}_{0.2}\text{O}_{3-8}$ is amorphous while the prepared Y_2O_3 and $\text{Y}(\text{OH})_3$ are crystallized thus their conductivity could be different. This

is possible. However, the quantity of the amorphous V-rich phase is very small, most of them are not contact with each other to form continuous pathways. Therefore, the probability of the high ionic conductivity of $\text{SrZr}_{0.8}\text{V}_{0.2}\text{O}_{3-\delta}$ is due to the amorphous V-rich phase is very low. The observed high ionic conductivity of single phase oxides, V-doped BaZrO_3 and CaZrO_3 , in the order of $10^{-3} \text{ S cm}^{-1}$ at 90°C in water (Supplementary Fig. 7), indicating the high ionic conductivity is likely due to the extrinsic oxygen vacancies. We tried to synthesise amorphous $\text{V}(\text{OH})_3$ but was unsuccessful.

The conductivity of V_2O_3 and $\text{V}(\text{OH})_3$ in Supplementary Fig. 7 is not to support the high ionic conductivity of $\text{SrZr}_{0.8}\text{V}_{0.2}\text{O}_{3-6}$. On the contrary, we separate prepared V_2O_3 and $\text{V}(\text{OH})_3$ and measure their conductivity against temperature in water, is to find out if the high ionic conductivity of $\text{SrZr}_{0.8}\text{V}_{0.2}\text{O}_{3-6}$ in water is related to the small residual V-rich phase, likely to be $\text{V}(\text{OH})_3$ due to the hydrolysis of SrV_2O_4 . Because the conductivity of pure V_2O_3 and $\text{V}(\text{OH})_3$ in water is in the order of $10^{-5} - 10^{-4} \text{ S cm}^{-1}$, it is unlikely that the high ionic conductivity of $\text{SrZr}_{0.8}\text{V}_{0.2}\text{O}_{3-6}$ around $10^{-2} \text{ S cm}^{-1}$ is due to the residual V-rich amorphous phase.

- suppl table 3: "t_Na" sodium transference number in a dense ceramic proton conductor?? (anyway mobility of small proton and large Na^+ cannot be compared) high t_OH- of 0.76 or undoped SrZrO_3 which has no Vo..? hydroxide transference number of 0.13 for Nafion? overall these values make the transference number results questionable - language proofreading is needed, e.g. "the phase was commonly existed..." and other examples

It is very difficult to find a method to identify the type of charge carriers and to measure the ion transfer number in water. Therefore we copy the method previously used to measure the ionic transfer of oxides in water. Precisely, the measured ion transfer number should be classified as cation and anion transfer numbers. For $\text{SrZr}_{0.8}\text{V}_{0.2}\text{O}_{3-6}$, the possible anions are O^{2-} and OH^- ions. Theoretically the O^{2-} migration may rely on oxygen vacancies, but do not rely on hydration. On the contrary, the migration of OH^- ions relies on hydration. Due the fact that the high ionic conductivity in these oxides relies on hydration, it is reasonably deduced that the mobile anions are OH^- ions, not O^{2-} ions.

As for cations, there are Sr^{2+} , Zr^{4+} , V^{3+} and H^+ (after hydration). The migration of large cations with high positive charge such as Sr^{2+} , Zr^{4+} , V^{3+} ions are very difficult at low temperature, e.g., near room temperature. On the contrary, the proton migration relies on hydration also much easier than other large cations. Therefore we reasonably deduce that the measured cation transfer is due to protons.

For all the investigated 6 oxides, pure and Y-doped $AZrO_3$, $A = Ca, Sr, Ba$, they are all mixed OH^-/H^+ ionic conductors. The measured OH^- ion transfer number is higher than that for H^+ ions.

This method itself is not perfect, as you spot Nafion contains 13% OH^- ionic conduction which is not correct. However, it may provide us with a guideline, that, cation or anion, which ions are the major charge carriers. In all the measurement for six oxides, anions, most likely OH^- ions, are the major charge carries when the oxides have been treated in water.

Besides $SrZrO_3$, we also measured the ionic transfer number for pure $CaZrO_3$ and $BaZrO_3$. They all exhibit mixed OH^-/H^+ while the transfer number for OH^- ions is relatively higher than that for H^+ ions. Although theoretically there is little oxygen vacancies in pure $AZrO_3$, $A = Ca, Sr, Ba$, the ionic conductivity of pure $AZrO_3$, $A = Ca, Sr, Ba$ in water is in the range of $10^{-3} - 10^{-5} S cm^{-1}$, which means some ions are mobile. The order of ionic conductivity in water for pure $AZrO_3$ is $BaZrO_3 < CaSrO_3 < SrZrO_3$. The measured ion transfer number can provide a guideline, cation or anion, which are the major charge carriers. It has been demonstrated anions, most likely OH^- ions are the major charge carriers in $AZrO_3$, $A = Ca, Sr, Ba$. The possible explanation is the intrinsic oxygen vacancies formed during the preparation process when fired at high temperature, although it is supposed to be very low concentration. Another possibility is the slowly dissolved perovskite in water may have a little but not the major contribution to the OH^- conduction of the hydrated pure oxides (Supplementary Table 4).

All the pure oxide, its conductivity in water is much lower than the Y-doped oxides (Fig. 4b, Supplementary Fig. 10). The measured ion transfer number of pure $AZrO_3$ does not mean the absolute values of their conductivities are also very high. They are two different concepts.

The language has been further polished.

Reviewer #2 (Remarks to the Author):

In this manuscript " A fast ceramic mixed OH^-/H^+ ionic conduct for low temperature fuel cells", the authors claimed that the conventional well-known protonic conducting perovskite oxides such as $SrZr_{0.8}Y_{0.2}O_3$ are actually mixed OH^- and H^+ conductors even at room temperature when the samples are sufficiently hydrated, and the conductivity reached as high as $0.01 S cm^{-1}$ at $90^\circ C$, which is comparable to the well known conductivity of protonic conducting perovskites at $600^\circ C$. They used different

characterization techniques such as solid-state NMR spectra, Raman spectra, and KIT measurement to confirm the presence of OH⁻ conduction pathway in the system. They further fabricated near ambient temperature solid oxide fuel cells and successfully demonstrated the single cell performance use hydrogen and ammonia as fuels, although the power density is not attractive, but as a proof-of-concept, is still interesting.

As a whole, the results in this manuscript is interesting, based on our past experience in this field and their experiments, the concept is reliable. The main innovative ideal of this research is that the typical proton-conducting perovskite oxides like SrZr_{0.8}Y_{0.2}O₃ are actually also OH⁻ conductor at low temperature with surprisedly high conductivity (reaching the level of 10⁻² S cm⁻¹). Actually, protonic conducting perovskites can conduct OH⁻ has been reported, such as the authors mentioned ref. 22, 23, and also by DFT calculations (J.Phys. Chem. C, 2020, 114, 8024-8033,<http://dx.doi.org/10.1002/aenm.202200392>)(although in these two DFT calculations papers, they mistook OH⁻ conduction as proton conduction). The work could be published in Nature Communications, however, significant improvements or additional works are still needed.

We sincerely thank this reviewer for the positive comments on the importance of this work and recommend publishing in Nature Communications after revision. The mentioned two papers regarding DFT calculation of OH⁻ ions in perovskite oxides have been cited in the revised version, references 21 and 27 respectively. These key references indicates theoretically migration of OH⁻ ions in perovskite oxides is feasible.

1) The authors believed that the improved conductivity in their study compared to the others in literature is due to the higher humidity was used in their experiment, leading to higher hydration degree of their material, to support this conclusion, they need to do additional experiment, such as TGA, to confirm that the hydration degree is indeed greatly improved with the increase of humidity of the surrounding atmosphere.

At a temperature below 100 °C, it is very difficult to carry out STA on these samples in highly humidified atmosphere to judge the degree of hydration because liquid water from the highly humidified atmosphere will condense and be physically absorbed by the oxide powder or directly condense and stay in the crucible thus difficult to judge how much water is due to physically absorbed and how much is due to chemically adsorbed at a temperature below 100 °C.

However, DFT calculation (new Fig. 6, Supplementary Figs. 17,18) indicates the diffusion of OH⁻ ions at low temperature relies on both oxygen vacancies and temporarily formed hydrogen bonds. The required high degree of hydration could be related to the second

requirement, the formation of hydrogen bonds, which is much easier to be formed in an environment with the presence of liquid water.

2) The explanation of the higher conductivity of $\text{SrZr}_{0.8}\text{Y}_{0.2}\text{O}_3$ than $\text{BzZr}_{0.8}\text{Y}_{0.2}\text{O}_3$ and $\text{CaZr}_{0.8}\text{Y}_{0.2}\text{O}_3$ is not very convincing and supportive, additional experiments or DFT calculations are needed.

DFT calculation (new Fig. 6, Supplementary Figs. 17,18) indicates at low temperature, both OH^- and H^+ ions are mobile, may have contribution to the ionic conduction.

The plausible explanation to the higher ionic conductivity in $\text{SrZr}_{0.8}\text{Y}_{0.2}\text{O}_{3-\delta}$ is the optimised results of two opposite effects, the requirement on 'free volume' for large ions such as OH^- ions, and the jumping distance of ions, which have been described in text. In specific, the conductivity of pure SrZrO_3 also exhibits the highest conductivity among AZrO_3 (A = Ca, Sr, Ba) (new Supplementary Fig. 10e), which means SrZrO_3 is the lattice with optimized 'free volume' and jumping distance for OH^- ions, and there could be a small amount of intrinsic oxygen vacancy in the fired oxides to allow OH^- ions diffusion.

3) For the conductivity measurement, the authors sintered the pellets at 1300oC, typically the protonic conducting electrolyte are very difficult to be sintered to dense pellets, and a sintering temperature of higher than 1600oC is usually needed, the authors need to provide the relative density of their sintered pellets.

The relative densities for all the investigated oxides are provided in the revised version under method and Supplementary Note 3. They are not very dense in order to facilitate the hydration of the samples in water to quickly achieve high ionic conductivity.

4) Again, for the conductivity measurement, the authors emerged the whole pellets in water, it is well known that the alkaline element in the alkaline contained perovskites like $\text{SrZr}_{0.8}\text{Y}_{0.2}\text{O}_3$ could be slowly dissolved in water with the formation of $\text{Sr}(\text{OH})_2$, while $\text{Sr}(\text{OH})_2$ is an electrolyte, the authors should give strong evidence that the measured conductivity is not from such contribution, at least not the main contribution. So, after the conductivity measurement, the ion concentrations in the water should be measured by ICP.

Thanks for the comments. The contribution of slowly dissolved perovskite cannot be completely ruled out. We did not measure the ion concentration by ICP, but we measured the conductivity and pH value of the water after the whole period of conductivity measurements. In general, the pH value of water increased after the conductivity measurements (Please see Supplementary Table 4) indicates the slow dissolution of perovskite oxides. However, the conductivity of water measured by conductivity meter is generally much lower than the conductivity of the oxide pellets measured by a.c. impedance, indicating the measured conductivity of the oxide may have a small contribution from the slowly dissolved perovskite, such as $\text{Sr}(\text{OH})_2$, but majority of the measured conductivity of the oxides are still from the sample itself. The introduction of extrinsic oxygen vacancies in oxides is important to achieve high ionic conductivity in water.

5) As the authors claimed that the low conductivity of previous works related to protonic conducting perovskites are due to the low humidified gas was used. However, by using high humidified gas in current study, the high concentration of water vapor will significantly dilute the air for the cathode and fuel gas for the anode, which will greatly reduce the overall cell energy efficiency. The authors should discuss about the negative points of their concept.

Thanks for the comments. The currently state-of-the-art hydrogen fuel cells based on acidic Nafion membrane electrolyte, such as those produced by Ballard Fuel Cells, used in hydrogen fuel cell powered cars, are humidified at 80 °C even higher temperature, at both anode, the H_2 fuel side, and cathode, the air side. The overall energy efficiency is still very high. These oxide electrolytes such as $\text{SrZr}_{0.8}\text{Y}_{0.2}\text{O}_{3-\delta}$ are of particularly suitable in fuel cells using aqueous solution as the fuels such as ammonia solution, methanol solution and ethanol solution as the fuel for direct ammonia, methanol and ethanol fuel cells. The oxide electrolyte will be hydrated by the aqueous fuels at the anode to ensure the high ionic conductivity.

The best application of these oxide ionic conductors is electrolyzers for green hydrogen production when both sides of the oxide electrolyte are exposed to liquid water to maximise the ionic conductivity.

We have added some text in the revised version to balance the negative points of these materials.

6). In their test, still precious metal Electrocatalysis are used. IN principle, by use OH⁻-conducting electrolyte, non precious metals can be used as the electrocatalysts, like in normal alkaline fuel cells, why the authors did not try other cost-effective catalysts?

The focus of this study is to report the mixed OH⁻/H⁺ ionic conduction in zirconate perovskite oxides. The fuel cell study is to demonstrate the feasibility of the application of these oxide electrolytes. For the hydrogen/air fuel cell, we deliberately choose the Pt/C catalysts as both anode and cathode in order to minimise the overpotential, achieving the highest open circuit voltage to be used to compare with the theoretical value in order to demonstrate the high ionic transfer number.

For the direct ammonia fuel cells, we also use the best known electrode materials to avoid any negative effects from the electrode. Actually, we have identified a good low-cost perovskite oxide cathode, LaCr_{0.25}Fe_{0.25}Co_{0.5}O₃₋₅, with comparable performance to the Pt/C cathode, suitable to be used as cathode for alkaline membrane fuel cells. This has been cited in the revised version, ref. 46.

We will try new electrode for fuel cells in future works.

7) The power out is not attractive in current study. As claimed in their discussion, the low power output is related to the thick electrolyte, but I am afraid due to the high water vapor concentration, the electrode reaction kinetics will be greatly retarded, I suggest the authors to conduct three different electrolyte thickness to find out the impact on the power output.

The mechanical strength of the thin SrZr_{0.8}Y_{0.2}O_{3-δ} pellet is not good thus we cannot use very thin electrolyte for the fuel cell measurements. However, this is not an issue. We have found a composition with excellent mechanical strength through doping strategy.

8) Up to hundred hours of fuel cell stability is also necessary to confirm the feasibility of the new concept. We should be careful that the conversion of Sr to form Sr(OH)₂ under high humidity could also bring OH⁻ conductivity in the material, however, this OH⁻ could be easily lost, such long-term stability test is a necessarily.

The conductivity of SrZr_{0.8}Y_{0.2}O₃₋₅ in humidified air is stable during the measured 130 hour (Fig. 4d) already demonstrate the stability of the ionic conductivity. If the

conductivity is mainly due to the formed $\text{Sr}(\text{OH})_2$, as you have stated, it cannot last for such a long time because $\text{Sr}(\text{OH})_2$ will inevitably react with CO_2 in air to form insulator SrCO_3 .

The performance stability of a direct ammonia fuel cell using pure ammonia solution as the fuel without adding KOH is shown in Supplementary Fig. 21c. During the measured 20 hours, the performance is relatively stable.

It should be noted that the degradation of fuel cell performance may be caused by several different reasons while degradation of the electrolyte is only one of the possible reasons. From this point of view, measuring the conductivity stability of the oxide in air is the best method to demonstrate the stability of the electrolyte (Fig. 4d).

Reviewer #3 (Remarks to the Author):

Authors of this manuscript claimed the perovskite oxide $\text{SrZr}_{0.8}\text{Y}_{0.2}\text{O}_{3-6}$ as a high ionic conductor (0.01 S/cm at 90°C in water and wet air) for near ambient temperature solid oxide fuel cells (NAT-SOFCs). However, this conductivity is 20 times lower than that (0.2 S/cm) Nafion film. Furthermore, this measured ion conductivity is also questionable. When $\text{SrZr}_{0.8}\text{Y}_{0.2}\text{O}_{3-6}$ was in water, some ions can be dissolved in water and thus the dissolved ions (instead of H^+/OH^- ions) would be main contributors for the measured ionic conductivity. This was consistent with its negligible cell performance (0.34 mW/cm²), which is 2000 times lower than that (> 1000 mW/cm²) of proton exchange membrane fuel cells (PEMFCs). Therefore, the manuscript is not only lack of novelty, but also has serious principle issues. It should be rejected.

Thanks for the comments.

As for the conductivity, 0.01 S cm⁻¹ for oxide ionic conductivity is high enough, the answer is, it is sufficient to be used in real electrochemical devices such as fuel cells and electrolyzers. Please see the plot in a paper, Nature 2001 below, which defined the required ionic conductivity for solid oxide fuel cells is 0.01 S cm⁻¹. Of course, the thin film technology is required to minimise the resistance from the electrolyte.

Brian C. H. Steele & Angelika Heinzl, Materials for fuel-cell technologies,
Nature volume 414, pages345–352 (2001)

ionic conductivity of YSZ attains this target value around 700°C , and for $\text{Ce}_{0.8}\text{Ca}_{0.2}\text{O}_{3-x}$ (CCO) the relevant temperature is 500°C . The use of thinner electrolyte films would allow the operating temperature to be lowered. But at present it seems that the minimum thickness for dense impermeable films that can be reliably mass produced using relatively cheap ceramic fabrication routes is around $10\text{--}15 \mu\text{m}$. The use of a thick-film electrolyte requires this component to be supported on an appropriate substrate. As the substrate is the principal structural component in these cells, it is necessary to optimize the conflicting requirements of mechanical strength and gaseous permeability.

An IT-SOFC configuration that seeks to retain the specific advantages of both the tubular and planar arrangements is being developed by Rolls-Royce³⁸. This integrated planar-stack concept incorporates multi-cell assemblies connected in series and supported by a ceramic substrate, and has many similar features to the original Westinghouse tubular design³⁹.

Most development work on planar IT-SOFC systems has involved thick-film YSZ electrolytes, and so far most groups have used anode (Ni-YSZ) substrates, which allow the electrolyte powder to be sintered to a dense film around $1,400^\circ\text{C}$. One of the problems associated with using porous, composite Ni-YSZ substrates is their relatively poor thermal-expansion compatibility with the YSZ thick film. Accordingly, several groups are examining porous substrates based on Ni-Al₂O₃ or Ni-TiO₂ compositions, with thin interfacial anodic regions incorporating Ni, YSZ and/or doped CeO₂. Although replacement of the YSZ can provide better thermal-expansion compatibility, problems still remain over the volume changes associated

Although the conductivity is lower than that for the famous Nafion membrane, it is enough to be used for solid oxide fuel cells (SOFCs) and solid oxide electrolytic cells (SOECs). Please note, the operating is decreased from $500\text{--}800^\circ\text{C}$ to 90°C with the application of hydrated zirconate perovskite electrolytes Westinghouse reported here.

A key point, the Nafion membrane is an acidic membrane, thus expensive precious metal catalysts such as Pt or Ir must be used in the electrode for proton exchange membrane fuel cells (PEMFCs) and proton exchange membrane electrolyser (PEM electrolyser). This makes the cost of PEMFCs and PEM electrolyser very high, not affordable.

On the contrary, $\text{SrZr}_{0.8}\text{Y}_{0.2}\text{O}_{3-5}$ is a kind of alkaline electrolyte, thus theoretically cheap electrode materials can be used to reduce the overall cost, as stated by reviewer 2.

Conventional polymeric alkaline membrane electrolyte is not chemically compatible with CO₂ thus air cannot be used at the cathode side of a fuel cell. However, our oxide mixed OH⁻/H⁺ ionic conductor exhibits good stability in humidified air (Fig. 4d) indicating that the conductivity is not affected by CO₂ in air, good chemical compatibility. This means 'free air' can be direct used as oxidant in a fuel cell, solving the key problem of conventional polymeric alkaline membrane fuel cells based on organic membrane and conventional alkaline fuel cells based on KOH electrolytes.

As for possible dissolution of $\text{SrZr}_{0.8}\text{Y}_{0.2}\text{O}_{3-5}$ in water, there is a slow dissolution, reflected in the increased conductivity and pH value (new Supplementary Table 4). However, as the conductivity of the oxide is much higher than the measured conductivity of the used water after the conductivity measurement, it can be deduced that the measured ionic conductivity of the hydrated oxides may contain a small contribution from the slowly

dissolved oxides, but the majority of the ionic conductivity is from the hydrated oxide itself.

The focus of this study is the mixed OH^-/H^+ conduction in hydrated zirconate perovskite oxides. The fuel cell measurement is to demonstrate the feasibility of the application of these ionic conductors to be used as electrolytes. Although the hydrogen/air fuel cell performance is not ideal due to not-optimised solid electrolyte/electrode interfaces, the high OCV of the H_2 /air fuel cell demonstrated the electrolyte is mainly an ionic conductor, suitable to be used as electrolyte for electrochemical devices. The performance of the direct ammonia/air fuel cell is comparable to the reported values, which has been clarified in the revised version.

REVIEWER COMMENTS

Reviewer #2 (Remarks to the Author):

In this manuscript, the authors claimed that OH⁻ conductivity is presents in the popular protonic conducting perovskite , I believe, it is understandable and the result is reliable. The well accepted mechanism for proton transfer in perovskite is the hydration of oxygen vacancies via the reaction $\text{VO}_{\text{oo}} + \text{OO}_{\text{x}} + \text{H}_2\text{O} = 2\text{OH}_{\cdot}$, the proton can hop from one oxygen to another oxygen, leading to proton conductivity. Actually, the hopping of OH⁻ from one site to another oxygen vacancy site, is also energetically favorable, as demonstrated by DFT calculation, from the the group, and also the authors' new results. Although $1 \times 10^{-2} \text{ S cm}^{-1}$ is still not comparable to the $10^{-1} \text{ S cm}^{-1}$ level for some anion exchange membranes, however, as a pioneer work, this is still a big breakthrough considering the high intrinsic stability of inorganic membrane for OH⁻ conduction. I am satisfied with the new improvement in the revised manuscript. I would suggest the publication of this interesting work in Nature communication. The impact of current finding is not only in the field of fuel cells, but also in the field of hydrogen generation from water splitting. Some additional comments are listed below for further improvement if they think are needed.

- 1) The density of the membrane is still too low under such circumstance,, the grain boundary diffusion via the vehicle mechanism may play an important role, the authors may discuss about it in the revised manuscript.
- 2) Since the membrane is mixed H⁺ and OH⁻ conductivity, it means the membrane can also permeate water vapor, thus it can act as a water permeation membrane, the authors may also test test if the membrane is really water permeable.
- 3) If possible, oxygen isotope may provide interesting information about the OH⁻ conduction inside the membrane.
- 4) electrocatalyst is also critical for the cell performance, so the authors should discuss in the manuscript about how the good catalysts should be and give some advices for the catalysts development.

Reviewer #4 (Remarks to the Author):

Start of the reviewer's report.

In this work, the authors have detected high electrical conductivity 0.01 S cm^{-1} at 90°C for ceramic materials in water. The NAT-SOFCs test is fine, although the power density is not very high.

However, it seems that this is not “the fast ceramic mixed OH-/H+ ionic conductor”, but “electrical conduction of the MIXTURE OF CERAMIC

MATERIAL AND WATER”. In fact, the conductivity without water is quite low. The conductivity should not be measured in water (strictly speaking using the mixture of water and sample with electrodes) like in Supplementary Figure 5. Can you measure the conductivity using the system like in Supplementary Figure 8? If bulk diffusion occurs in the ceramic material, the structure and migration mechanism in the bulk system should be clarified. Neutron-diffraction analysis should be performed to show the presence of hydrogen and additional oxygen atoms in the bulk perovskite after the conductivity measurements and to show no hydrogen and additional oxygen atoms before the measurements. Tracer diffusion coefficients $D(\text{OH}^-)$ and $D(\text{H}^+)$ should be measured and compared with $D(\sigma)$ data estimated using the conductivity σ data and with $D(\text{O})$, $D(\text{OH}^-)$ and $D(\text{H}^+)$ calculated by AIMD, which can give conclusive evidences for the bulk OH⁻/H⁺ diffusion.

In particular, the use of H and/or O tracer can identify the mobile species. MSD from AIMD should show the long-range migration, which is not shown in the present MS and SI.

The authors claimed that “The migration of OH⁻ ions relies on oxygen vacancies and the temporary formation of hydrogen bonds. This is consistent with the low conductivity of SrZr_{0.8}Y_{0.2}O₃₋₆ when the temperature is above 100 °C at ambient pressure. This is because in the absence of liquid water, it is hard to facilitate the hydrogen bond formation to diffuse OH⁻ ions.” In conventional ceramic proton conductors, it is known that the OH⁻ ions can exist at several hundred degree in Celsius to facilitate the H⁺ diffusion. The temporary formation of hydrogen bond and its disappearance above 100 oC in SrZr_{0.8}Y_{0.2}O₃₋₆ are not clear and should be supported by direct experimental evidences such as NMR, Raman and IR.

TG-MS could be useful to quantify the OH/H species. What are the water content x in SrZr_{0.8}Y_{0.2}O₃₋₆xH₂O and its temperature dependence? The authors should quantify the carrier concentration.

End of the reviewer’s report.

Manuscript ID: NCOMMS-22-36610A-Z

Title: A fast ceramic mixed OH⁻/H⁺ ionic conductor for low temperature fuel cells

to ***Nature Communications***.

Dear Reviewers,

First of all, we sincerely thank both reviewers for their very positive and constructive comments, which help to further improve the quality of this paper. The responses to reviewers' comments below are highlighted in blue.

REVIEWER COMMENTS

Reviewer #2 (Remarks to the Author):

In this manuscript, the authors claimed that OH⁻ conductivity is presents in the popular protonic conducting perovskite , I believe, it is understandable and the result is reliable. The well accepted mechanism for proton transfer in perovskite is the hydration of oxygen vacancies via the reaction $\text{VO}_{\text{oo}} + \text{OOx} + \text{H}_2\text{O} = 2\text{OH}$., the proton can hop from one oxygen to another oxygen, leading to proton conductivity. Actually, the hopping of OH. from one site to another oxygen vacancy site, is also energetically favorable, as demonstrated by DFT calculation, from the the group, and also the authors' new results. Although $1 \times 10^{-2} \text{ S cm}^{-1}$ is still not comparable to the $10^{-1} \text{ S cm}^{-1}$ level for some anion exchange membranes, however, as a pioneer work, this is still a big breakthrough considering the high intrinsic stability of inorganic membrane for OH⁻ conduction. I am satisfied with the new improvement in the revised manuscript. I would suggest the publication of this interesting work in Nature communication. The impact of current finding is not only in the field of fuel cells, but also in the field of hydrogen generation from water splitting. Some additional comments are listed below for further improvement if they think are needed.

We sincerely thank this reviewer for the very positive comments on this work which is very encouraging. Although

an ionic conductivity of 0.01 S cm^{-1} is just enough to be used as thin film electrolyte for solid oxide fuel cells (SOFCs) and solid oxide electrolytic cells (SOECs) (Figure 4 in Brian C. H. Steele* & Angelika Heinze. Materials for fuel-cell technologies, Nature, 414 (2001) 345-352), higher ionic conductivity would be beneficial, particularly for electrolyzers to split water for green hydrogen production.

Although the intrinsic ionic conductivity of $\text{SrZr}_{0.8}\text{Y}_{0.2}\text{O}_{3-\delta}$ in pure water is around 0.01 S cm^{-1} , when it is exposed to strong alkaline solution, such as 3 M KOH, the conductivity is very actually high. As shown in Figure 7d, the impedance of direct ammonia fuel cell with 35 wt% $\text{NH}_3\cdot\text{H}_2\text{O}$ solution + 3 M KOH as the fuel, when the operating temperature is above 40°C , the series resistance of the whole fuel cell is only $1.1 \Omega \text{ cm}^2$. The corresponding ionic conductivity is about 0.11 S cm^{-1} if accounting conductive area from the active area of fuel cell, i.e., 1 cm^2 , which is far higher than 0.01 S cm^{-1} for pure $\text{SrZr}_{0.8}\text{Y}_{0.2}\text{O}_{3-\delta}$ in pure water. We did not carry out more investigation of this high ionic conductivity because the relative density of the $\text{SrZr}_{0.8}\text{Y}_{0.2}\text{O}_{3-\delta}$ was quite low which means KOH solution may diffuse into the holes of the ceramic $\text{SrZr}_{0.8}\text{Y}_{0.2}\text{O}_{3-\delta}$, forming a continuous pathway of OH^- ions, similar to the case when polymeric alkaline membrane is used as the electrolyte while the mixture of $\text{NH}_3\cdot\text{H}_2\text{O}$ and KOH solution was used as the fuels for alkaline membrane ammonia fuel cells. We also thought of measuring the conductivity of the $\text{SrZr}_{0.8}\text{Y}_{0.2}\text{O}_{3-\delta}$ pellet in concentrated KOH solution using the conductivity testing jig shown in originally Supplementary Figure 5 but it will be difficult to separate the conductivity of KOH solution around the $\text{SrZr}_{0.8}\text{Y}_{0.2}\text{O}_{3-\delta}$

o pellet from the measured total conductivity.

Reviewer 4 advises us to use the H-cell shown in originally Supplementary Figure 8 to measure the conductivity in water, which is a good idea. Using the H-cell to measure the conductivity of a dense oxide pellet, the measured conductivity will belong to the pellet itself thus the high ionic conductivity of KOH around the pellet will not be measured. Considering $\text{SrZr}_{0.8}\text{Y}_{0.2}\text{O}_{3-\delta}$ is not dense enough thus KOH may still diffuse into the holes of the pellet, we used a dense $\text{CaZr}_{0.8}\text{Y}_{0.2}\text{O}_{3-\delta}$ pellet to verify this new ionic conductivity measurement method. It was found difficult to seal the H-cell with the electrode and wires on the electrodes, we decided to use a home-made fuel cell test jig, shown in the new Supplementary Figure 29, the one used to collect the direct ammonia fuel cell data in Figure 7, to measure the ionic conductivity of $\text{SrZr}_{0.8}\text{Y}_{0.2}\text{O}_{3-\delta}$ and $\text{CaZr}_{0.8}\text{Y}_{0.2}\text{O}_{3-\delta}$. It was found the measured conductivity of $\text{SrZr}_{0.8}\text{Y}_{0.2}\text{O}_{3-\delta}$ in water using the fuel cell testing jig is comparable to those measured in water when using the jig shown originally Supplementary Figure 5, as shown in new Supplementary Figure 30b-d. Therefore the original conductivity data measured in water is still accurate. However, it was found that, the conductivity of both less dense $\text{SrZr}_{0.8}\text{Y}_{0.2}\text{O}_{3-\delta}$ and dense $\text{CaZr}_{0.8}\text{Y}_{0.2}\text{O}_{3-\delta}$ in KOH solution, is much higher than those in water. For example, at 90°C , the conductivity of dense $\text{CaZr}_{0.8}\text{Y}_{0.2}\text{O}_{3-\delta}$ in 6M KOH solution is 0.106 S cm^{-1} which is 100 times higher than that of dense $\text{CaZr}_{0.8}\text{Y}_{0.2}\text{O}_{3-\delta}$ in water, as shown in new Supplementary Figure 31. This higher ionic conductivity of 0.106 S cm^{-1} is comparable to the best acidic Nafion membrane, potentially can be used as electrolyte to replace the polymeric separating membrane of conventional alkaline or alkaline membrane electrolyzers while both sides are exposed to concentrated KOH aqueous solution. It can also be used as electrolyte for some fuel cells such as direct ammonia fuel cells.

In order to investigate the mechanism of the enhanced ionic conductivity of oxide materials when exposed in concentrated KOH solution, we have carried out SEM observation and element mapping of the dense $\text{CaZr}_{0.8}\text{Y}_{0.2}\text{O}_{3-\delta}$

o pellet after the conductivity measurement in KOH solution without washing, as shown in new Supplementary

34. It was found element K is distributed across the whole cross section of the $\text{CaZr}_{0.8}\text{Y}_{0.2}\text{O}_{3-\delta}$ pellet, indicating KOH has entered the oxide pellet, forming continuous pathways for OH^- ions, leading to much higher ionic conductivity. However, when comparing the ionic conductivity of doped $\text{CaZr}_{0.8}\text{Y}_{0.2}\text{O}_{3-\delta}$ and pure CaZrO_3 in 1M KOH aqueous solution, the ionic conductivity of doped $\text{CaZr}_{0.8}\text{Y}_{0.2}\text{O}_{3-\delta}$ is five times higher than that of pure CaZrO_3 , indicating OH^- ions can still transfer through the oxygen vacancies in the $\text{CaZr}_{0.8}\text{Y}_{0.2}\text{O}_{3-\delta}$ lattice thus the oxygen vacancies still play a crucial role to achieve high ionic conductivity (Supplementary Figure 31d).

At this stage, we assume KOH only enter the grain boundary of the oxide pellets. Whether it enter the oxide lattice, theoretically it is unlikely but we cannot determine at this stage. As this paper is focused on the mixed OH^-/H^+ ionic conduction in perovskite oxide $\text{SrZr}_{0.8}\text{Y}_{0.2}\text{O}_{3-\delta}$, due to limited time and space, we cannot answer all these questions of new phenomenon in a single paper. We aim to investigate whether KOH enter the lattice in future papers as the enhanced ionic conductivity when exposed in oxide materials have been observed in other oxide ionic conductors investigated in our group at University of Warwick. Seems this is a general phenomenon.

The high ionic conductivity of 0.1 S cm^{-1} at 90°C for $\text{CaZr}_{0.8}\text{Y}_{0.2}\text{O}_{3-\delta}$ in 6M KOH addresses the requirements of high ionic conductivity of the electrolyte materials in some applications, such as alkaline or alkaline membrane electrolysers.

1) The density of the membrane is still too low under such circumstance, the grain boundary diffusion via the vehicle mechanism may play an important role, the authors may discuss about it in the revised manuscript.

Yes, the relative density of $\text{SrZr}_{0.8}\text{Y}_{0.2}\text{O}_{3-\delta}$ is not high enough thus the measured conductivity may contains the ionic diffusion across the grain boundary via the vehicle mechanism, which has been added in the revised version. We also carried the conductivity measurements of dense $\text{CaZr}_{0.8}\text{Y}_{0.2}\text{O}_{3-\delta}$ (relative density 99%) in both water and KOH aqueous solutions. It was found the conductivity in aqueous KOH solution is much higher than that in water, indicating KOH, precisely K^+ and OH^- ions may also diffuse into the grain boundary of the oxides to form continuous pathways for ions leading to very high ionic conductivity, while the oxygen vacancies in the oxide lattice also plays an crucial role, leading to much higher ionic conductivity in alkaline solutions, as stated above. The high OH^- ionic conductivity of dense $\text{CaZr}_{0.8}\text{Y}_{0.2}\text{O}_{3-\delta}$ pellet is also demonstrated by the higher current density when 1 M KOH dissolved in D_2O was used at one side of the electrolyser (Supplementary Figures 32,33).

2) Since the membrane is mixed H^+ and OH^- conductivity, it means the membrane can also permeate water vapor, thus it can act as a water permeation membrane, the authors may also test test if the membrane is really water permeable.

Thanks for the advice. We have carried out the experiments on electrolyser when dense $\text{CaZr}_{0.8}\text{Y}_{0.2}\text{O}_{3-\delta}$ was used as the electrolyte. We used D_2O in HER catalytic side and H_2O in OER catalytic side. We used FTIR to trace the crossover of D in electrolysis cell. When voltage was not applied on the cell, the D signal did not appear in the other side, indicating the D_2O cannot automatically diffuse cross the dense $\text{CaZr}_{0.8}\text{Y}_{0.2}\text{O}_{3-\delta}$ (blue line in new Supplementary

Figure 28b), as expected. However, in the electrolysis process at 2 V, D₂O and OD⁻ ions were observed on other side of the cell demonstrating the OH⁻ and OD⁻ ionic conduction of the CaZr_{0.8}Y_{0.2}O_{3-δ}. Due to the low relative density of SrZr_{0.8}Y_{0.2}O_{3-δ}, D₂O may also direct diffuse across the pellet through the holes thus we choose to use the dense CaZr_{0.8}Y_{0.2}O_{3-δ} pellet (relative density 99%) as the electrolyte for the electrolysis experiments.

3) If possible, oxygen isotope may provide interesting information about the OH⁻ conduction inside the membrane.

We have checked that the price for ¹⁸O which is too high to be affordable. However, the experiments on electrolysis of pure D₂O and 1M KOH dissolved in D₂O using dense CaZr_{0.8}Y_{0.2}O_{3-δ} as the electrolyte indicates CaZr_{0.8}Y_{0.2}O_{3-δ} is an OH⁻ ionic conductor (new Supplementary Figures 32, 33). From solid state NMR experiments, protons and/or proton containing species such as OH⁻ ions are also found in CaZr_{0.8}Y_{0.2}O_{3-δ} and protons are in the vicinity of Y³⁺ ions (new Supplementary Figure 18). We presume SrZr_{0.8}Y_{0.2}O_{3-δ} and CaZr_{0.8}Y_{0.2}O_{3-δ} exhibit similar ionic conduction mechanism.

4) electrocatalyst is also critical for the cell performance, so the authors should discuss in the manuscript about how the good catalysts should be and give some advices for the catalysts development.

Perovskite oxides have been widely investigated as efficient electrocatalysts for different electrochemical reactions. These perovskite oxide electrocatalysts may have good chemical and thermal compatibility with the perovskite oxide electrolyte reported in this paper, can be combined with perovskite oxides to develop efficient electrochemical devices. As the focus of this paper is on the ionic conducting electrolyte, it is better not to diversify the topic to electro-catalysts. Therefore we just added the following short phrase in the discussion part:

‘Perovskite oxides have been reported as excellent electrolysts for both high temperature solid oxide fuel cells and low temperature electrochemical devices such as fuel cells and electrolyzers^{2,51,52,61,62}. If some multi-valence transition elements are introduced into the B-sites of SrZr_{0.8}Y_{0.2}O_{3-δ}, mixed ionic-electronic conductors may be formed which may be potential electro-catalysts for electrochemical devices.’

Reviewer #4 (Remarks to the Author):

Start of the reviewer’s report.

In this work, the authors have detected high electrical conductivity 0.01 S cm⁻¹ at 90oC for ceramic materials in water. The NAT-SOFCs test is fine, although the power density is not very high.

However, it seems that this is not “the fast ceramic mixed OH⁻/H⁺ ionic conductor”, but “electrical conduction of the MIXTURE OF CERAMIC

MATERIAL AND WATER”. In fact, the conductivity without water is quite low. The conductivity should not be measured in water (strictly speaking using the mixture of water and sample with electrodes) like in Supplementary

Figure 5. Can you measure the conductivity using the system like in Supplementary Figure 8? If bulk diffusion occurs in the ceramic material, the structure and migration mechanism in the bulk system should be clarified. Neutron-diffraction analysis should be performed to show the presence of hydrogen and additional oxygen atoms in the bulk perovskite after the conductivity measurements and to show no hydrogen and additional oxygen atoms before the measurements. Tracer diffusion coefficients $D(\text{OH}^-)$ and $D(\text{H}^+)$ should be measured and compared with $D(\sigma)$ data estimated using the conductivity σ data and with $D(\text{O})$, $D(\text{OH}^-)$ and $D(\text{H}^+)$ calculated by AIMD, which can give conclusive evidences for the bulk OH^-/H^+ diffusion.

In particular, the use of H and/or O tracer can identify the mobile species. MSD from AIMD should show the long-range migration, which is not shown in the present MS and SI.

Sincerely thank this reviewer for the positive comments. The conductivity of pure water itself is very low. However, we cannot rule out the possibility that protons from the dissociated H_2O may combine with water to form H_3O^+ ions. The protons may diffuse through water molecule, so called vehicles mechanism, the same as Nafion membrane.

(Note, This is newly added equation 3 in the revised version)

The OH^- ions may transfer through the perovskite lattice to form OH^- ions conduction. We cannot rule out OH^- ions may also diffuse through the grain boundary. However, the obvious difference in ionic conductivity of $\text{BaZr}_{0.8}\text{Y}_{0.2}\text{O}_{3-\delta}$, $\text{SrZr}_{0.8}\text{Y}_{0.2}\text{O}_{3-\delta}$ and $\text{CaZr}_{0.8}\text{Y}_{0.2}\text{O}_{3-\delta}$ indicate, in water, majority of the ions have to diffuse through the bulk materials. If through the grain boundary only, the measured ionic conductivity of $\text{BaZr}_{0.8}\text{Y}_{0.2}\text{O}_{3-\delta}$, $\text{SrZr}_{0.8}\text{Y}_{0.2}\text{O}_{3-\delta}$ and $\text{CaZr}_{0.8}\text{Y}_{0.2}\text{O}_{3-\delta}$ should be at a similar level because their grain boundary is similar, which is inconsistent with the measured conductivity in water, i.e., $\text{SrZr}_{0.8}\text{Y}_{0.2}\text{O}_{3-\delta}$ exhibits much higher ionic conductivity than those for $\text{BaZr}_{0.8}\text{Y}_{0.2}\text{O}_{3-\delta}$ and $\text{CaZr}_{0.8}\text{Y}_{0.2}\text{O}_{3-\delta}$.

In 1M and 6M KOH solution, the ionic conductivity of doped $\text{CaZr}_{0.8}\text{Y}_{0.2}\text{O}_{3-\delta}$ is much higher than that for undoped CaZrO_3 (Supplementary Figure 31d), indicating oxygen vacancies in $\text{CaZr}_{0.8}\text{Y}_{0.2}\text{O}_{3-\delta}$ lattice greatly facilitates the transfer of ions, leading to higher ionic conductivity.

As the high ionic conductivity is associated to liquid water, it is important to measure the conductivity in water. Although this is the first report to measure the conductivity of a ceramic material in water, for the alkaline membrane community, all the ionic conductivity for polymeric alkaline membranes are measured in water.

Thank for the advice on using the H-cell type jig shown in supplementary Figure 8 to measure the conductivity in water. We have adopted this method. As shown in new Supplementary Figure 29, the measured conductivity using the fuel cell test jig in new Supplementary Figure 29, similar to the H-cell jig, the measured conductivity is comparable to the values measured in water using the jig shown in supplementary Figure 5. Therefore the original reported conductivity measured in water using the jig shown in supplementary Figure 5 is still reliable.

This reviewer advises to use neutron diffraction to confirm the presence of protons in the oxide lattice. However, the UK neutron diffraction facility, ISIS, the next call for beam time was open on 1st March 2023, decision will be made in July 2023. Thus it is difficult to carry out neutron diffraction for these oxides in time. However, there is another technology, which is equally good enough to detect hydrogen and hydrogen-containing species, such as OH⁻ ions in oxide lattice.

Besides neutron diffraction, solid state ¹H NMR can also detect protons in oxide materials. The solid state ¹H NMR can observe hydrogen or hydrogen related species such as OH⁻ ions only. There will be no ¹H NMR signal if no hydrogen or related species are present in the oxides. In our experiments, there are clear signals for protons in solid state ¹H NMR measurements indicating not only that there are protons in our materials but also cross polarization for ¹H to ⁸⁹Y has been detected in both SrZr_{0.8}Y_{0.2}O_{3-δ} and CaZr_{0.8}Y_{0.2}O_{3-δ} samples meaning that ¹H are in proximity to ⁸⁹Y atoms (Figure 5 and new Supplementary Figure 15, 16, 18).

To confirm the OH⁻ ionic conduction, we have carried out experiments on electrolysis of D₂O using dense CaZr_{0.8}Y_{0.2}O_{3-δ} pellet as the electrolyte. We used D₂O in HER catalytic side and H₂O in OER catalytic side. We used FTIR to trace the crossover of isotope D in electrolysis cell. OD⁻ ions were detected on other side of the cell only after electrolysis process indicating it exhibits OH⁻ ionic conduction. Details shown in new Supplementary Figures 32, 33 and new Supplementary Note 3. As the relative density of SrZr_{0.8}Y_{0.2}O_{3-δ} is not very high, we choose to use dense CaZr_{0.8}Y_{0.2}O_{3-δ} pellet (relative density 99%) for the electrolysis experiments. However, from solid state NMR, there are two types of hydrogen and hydrogen-containing species in both SrZr_{0.8}Y_{0.2}O_{3-δ} and CaZr_{0.8}Y_{0.2}O_{3-δ}. It is reasonably deduced that SrZr_{0.8}Y_{0.2}O_{3-δ} may also exhibit H⁺ ionic conduction, which has been confirmed by the concentration cell measurements (Supplementary Table 3).

The authors claimed that “The migration of OH⁻ ions relies on oxygen vacancies and the temporary formation of hydrogen bonds. This is consistent with the low conductivity of SrZr_{0.8}Y_{0.2}O_{3-δ} when the temperature is above 100 °C at ambient pressure. This is because in the absence of liquid water, it is hard to facilitate the hydrogen bond formation to diffuse OH⁻ ions.” In conventional ceramic proton conductors, it is known that the OH⁻ ions can exist at several hundred degree in Celsius to facilitate the H⁺ diffusion. The temporary formation of hydrogen bond and its disappearance above 100 oC in SrZr_{0.8}Y_{0.2}O_{3-δ} are not clear and should be supported by direct experimental evidences such as NMR, Raman and IR.

TG-MS could be useful to quantify the OH/H species. What are the water content x in SrZr_{0.8}Y_{0.2}O_{3-δ}xH₂O and its temperature dependence? The authors should quantify the carrier concentration.

The conclusion that the high ionic conductivity of the oxides relies on liquid water is based on the observed sharp decrease in conductivity when the temperature is above 100°C at ambient pressure. The existence of some OH⁻ species at high temperature proton-conducting oxides have been confirmed by both STA (supplementary Figure 19), high temperature Raman (supplementary Figures 20 & 21) and high temperature solid state NMR measurements (supplementary Figure 16). However, this did not lead to high OH⁻ ionic conductivity of these oxides at high temperatures (above 100 °C). Assuming the diffusion coefficient of OH⁻ ions are very high at high temperature, then the possible reason of the low OH⁻ ionic conductivity is the very low concentration of OH⁻ species in the absent of liquid water and/or the lack of key hydrogen bonds to facilitate the OH⁻ ion mobility. High temperature Raman

measurements indicates the OH⁻ related signal almost disappears at 300 °C for both SrZr_{0.8}Y_{0.2}O_{3-o} and CaZr_{0.8}Y_{0.2}O_{3-o} (Supplementary Figures 20,21).

‘The peaks at above 3600 cm⁻¹ in Supplementary Figs 20b, 21b are assigned to the hydroxyl (OH⁻) groups formed in the perovskite oxide upon hydration⁴⁷⁻⁴⁹. These peaks almost disappear at above 300 °C indicating the departure of OH⁻.’

An important finding is, we measured the STA of the partially hydrated SrZr_{0.8}Y_{0.2}O_{3-o} and CaZr_{0.8}Y_{0.2}O_{3-o}. As shown in new Supplementary Figure 19. For SrZr_{0.8}Y_{0.2}O_{3-o}, the weight kept losing between room temperature and 850°C indicating the hydrated water can be held in the lattice for a much higher temperature. It is assumed the weight loss above 100°C is due to the loss of hydrated water x in SrZr_{0.8}Y_{0.2}O_{3-o}·xH₂O, or SrZr_{0.8}Y_{0.2}O_{2.95}·xH₂O, then the estimated x value is 0.575. While in CaZr_{0.8}Y_{0.2}O_{2.95}·xH₂O, x is estimated as only 0.065. The fact is, the ionic conductivity of SrZr_{0.8}Y_{0.2}O_{3-o} in water is much higher than that of CaZr_{0.8}Y_{0.2}O_{3-o}. This indicates the high ionic conductivity of SrZr_{0.8}Y_{0.2}O_{3-o} in water is closely related to the high amounts of hydrated water.

Unfortunately the TG-MS in Warwick, Mass Spectrometry is under repairing, not accessible at the moment. *In situ* FTIR, the sample stage in FTIR can only be heated up to 60 °C, not suitable for this work. High temperature NMR in Warwick, the probe for measuring 1H MAS NMR can be heated up to 250 °C. However, huge background noise and peak shift were observed when heated up to 250 °C, thus we only measured to 125 °C. For high temperature Raman, black body radiation of powder sample and detector saturation at high temperatures causes low quality of signals.

We would be very grateful to you if this important paper could be published in the prestigious Nature Communications.

REVIEWER COMMENTS

Reviewer #2 (Remarks to the Author):

From my side, I am happy with the new improvement of the revised manuscript, I would suggest the acceptance of this manuscript. It is better the authors give some discussion about the challenges of such mixed OH⁻/H⁺ conducting fuel cells in practical applications in the end of the manuscript.

Reviewer #4 (Remarks to the Author):

The authors have revised the MS to some extent. I have some comments.

The authors claimed that "If through the grain boundary only, the measured ionic conductivity of BaZr_{0.8}Y_{0.2}O₃₋₆, SrZr_{0.8}Y_{0.2}O₃₋₆ and CaZr_{0.8}Y_{0.2}O₃₋₆ should be at a similar level because their grain boundary is similar, which is inconsistent with the measured conductivity in water, i.e., SrZr_{0.8}Y_{0.2}O₃₋₆ exhibits much higher ionic conductivity than those for BaZr_{0.8}Y_{0.2}O₃₋₆ and CaZr_{0.8}Y_{0.2}O₃₋₆."

This discussion is speculative. Can you examine the grain-boundary and bulk conductivities by equivalent circuit analysis of impedance data? Then, you might show clearly the grain-boundary conductivities of these materials.

The authors have detected OD by FTIR, which suggests the OD conduction. It is fine. Can you measure also using D₂(180) or H₂(180)? Only D moves or D(180) moves? It would be useful to indicate the OH (OD) diffusion.

Is it possible to estimate the transport numbers of OH⁻, O₂⁻, electron and H⁺, respectively?

If neutron is available, the presence of bulk OH and/or OD should be shown. The authors can analyse the oxygen occupancy using quality XRD data, which would be evidence for additional oxygen atoms due to the hydration in bulk state. But, the quality of XRD data in Supplementary Fig. 1 seems not so good. Impurity or non-identified small peaks exist and the 2θ range is narrow. The O occupancy seems inconsistent with the water content in the sample. This should be discussed and validated using the TG data.

Additionally, the reflection index is not (hkl) but hkl in Supplementary Fig. 1a. In Supplementary Table 1, not P bnm but Pbnm and this should be italic. x, y, and z should also be italic. The U in Uiso should be italic.

Manuscript ID: NCOMMS-22-36610C

Title: A fast ceramic mixed OH⁻/H⁺ ionic conductor for low temperature fuel cells

to ***Nature Communications***.

Dear Reviewers,

First of all, we sincerely thank both reviewers for their very positive and constructive comments, which help to further improve the quality of this paper. The responses to reviewers' comments below are highlighted in blue. The quality of the paper has been further improved. Hopefully it can be accepted for publication in Nature Communications.

REVIEWER COMMENTS

Reviewer #2 (Remarks to the Author):

From my side, I am happy with the new improvement of the revised manuscript, I would suggest the acceptance of this manuscript. It is better the authors give some discussion about the challenges of such mixed OH⁻/H⁺ conducting fuel cells in practical applications in the end of the manuscript.

Thanks for the very positive comments. To identify a good ceramic OH⁻/H⁺ ionic conductor which can work at near ambient temperature is just a first step. For practical applications in electrochemical devices, all the problems we have in high temperature solid oxide electrochemical cells, we may potentially face them in the low temperature SOFCs, SOECs, such as identification of matched electrode materials, the electrolyte/electrode interface. It is still a long way to go. At the end of the discussion part, we have added the following comment.

‘For practical application of these new oxide ionic conducting materials in electrochemical devices such as fuel cells and electrolyzers, the matched electrode materials, the electrode/electrolyte interface etc. must be good which need further investigation.’

Reviewer #4 (Remarks to the Author):

The authors have revised the MS to some extent. I have some comments.

Thanks this reviewer for the positive comments.

The authors claimed that “If through the grain boundary only, the measured ionic conductivity of $\text{BaZr}_{0.8}\text{Y}_{0.2}\text{O}_{3-6}$, $\text{SrZr}_{0.8}\text{Y}_{0.2}\text{O}_{3-6}$ and $\text{CaZr}_{0.8}\text{Y}_{0.2}\text{O}_{3-6}$ should be at a similar level because their grain boundary is similar, which is inconsistent with the measured conductivity in water, i.e., $\text{SrZr}_{0.8}\text{Y}_{0.2}\text{O}_{3-6}$ exhibits much higher ionic conductivity than those for $\text{BaZr}_{0.8}\text{Y}_{0.2}\text{O}_{3-6}$ and $\text{CaZr}_{0.8}\text{Y}_{0.2}\text{O}_{3-6}$.”

This discussion is speculative. Can you examine the grain-boundary and bulk conductivities by equivalent circuit analysis of impedance data? Then, you might show clearly the grain-boundary conductivities of these materials.

We fully agree that this discussion is speculative that is why we only put it in the response to the reviewer’s comments, did not put in the article, neither the main article nor the supplementary information. Most of the impedance of these oxide sample, we can only see the total resistance because the grain boundary response is quite small. Therefore it is difficult to get good conclusion from impedance spectra. As this paper is already very long, combining two sets of data for pure and Y-doped SrZrO_3 and pure and doped CaZrO_3 , it is hard to further expand it. Therefore we think it is better leave the question for future investigation.

In supplementary Figure 31d, the ionic conductivity of pure CaZrO_3 is about 20% of that for $\text{CaZr}_{0.8}\text{Y}_{0.2}\text{O}_{3-6}$ in 1M and 6M KOH solution indicate the oxygen vacancies in the perovskite lattice, in the grain or bulk, greatly facilitate the transfer of OH^- ions in the perovskite oxides.

The authors have detected OD by FTIR, which suggests the OD conduction. It is fine. Can you measure also using $\text{D}_2(180)$ or $\text{H}_2(180)$? Only D moves or $\text{D}(180)$ moves? It would be useful to indicate the OH^- (OD) diffusion.

We have bought 50g ^{18}O water (cost £2800) from CK Isotopes Ltd for this experiment. The electrolysis of ^{18}O water using dense $\text{CaZr}_{0.8}\text{Y}_{0.2}\text{O}_{3-6}$ as the electrolyte. Raman experiments indicates the transfer of ^{18}O through a dense $\text{CaZr}_{0.8}\text{Y}_{0.2}\text{O}_{3-6}$ pellet a DC voltage was applied. Combining with the previous experiments on electrolysis of D_2O , further confirm the OH^- ionic conduction of perovskite oxides. The data are shown in added Supplementary Figure

32.

Is it possible to estimate the transport numbers of OH⁻, O²⁻, electron and H⁺, respectively?

The ion transport numbers of anions (OH⁻ ions) and cations have been estimated by concentration cell measurements. The data are shown in Supplementary Table 3. Although the measured cation is Na⁺ ions, we believe in the oxides, it is likely to be H⁺ ions. As for the transport number of electrons in the oxide materials, these has been figured out by comparison of ac and dc conductivity, shown as Y2 axes with title 'ion transfer number ti' in the conductivity plots, such as Figure 4a, 4b. The electron transfer number equals to 1 – ti.

If neutron is available, the presence of bulk OH and/or OD should be shown. The authors can analyse the oxygen occupancy using quality XRD data, which would be evidence for additional oxygen atoms due to the hydration in bulk state. But, the quality of XRD data in Supplementary Fig. 1 seems not so good. Impurity or non-identified small peaks exist and the 2theta range is narrow. The O occupancy seems inconsistent with the water content in the sample. This should be discussed and validated using the TG data.

Although neutron is a powerful technique for materials characterization, it is difficult to access.

At ISIS, there was a call on TS2 instruments before July 2023 which do not have neutron

diffraction. The application of neutron diffraction in TS1 will be open for application in

September 2023. <https://www.isis.stfc.ac.uk/Pages/Apply-for-beamtime.aspx>

We will apply for beam time but there is guarantee that we can get it. Solid state 1H NMR and high temperature Raman already confirm the existence of OH⁻ species in the hydrated oxide samples.

We have carried out new XRD data with scanning range of two theta 5 – 140 degree, the maximum range we can have. Some samples, we collect the XRD data for 15 hours. The patterns are more or less the same as the previous ones. Only difference is sample BaZr_{0.8}Y_{0.2}O_{3-δ}, 1% of Y₂O₃ was found. There are no extra peaks for the second phase at two theta above 90°.

We agree it is difficult to determine the oxygen vacancies by XRD. Fortunately, for sample SrZr_{0.8}Y_{0.2}O_{3-δ} fired in air, before hydration, the oxygen vacancy happens to be able to be refined. It show the oxygen vacancies are mainly on the O₂ sites, the estimated δ in BaZr_{0.8}Y_{0.2}O_{3-δ} is 0.07, more precisely BaZr_{0.86}Y_{0.14}O_{2.93}.

The water content in the hydrated BaZr_{0.86}Y_{0.14}O_{2.93} was measured by STA. The water measured by STA is the

total water include three parts, the physically absorbed water; the water molecules integrated with oxygen vacancies to form OH- species which has been observed by both solid state ¹H NMR and high temperature Raman; the hydrated water, which is z in BaZr_{0.86}Y_{0.14}O_{2.93}·zH₂O. The quantity of hydrated water, the z value could be very different from different oxides. Not all the measured water by STA are from those integrated with the oxygen vacancies. We have added some text in 'TG and Raman spectra analysis' part to make this clearer.

Additionally, the reflection index is not (hkl) but hkl in Supplementary Fig. 1a. In Supplementary Table 1, not P bnm but Pbnm and this should be italic. x, y, and z should also be italic. The U in Uiso should be italic.

Thanks for the advice and the errors have been corrected.

REVIEWER COMMENTS

Reviewer #4 (Remarks to the Author):

The authors revised the paper well. I have a few comments.

A major problem of this work is that the authors have not extracted the bulk and grain-boundary conductivities using the impedance data and equivalent circuit analyses. Although I have requested this in the previous comments, it seems that the authors were not able to perform well the analyses. At lower temperatures (e.g., 250 K), the extraction may be possible. I think that this should be carried out before publication. The authors can validate the data using the KK residuals.

Following my comments, the authors discussed the transport numbers of OH⁻. Can you estimate also the transport number of H⁺? You can compare the OH⁻ conductivity calculated using the transport number with the OH⁻ conductivities of other materials in the literature.

Using neutron diffraction, the presence of bulk OH and/or OD can be shown also in hydrated sample. The authors can utilize the "rapid access proposal", which I used before. If you had followed my comments and applied the suggestions made last November/December, you would have had the data by now.

Additionally, the physical quantities such as δ , t in $t\text{Na}^+$, R , T , t in $t\text{OH}^-$, x , y and z in $\text{SrZr}_{1-x}\text{Y}_x\text{O}_{3-y}(\text{OH})_2 \cdot z\text{H}_2\text{O}$ should be italic. 0 in E0 and liquid in pliquid should be non-italic.

Manuscript ID: NCOMMS-22-36610D

Title: A fast ceramic mixed OH⁻/H⁺ ionic conductor for low temperature fuel cells

to ***Nature Communications***.

Dear Reviewers,

First of all, we sincerely thank the reviewer for the very positive and constructive comments, which help to further improve the quality of this paper. The responses to reviewer's comments below are highlighted in blue. The quality of the paper has been further improved. Hopefully it can be accepted for publication in Nature Communications.

REVIEWER COMMENTS

Reviewer #4 (Remarks to the Author):

The authors revised the paper well. I have a few comments.

Thank this reviewer for the very positive comments.

A major problem of this work is that the authors have not extracted the bulk and grain-boundary conductivities using the impedance data and equivalent circuit analyses. Although I have requested this in the previous comments, it seems that the authors were not able to perform well the analyses. At lower temperatures (e.g., 250 K), the extraction may be possible. I think that this should be carried out before publication. The authors can validate the data using the KK residuals.

We fully agree that if we can examine the grain-boundary and bulk conductivities, we can analyze the ions diffusion

in bulk and grain-boundary better. However, for most of the impedance we tested at room temperatures and above, we can only see the total resistance, that's why we did not get good conclusion from previous impedance spectra. Really thank the reviewer's advice to try lower temperatures like 250 K. This time, we have purchased a freezer to achieve a measurement temperature of $-25\text{ }^{\circ}\text{C}$ and a new electrochemical workstation VIONIC which can measure the impedance from quite a high frequency 10 MHz (the previously used Solartron 1470 can only measure from 1MHz). We have remeasured the impedance for $\text{BaZr}_{0.8}\text{Y}_{0.2}\text{O}_{3-\delta}$, $\text{SrZr}_{0.8}\text{Y}_{0.2}\text{O}_{3-\delta}$ and $\text{CaZr}_{0.8}\text{Y}_{0.2}\text{O}_{3-\delta}$ samples using the new equipment, and obtained similar total conductivities results as before. All the impedance data has been analysed with equivalent electrochemical circuit and validated by Kramers-Kronig analyses, i.e, the real and imaginary residuals are lower than 1%. The example analyses are showed in new Supplementary Fig. 9. Fortunately, we observed a complete grain-boundary response in the impedance curves obtained at $-25\text{ }^{\circ}\text{C}$ (as shown in new Supplementary Fig. 9a,d,g). For the impedance curves measured between room temperature and $90\text{ }^{\circ}\text{C}$, the grain-boundary response of $\text{SrZr}_{0.8}\text{Y}_{0.2}\text{O}_{3-\delta}$ and $\text{CaZr}_{0.8}\text{Y}_{0.2}\text{O}_{3-\delta}$ become weaker and weaker when temperature increases, and we could not extract bulk and grain-boundary conductivities from equivalent circuit simulation at temperatures above $30\text{ }^{\circ}\text{C}$. For example, from Figure 1 below, it is obvious that a semicircular curve representing grain-boundary response was not successfully simulated, only total resistance (bulk + grain boundary) was obtained for $\text{SrZr}_{0.8}\text{Y}_{0.2}\text{O}_{3-\delta}$ sample measured at 40 and $90\text{ }^{\circ}\text{C}$. Nonetheless, the bulk and grain-boundary conductivities extracted at $-25\text{ }^{\circ}\text{C}$ and $25\text{ }^{\circ}\text{C}$ are sufficient to draw conclusions (as shown in new Supplementary Fig. 8c,d). At $-25\text{ }^{\circ}\text{C}$ the bulk conductivity of $\text{BaZr}_{0.8}\text{Y}_{0.2}\text{O}_{3-\delta}$ is the highest as expected due to its larger lattice volume, but when the conductivity test temperature reached room temperature and above (i.e., liquid water environment), not only the grain boundary conductivity but also the bulk conductivity of $\text{SrZr}_{0.8}\text{Y}_{0.2}\text{O}_{3-\delta}$ increases rapidly and is higher than the other two samples, both of them contributing to the highest total conductivity among the three $\text{AZr}_{0.8}\text{Y}_{0.2}\text{O}_{3-\delta}$ ($A = \text{Ca}, \text{Sr}, \text{Ba}$) oxides. This shows to a certain extent that the conductive ions (OH^-/H^+) not only diffuse in the perovskite lattice but may also transport through the grain boundaries.

Figure 1. Representative fitted impedance curve for $\text{SrZr}_{0.8}\text{Y}_{0.2}\text{O}_{3-\delta}$ at 40 and $90\text{ }^{\circ}\text{C}$. Enlarged impedance spectra is displayed in insert.

The bulk diffusion occurring in the perovskite $\text{SrZr}_{0.8}\text{Y}_{0.2}\text{O}_{3-\delta}$ has been directly verified by the neutron diffraction

analyses on the deuterated $\text{SrZr}_{0.8}\text{Y}_{0.2}\text{O}_{3-\delta}$ which shows the presence of hydrogen/ deuterium in the perovskite lattice, as stated in ‘Neutron diffraction analysis’ section, new Supplementary Fig. 25, new Supplementary Note 3 and new Supplementary Tables 6,7. For the ion diffusion occurring in the grain boundary, we have stated in ‘Conductivity measurements using fuel cell testing jig’ section in the previous version of manuscript that, the conductivity of the 1500 °C sintered pellet (with higher relative density) is slightly lower than that sintered at 1300 °C. Ions particularly protons may diffuse along the grain boundary together with water molecule via a vehicle mechanism, similar to that in Nafion membrane, causing high H^+ ionic conduction. Also, the surface protonic conductivity in wet atmosphere has been reported on porous oxides. In addition, we do not rule out the possibility that free unbonded protons and OH^- ions in the environment (water) migrate through the grain boundaries using O-H channels in the very trace amounts of $\text{Sr}(\text{OH})_2$ and $\text{Y}(\text{OH})_3$ residue located in the grain boundaries of hydrated $\text{SrZr}_{0.8}\text{Y}_{0.2}\text{O}_{3-\delta}$.

Following my comments, the authors discussed the transport numbers of OH^- . Can you estimate also the transport number of H^+ ? You can compare the OH^- conductivity calculated using the transport number with the OH^- conductivities of other materials in the literature.

We have estimated the proton transference number (t_+) by chronoamperometry in conjunction with EIS measurements (as shown in Methods and new Supplementary Fig. 11). The proton transference number of $\text{SrZr}_{0.8}\text{Y}_{0.2}\text{O}_{3-\delta}$ pellet at room temperature is ~ 0.27 which is close to the cation transport number (t_{Na^+}), 0.303, estimated by concentration cell measurements. This verifies that the cation transport number measured in the concentration cell almost represents proton conduction. It is reliable to present proton and OH^- conduction ability with the obtained transport numbers of Na^+ (t_{Na^+}) and OH^- (t_{OH^-}) from concentration cell measurements in Supplementary Table 3.

The hydrated $\text{SrZr}_{0.8}\text{Y}_{0.2}\text{O}_{3-\delta}$ is a mixed OH^-/H^+ conductor, about 70% of the measured ionic conductivity is due to the transfer of OH^- ions (new Supplementary Table 3). The OH^- conductivity of $\text{SrZr}_{0.8}\text{Y}_{0.2}\text{O}_{3-\delta}$ at room temperature calculated using the transport number of OH^- is 2.45 mS cm^{-1} . We have compared the OH^- conductivity with literature results as shown in new Supplementary Table 5. The OH^- conductivity of $\text{SrZr}_{0.8}\text{Y}_{0.2}\text{O}_{3-\delta}$ is close to that of some polymer-based alkaline membranes. The activation energy of $\text{SrZr}_{0.8}\text{Y}_{0.2}\text{O}_{3-\delta}$ in H_2O is $0.179 \pm 0.006 \text{ eV}$ which is close to the 0.17 eV for Nafion membrane in H_2O and within the range of activity energy for OH^- conducting polymers ($0.12\text{--}0.26 \text{ eV}$).

Using neutron diffraction, the presence of bulk OH and/or OD can be shown also in hydrated sample. The authors can utilize the “rapid access proposal”, which I used before. If you had followed my comments and applied the suggestions made last November/December, you would have had the data by now.

Really thank the reviewer’s advice for the “rapid access proposal”, from which we finally got the chance to collect neutron powder diffraction (NPD) data at room temperature on the time-of-flight high-resolution powder diffractometer (HRPD) at ISIS. We have two new co-authors, Dr. Dominic Fortes and Dr. Christopher M. Howard, collected the HRPD neutron diffraction data for vacuum dried and deuterated samples of $\text{SrZr}_{0.8}\text{Y}_{0.2}\text{O}_{3-\delta}$ and a new co-author Dr. Alex J. Brown helped to analyze the NPD data. The analysis details are shown in ‘Neutron diffraction

analysis' section, new Supplementary Fig. 25, new Supplementary Note 3 and new Supplementary Tables 6,7. For the dried $\text{SrZr}_{0.8}\text{Y}_{0.2}\text{O}_{3-\delta}$ sample, refinement of the oxygen occupancies shows there are ~10% oxygen vacancies in the perovskite $\text{SrZr}_{0.8}\text{Y}_{0.2}\text{O}_{3-\square}$ *Pbnm* phase, which is consistent with the expected number of vacancies given the Zr/Y cation ratio and consistent with the oxygen occupancy obtained by the XRD Rietveld refinement. For the deuterated sample, i.e., washed $\text{SrZr}_{0.8}\text{Y}_{0.2}\text{O}_{3-\square}$ powders further hydrothermally deuterated at 225 °C for 12 h, refinement of the oxygen occupancies was found to converge towards unity, consistent with uptake of OD into vacancies. Deuterium was found to stably refine to a position of (0.677(6), 0.141(6), 0.042(4)), with the corresponding R_{wp} decreasing from 4.121% to 4.031% after the addition of D in the structure. The refined D occupancy of 0.042(5) is consistent with the approximate amount of OH (OD) expected from the number of oxygen vacancies given refinement of the anhydrous structure above. The O-D lengths refined to a value of 1.05(4) Å, consistent with expected O-H(D) lengths and with other reported O-H(D) containing perovskites. Several pieces of information give us confidence to state the presence of bulk OH and/or OD in the hydrated sample, which explains that the considerable conductivity of the hydrated $\text{SrZr}_{0.8}\text{Y}_{0.2}\text{O}_{3-\square}$ sample is indeed partly derived from the migration of protons and OH^- in the lattice.

Additionally, the physical quantities such as δ , t in $t\text{Na}^+$, R , T , t in $t\text{OH}^-$, x , y and z in $\text{SrZr}_{1-x}\text{Y}_x\text{O}_{3-y}(\text{OH})_{2y-z}\text{H}_2\text{O}$ should be italic. 0 in E0 and liquid in pliquid should be non-italic.

Thanks for the advice and all the errors have been corrected. We have revised the physical quantities to be italic, including those in figures. This is why most of the figures are marked as revised, but in fact the content for all the figures and tables from the previous version have not changed.

REVIEWER COMMENTS

Reviewer #4 (Remarks to the Author):

I salute the authors for their efforts. Most of my previous requests have been met. The following minor points on the neutron data analysis could be improved.

The occupancy values $g(\text{Zr})=0.8$ and $g(\text{Y})=0.2$ are questionable due to the presence of Y hydroxide impurities. The occupancies should be refined and analysed so that the overall composition is Zr:Y = 4:1. The refined occupancies are consistent with the compositional analysis and electrical neutrality?

The fitting is not very good. The refinement of the anisotropic atomic parameters of the oxygen atoms can improve the Rietveld fitting. The occupancy factor $g(\text{Sr})$ should also be refined. The units for the atomic displacement parameters should be added.

to ***Nature Communications***.

Dear Reviewers,

First of all, we sincerely thank the reviewer for the positive and constructive comments, which help to further improve the quality of this paper. The responses to the reviewer's comments below are highlighted in blue. The quality of the paper has been further improved. We believe the current version is suitable to be accepted for publication in Nature Communications.

REVIEWER COMMENTS

Reviewer #4 (Remarks to the Author):

I salute the authors for their efforts. Most of my previous requests have been met. The following minor points on the neutron data analysis could be improved.

We sincerely thank the reviewer for the affirmation of our previous revision and the comments on the minor points concerning the neutron diffraction refinements.

The occupancy values $g(\text{Zr})=0.8$ and $g(\text{Y})=0.2$ are questionable due to the presence of Y hydroxide impurities. The occupancies should be refined and analysed so that the overall composition is Zr:Y = 4:1.

Following the reviewer's advice, we refined the oxygen with anisotropic displacement

parameters for both the dried and deuterated samples, giving some improvements in the fits. From the new fit for the deuterated sample, the phase fraction of $\text{Y}(\text{OH})_3$ is about 6.9%. Considering the very low solubility of $\text{Y}(\text{OH})_3$ in water, the loss of Y content during the hydration and hydrothermal deuteration process of the $\text{SrZr}_{0.8}\text{Y}_{0.2}\text{O}_{3-\delta}$ sample is negligible. Therefore, it is reasonable to infer that the overall Zr:Y ratio for the deuterated sample (93.1% of perovskite phase and 6.9% of $\text{Y}(\text{OH})_3$) should still be 4:1, then the approximate Zr/Y ratio in the primary perovskite phase is calculated to be Zr 0.86, Y 0.14. This ratio is consistent with the chemical formula of perovskite phase $\text{SrZr}_{0.86}\text{Y}_{0.14}\text{O}_{2.93}$ we determined from XRD refinement (as mentioned in 'TG and Raman spectra analysis' section and Supplementary Table 1). As pointed out in the main text, neutron diffraction is not particularly sensitive to differentiate between Zr/Y due to their similar neutron scattering lengths. Manually changing the Zr/Y ratios was not found to significantly alter the statistical quality of the fit. Therefore, the Zr and Y occupancy was fixed to 0.86 and 0.14 respectively in the subsequent refinement, which gives a 4:1 of Zr:Y ratio for the total deuterated sample.

The refined occupancies are consistent with the compositional analysis and electrical neutrality?

With the anisotropic oxygens the deuterium occupancy slightly increased closer to 0.07, i.e. total D content close to 0.14 per formula unit. The strontium site occupancy was refined to a value very close to one and was therefore fixed to a value of one. Refinement of the oxygen occupancies were found to converge towards unity, consistent with uptake of OD into vacancies. Therefore the final refined structure has an approximate composition of $\text{SrZr}_{0.86}\text{Y}_{0.14}\text{O}_{2.86}(\text{OD})_{0.14}$ which is consistent with charge neutrality and the rest of our compositional characterisations. Details are given in the 'Neutron diffraction analysis' section.

For the dried $\text{SrZr}_{0.8}\text{Y}_{0.2}\text{O}_{3-\delta}$ sample, the refinement of the anisotropic atomic parameters of the O1 site improved the Rietveld fitting of the perovskite phase. The O1 site was found to refine close to a value of unity, hence was fixed to one. The O2 site refined to an occupancy of 0.963(6), which gives an overall oxygen content of 2.93 within error i.e., $\delta = 0.07$. Note we did not refine the O2 oxygen site with anisotropic displacement parameters for the dried sample due to the possibility of correlation with the oxygen partial occupancies. Then the occupancies of Zr and Y were fixed to values of 0.86 and 0.14 respectively which were calculated from the oxygen vacancies to ensure electrical charge neutrality. The occupancy of the strontium site was refined to be close to one and was therefore set to one. Details are given in the Supplementary Note 3. The compositional analysis for the dried sample $\text{SrZr}_{0.86}\text{Y}_{0.14}\text{O}_{2.93}$ is consistent with the XRD refinement on dried sample and neutron diffraction analysis on the deuterated sample.

The fitting is not very good. The refinement of the anisotropic atomic parameters of the

oxygen atoms can improve the Rietveld fitting.

As mentioned above, we refined the oxygen with anisotropic displacement parameters for both the dried and deuterated samples to improve the Rietveld fitting. Additionally, the $Y(OH)_3$ secondary phase for the deuterated sample has relatively broad peaks, we improved the fit to these peaks with the use of a Gaussian microstrain term. The new fits are shown in Supplementary Fig. 25 and Supplementary Tables 6,7. The anisotropic displacement parameters refined for the oxygen site are also listed in the Supplementary Tables 6,7.

The occupancy factor $g(\text{Sr})$ should also be refined.

The Sr A-site occupancy was refined for both the dried and deuterated samples. We found in both cases the Sr occupancy refined to values very close to or just above 1, hence the occupancy was fixed to a value of 1.

The units for the atomic displacement parameters should be added.

The unit for the atomic displacement parameters has been added to the neutron diffraction refinement tables (Supplementary Tables 6,7).

REVIEWERS' COMMENTS

Reviewer #4 (Remarks to the Author):

Now the authors have successfully analysed the neutron diffraction data. I think that the present MS is acceptable for publication in this journal.

Manuscript ID: NCOMMS-22-36610F

Title: A fast ceramic mixed OH/H^P ionic conductor for low temperature fuel cells

to ***Nature Communications***.

Dear Reviewers,

We sincerely appreciate the reviewer for the positive and constructive comments, which help to further improve the quality of this paper. The responses to the reviewer's comments below are highlighted in blue.

REVIEWERS' COMMENTS

Reviewer #4 (Remarks to the Author):

Now the authors have successfully analysed the neutron diffraction data. I think that the present MS is acceptable for publication in this journal.

We sincerely thank the reviewer for the affirmation of our revision.